# CIFAR-10-WAREHOUSE: BROAD AND MORE REALISTIC TESTBEDS IN MODEL GENERALIZATION ANALYSIS

**Xiaoxiao Sun**[1*]**, Xingjian Leng**[1*]**, Zijian Wang**[2*]**, Yang Yang**[1*]**, Zi Huang**[2]**, Liang Zheng**[1]

[1] The Australian National University

[2] The University of Queensland

{first-name.last-name}@anu.edu.au[1]
zijian.wang@uq.edu.au, huang@itee.uq.edu.au

## ABSTRACT

Analyzing model performance in various unseen environments is a critical research problem in the machine learning community. To study this problem, it is important to construct a testbed with out-of-distribution test sets that have broad coverage of environmental discrepancies. However, existing testbeds typically either have a small number of domains or are synthesized by image corruptions, hindering algorithm design that demonstrates real-world effectiveness. In this paper, we introduce CIFAR-10-**W**arehouse, consisting of 180 datasets collected by prompting image search engines and diffusion models in various ways. Generally sized between 300 and 8,000 images, the datasets contain natural images, cartoons, certain colors, or objects that do not naturally appear. With CIFAR-10-W, we aim to enhance the evaluation and deepen the understanding of two generalization tasks: domain generalization and model accuracy prediction in various out-of-distribution environments. We conduct extensive benchmarking and comparison experiments and show that CIFAR-10-W offers new and interesting insights inherent to these tasks. We also discuss other fields that would benefit from CIFAR-10-W. Data and code are available at `https://sites.google.com/view/CIFAR-10-warehouse/`.

## 1 INTRODUCTION

Analyzing and improving the generalization ability of deep learning models under out-of-distribution (OOD) environments has been of keen interest in the machine learning community. On various OOD test sets, accuracy prediction (AccP) (Deng & Zheng, 2021) investigates unsupervised risk proxies correlated with model accuracy, while domain generalization (DG) (Muandet et al., 2013; Min et al., 2022; Kim et al., 2023) aims to improve the average model accuracy when taking knowledge acquired from an arbitrary number of related domains and apply it to previously unseen domains. Their algorithm design and evaluation rely on datasets that have multiple OOD domains.

In the community, such multi-domain datasets exist, but they have their respective limitations. For example, PACS and DomainNet (Peng et al., 2019), commonly used in DG, contain 4 and 6 domains, respectively. While their images are from the real world, the small number of domains may limit the effectiveness and generalizability of the algorithms. In comparison, CIFAR-10-C (Hendrycks & Dietterich, 2019) and ImageNet-C (Hendrycks & Dietterich, 2019) have more domains, *i.e.,* 50 and 75, respectively, but both are synthetic and have limited reflection on real-world scenarios. In the iWILDs-Cam dataset (Beery et al., 2021), there are 323 real-world domains captured by different cameras, but it was originally intended for animal counting: there are typically multiple objects in an image, and object categories in each domain are incomplete. As such, existing works (Miller et al., 2021) usually merge the 323 domains into a few (*e.g.*, 2) for label space completeness.

To address the lack of appropriate multi-domain datasets, this paper introduces CIFAR-10-Warehouse, or CIFAR-10-W, a collection of 180 datasets, where each dataset has the same categories as CIFAR-10 and is viewed as a different domain. Specifically, 143 of them are real-world ones, collected by searching various image-sharing platforms with various text prompts, such as *a cartoon deer*

---

*Equal contribution.

Table 1: **Dataset comparison**. We list key statistics of CIFAR-10-W and existing alternatives commonly used for accuracy prediction (AccP) and domain generalization (DG). CIFAR-10-W is advantageous in its larger number of real-world domains. The synthetic CIFAR-10 testbeds (*e.g.*, CIFAR-10-C̄) may have infinitely many domains by varying corruption types and intensity.

| Datasets | # domains | # test images | # classes | corrupted? | image size | description |
|---|---|---|---|---|---|---|
| CIFAR-10.1 (Recht et al., 2018) | 1 | 2,000 | 10 | No | $32 \times 32$ | AccP |
| CIFAR-10.2 (Lu et al., 2020) | 1 | 2,000 | 10 | No | $32 \times 32$ | |
| CIFAR-10-C̄ (Mintun et al., 2021) | 50 | 500,000 | 10 | Yes | $32 \times 32$ | AccP |
| CIFAR-10.1-C̄ (Mintun et al., 2021) | 50 | 100,000 | 10 | Yes | $32 \times 32$ | AccP |
| CIFAR-10.2-C̄ (Mintun et al., 2021) | 50 | 100,000 | 10 | Yes | $32 \times 32$ | AccP |
| CIFAR-10-C (Hendrycks & Dietterich, 2019) | 19 | 950,000 | 10 | Yes | $32 \times 32$ | AccP & DG |
| CIFAR-10.1-C (Hendrycks & Dietterich, 2019) | 19 | 190,000 | 10 | Yes | $32 \times 32$ | AccP & DG |
| ImageNet-C (Hendrycks & Dietterich, 2019) | 75 | 3,750,000 | 1000 | Yes | $224 \times 224$ | - |
| Colored MNIST (Arjovsky et al., 2019) | 3 | 70,000 | 2 | No | $28 \times 28$ | |
| Rotated MNIST (Ghifary et al., 2015) | 6 | 70,000 | 10 | No | $28 \times 28$ | |
| VLCS (Fang et al., 2013) | 4 | 10,729 | 5 | No | $224 \times 224$ | |
| Office-Home (Venkateswara et al., 2017) | 4 | 15,588 | 65 | No | $224 \times 224$ | |
| PACS (Li et al., 2017a) | 4 | 9,991 | 7 | No | $224 \times 224$ | DG |
| Terra Incognita (Beery et al., 2018) | 4 | 24,788 | 10 | No | $224 \times 224$ | |
| DomainNet (Peng et al., 2019) | 6 | 586,575 | 345 | No | $224 \times 224$ | |
| **CIFAR-10-W** | **180** | 608,691 | 10 | No | $224 \times 224$ | AutoEval & DG |

or *a yellow dog*. The rest 37 are generated using stable diffusion (Rombach et al., 2022), using natural or unnatural prompts. CIFAR-10-W has a total of 608,691 images, and each domain typically has 300 to $8,000$ images. In Table 1, we summarize CIFAR-10-W and several notable datasets that can be utilized to evaluate AccP and DG methods. It is important to highlight that datasets commonly used for AccP tasks typically consist of a single set, like CIFAR-10.1 (Recht et al., 2018) and CIFAR-10.2 (Lu et al., 2020), or multiple sets generated by corrupting one single dataset. In comparison, CIFAR-10-W has more domains with a broad distribution coverage, in which most are real-world, thus offering an ideal test bed for generalization studies.

Domains in CIFAR-10-W have a broad distribution coverage of the 10 classes in the original CIFAR-10, such as colors, image styles and unnatural compositions. Moreover, most datasets in CIFAR-10-W are composed of real-world images; the rest are generated by stable diffusion. This allows us to study model generalization on a broad spectrum of distributions. Further, each domain in CIFAR-10-W covers images from all ten classes while keeping a moderate extent of the imbalance ratio of class distribution. The latter is useful when such data-centric research has not reached full maturity, and we can always create class absence from these data.

We conducted benchmarking of popular accuracy prediction and domain generalization methods on CIFAR-10-W, resulting in interesting observations. Specifically, we found that domain generalization methods consistently improve performance over the baseline on near-OOD datasets, but their effectiveness decreases on far-OOD datasets. Furthermore, we discover obtaining accurate accuracy predictions becomes more challenging for sets that exhibit a significant domain gap with the classifier training set. Lastly, we discussed the potential benefits of CIFAR-10-W for other research fields.

## 2 DATA COLLECTION

**Diffusion model generated data.** CIFAR-10-W includes 37 datasets generated by Stable-diffusion-2-1 (Rombach et al., 2022). Among them, 12 sets (CIFAR-10-W DF) are generated by using promopt *'high quality photo of {color}{class name}'*, where color is chosen from the 12 options shown in Fig. 1 **(A)**. Besides, we add 'cartoon' in the prompts, *i.e.*, *'high quality cartoon photo of {color}{class name}'*, to generate another 12 sets (CIFAR-10-W DF.c). In addition, we use some special prompts in which the background, style and target objects do not naturally co-exist, to generate 13 sets (CIFAR-10-W DF.h). Details of these unnatural prompts are provided in the Appendix.

**Real-world searched data.** CIFAR-10-W consists of 143 datasets that are collected through targeted keyword searches with specific conditions, such as color or style (cartoon). These searches were conducted across seven different search engines, including Google, Bing, Baidu, 360, Sogou, and stock photography/footage providing website, Pexels, as well as a photo/video sharing social website, Flickr. Fig. 1 **(A)** illustrates the color options utilized for image searching. Among the search engines,

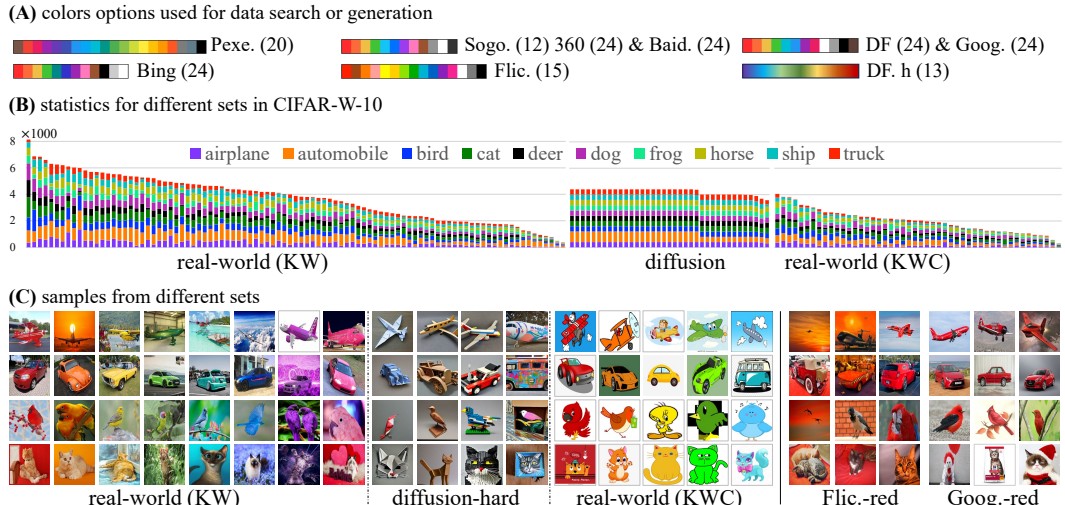

**(A)** colors options used for data search or generation

Pexe. (20)  Sogo. (12) 360 (24) & Baid. (24)  DF (24) & Goog. (24)

Bing (24)  Flic. (15)  DF. h (13)

**(B)** statistics for different sets in CIFAR-W-10

airplane ◼ automobile ◼ bird ◼ cat ◼ deer ◼ dog ◼ frog ◼ horse ◼ ship ◼ truck

real-world (KW)  diffusion  real-world (KWC)

**(C)** samples from different sets

real-world (KW)  diffusion-hard  real-world (KWC)  Flic.-red  Goog.-red

Figure 1: **Colors, sources, statistics, and examples of CIFAR-10-W**. (**A**) Datasets of CIFAR-10-W are collected from 8 sources: 7 search engines (*e.g.,* Google and Bing) and the diffusion model, where numbers after each source denote the number of datasets. We also depict color and style options used in prompting search and generation. (**B**) Distribution of the number of images for each category in datasets searched by keywords (KW), keywords plus cartoon (KWC) and diffusion under specific color conditions. (**C**) Sample images from different domains are shown.

Baidu, 360, and Sogou offer the same set of 12 color options, which differ by one or two colors compared to those provided by Google and Bing. Flickr provides 15 color options, while Pexels offers 20 color options. Additionally, for each color in Google, Bing, Baidu, and 360, an additional search is conducted using the category name followed by the term 'cartoon' to retrieve a separate dataset (indicated by ×2 in Fig. 1). Finally, there are 95 sets searched by **k**ey**w**ords with color options (hereafter called CIFAR-10-W KW) and 48 sets searched by **k**ey**w**ords with **c**artoon and color options (hereafter called CIFAR-10-W KWC).

**Dataset statistics.** We carefully create the CIFAR-10-W dataset by manually removing noisy data from all sets, and the resulting numbers of images per dataset/category are shown in Fig. 1 (**B**). For each set, the minimum number of images is 300, and the maximum 8,000, while most are between 1,000 and 6,000. There is also a moderate extent of class imbalance with the real-world datasets, with varying numbers of instances across different categories. In Fig. 1 (**C**), we provide sample images from CIFAR-10-W. We can see that images of the same category but searched by different colors exhibit distinct content. Moreover, images of the same category searched using one of the color options also showcase notable variations (*e.g.*, Flic.-red). In addition, using the same search keyword and color option results in different images on different platforms (*e.g.*, Flic.-red vs. Goog.-red).

**Privacy and license.** Some privacy information may be presented in CIFAR-10-W, such as license plate on the *vehicle*. To address potential privacy concerns, we manually blur the human faces and license plates. CIFAR-10-W inherits the licenses from its respective sources, provided that those licenses are explicitly stated. In cases where a license is not explicitly mentioned, CIFAR-10-W is distributed under license CC BY-NC 4.0 [1], which restricts its use for non-commercial purposes.

## 3 TASK I: MODEL ACCURACY PREDICTION ON UNLABELED SETS

### 3.1 BENCHMARKING SETUP

**Datasets.** We use CIFAR-10 dataset to train the classifiers to be evaluated and the 180 datasets in CIFAR-10-W as test sets, where all images are resized to the size of $32 \times 32$ (experiments of large-size $224 \times 224$ images are provided in the Appendix). Meanwhile, we use all the 188 corrupted sets from CIFAR-10-C, CIFAR-10.1-C, and CIFAR-10.2 as test sets for comparison. Because the

---

[1]https://creativecommons.org/licenses/by-nc-sa/4.0/

188 sets are all corrupted versions of CIFAR-10, CIFAR-10.1, and CIFAR-10.2, we name them collectively as CIFAR-10 corruption series or CIFAR-10-Cs hereafter.

**Classifiers.** We conducted evaluations on a total of 73 classifiers (among them, 30 are evaluated in the Appendix), including ResNet, VGG, DenseNet, and others.

**Methods to be evaluated.** We evaluate 8 existing accuracy prediction methods on CIFAR-10-Cs and CIFAR-10-W. These methods can be broadly categorized into two groups: **prediction score-based** and **feature-based**. Prediction score-based methods use the final layer output, *i.e.*, Softmax to connect with classification accuracy. They include prediction score (Pred. score), entropy (Hendrycks & Gimpel, 2017), average thresholded confidence with maximum confidence score function (ATC-MC) (Garg et al., 2022), difference of confidence (DoC) (Guillory et al., 2021), and nuclear norm (Deng et al., 2023). Feature-based methods use the output from the penultimate layer to design a dataset representation related to model accuracy on each test set, including Fréchet distance (FD) (Deng & Zheng, 2021) and Bag-of-Prototypes (BoP) (Tu et al., 2023). Besides, we modify the model-centric method 'agreement on the line' (AoL) (Baek et al., 2022) for accuracy prediction on **m**ultiple unseen **s**ets for a given model, which is referred to as MS-AOL.

**Metrics.** The mean absolute error (MAE) is used to measure error between the predicted and ground-truth accuracy. A lower MAE indicates a more precise AccP and vice versa. Spearman's rank correlation $\rho$ (Spearman, 1961) is used to quantify the correlation strength between different accuracy indicators and ground-truth accuracy. When using MAE as a metric, the leave-one-out evaluation strategy is implemented. This involves training a linear regressor using 179 sets and then using the trained regressor to predict the accuracy of the remaining single test set. MAE (%) of each accuracy prediction method is calculated on the 180 test sets. Please note that error bars are not relevant in the context of this task. This is attributed to the specific nature of the AccP setting, where a fixed classifier is subjected to evaluation. Consequently, the AccP methods do not have variables, culminating in a variance of zero in their results of five repeated runs.

### 3.2 BENCHMARKING RESULTS AND MAIN OBSERVATIONS

In Table 2, we compare the performance of eight accuracy prediction methods on both existing datasets CIFAR-10-Cs and the newly collected CIFAR-10-W. The benchmarking results provide valuable insights and allow us to analyze the performance of these methods.

**CIFAR-10-W offers a more challenging testbed for AccP methods compared to commonly used synthetic test sets.** As indicated in Table 2, the baseline methods typically exhibit higher MAE on CIFAR-10-W compared to commonly used synthetic datasets. Specifically, when using different methods to predict accuracy on various test sets using the ResNet44 classifier, the average MAE values for all methods are 3.62% on CIFAR-10-Cs, while being much higher on CIFAR-10-W, *i.e.*, 6.98%, 5.26%, 9.14%, and 6.65% for test sets generated by diffusion, searched by keywords (KW), searched by keywords plus cartoon (KWC), and all 180 test sets. We observe a consistent trend when evaluating other classifiers, where AccP methods exhibit lower accuracy on the more real-world and diverse CIFAR-10-W dataset suite compared to the corrupted sets. Specifically, Fig 2 (right) shows the correlation between accuracy and MS-AoL accuracy prediction error for CIFAR-10-Cs and CIFAR-10-W datasets, respectively. The broader data spread on the y-axis of CIFAR-10-W suggests it presents more challenges. All these results suggest that the complexity and diversity of CIFAR-10-W pose additional challenges for AccP methods, making them less accurate in predicting model performance on such diverse and real-world datasets compared to the corrupted sets.

**Superior accuracy prediction methods are more consistently reflected on CIFAR-10-W.** When evaluating on CIFAR-10-Cs (188 sets, third column in Table 2), the best-performing methods are Pred.S (0.9), BoP, and BoP for ResNet44, RepVGG-A0, and ShulffleNetV2 classifiers, respectively. However, when testing on CIFAR-10-W (180 sets, last column), MS-AOL performs the best among the three classifiers. Meanwhile, when conducting a more comprehensive evaluation with 40 classifiers, MS-AOL remains the best on both CIFAR-10-Cs and CIFAR-10-W, which is consistent with the results of single classifiers on CIFAR-10-W. Also, the performance of other methods such as BoP and nuclear norm is also stable on CIFAR-10-W.

**The more dissimilar the distributions of the classifier training set and the unseen test sets are, the more challenging it becomes for good accuracy prediction results.** AccP methods tend to

Table 2: **Evaluation of AccP methods on CIFAR-10 Cs and CIFAR-10-W**. $C_{Cs}^{10}$ and $C_W^{10}$ denote CIFAR-10 Cs and CIFAR-10-W, *resp*. We use MAE (%) to indicate estimation precision. Besides individually reporting accuracy prediction results for three classifiers, we also provide average results of (40 classifiers). For each classifier or "avg multiple classifiers (40)", the best and second best methods for each domain category are highlighted in **blue** and **bold**, respectively.

| classifier | Method | $C_{Cs}^{10}$ All | $C_w^{10}$- diffusion (37) DF.h | DF.c | DF | All | $C_w^{10}$- KW (95) Goog. | Bing | Baid. | 360 | Sogo. | Flic. | Pexe. | All | $C_w^{10}$- KWC (48) Goog. | Bing | Baid. | 360 | All | $C_w^{10}$ All |
|---|---|---|---|---|---|---|---|---|---|---|---|---|---|---|---|---|---|---|---|---|
| ResNet44 | Pred. s (0.7) | 3.46 | 9.61 | 6.63 | 6.40 | 7.60 | 5.53 | 6.48 | 6.59 | 5.16 | 7.55 | 6.81 | 6.05 | 6.30 | 11.37 | 10.35 | 13.54 | 10.76 | 11.50 | 7.96 |
| | Pred. s (0.8) | 3.37 | 9.38 | 6.36 | 5.47 | 7.13 | 5.30 | 6.82 | 6.46 | 4.52 | 7.46 | 6.31 | 5.29 | 5.97 | 10.45 | 10.36 | 12.53 | 10.11 | 10.86 | 7.51 |
| | Pred. s (0.9) | 3.06 | 9.07 | 5.70 | 4.94 | 6.64 | 4.35 | 6.00 | 5.69 | 4.36 | 7.29 | 5.71 | 4.33 | 5.31 | 9.67 | 10.46 | 10.47 | 8.83 | 9.86 | 6.80 |
| | Entropy | 3.14 | 9.35 | 6.51 | 5.55 | 7.19 | 4.85 | 6.10 | 5.99 | 4.67 | 7.62 | 6.43 | 5.33 | 5.83 | 10.58 | 10.91 | 12.28 | 10.37 | 11.03 | 7.50 |
| | ATC-MC | 3.10 | 9.06 | 5.81 | 5.01 | 6.69 | 4.40 | 5.77 | 5.85 | 4.59 | 7.40 | 5.95 | 4.62 | 5.45 | 10.08 | 10.34 | 10.65 | 9.21 | 10.07 | 6.94 |
| | DoC | 3.13 | 9.34 | 6.19 | 5.43 | 7.05 | 4.76 | 6.29 | 6.16 | 4.74 | 7.28 | 6.34 | 5.12 | 5.77 | 10.21 | 10.31 | 12.33 | 9.92 | 10.69 | 7.35 |
| | Nul.norm | 3.92 | 4.22 | 3.33 | 8.35 | 5.27 | 3.03 | 11.81 | 7.85 | 5.72 | 3.52 | 2.12 | 2.87 | 4.97 | 6.22 | 5.51 | 9.93 | 7.53 | 7.30 | 5.65 |
| | FD | 5.76 | 6.08 | 5.98 | 14.43 | 8.75 | 4.65 | 11.28 | 8.49 | 6.35 | 4.02 | 3.22 | 4.56 | 5.87 | 5.94 | 7.71 | 13.65 | 9.63 | 9.23 | 7.36 |
| | BoP+JS | 3.41 | 4.34 | 2.81 | 6.41 | 4.52 | 2.31 | 10.94 | 5.52 | 5.01 | 5.50 | 4.45 | 1.67 | 4.75 | 3.33 | 5.25 | 7.78 | 5.12 | 5.37 | 4.87 |
| | MS-AoL | 3.80 | 8.35 | 11.66 | 6.80 | 8.92 | 2.20 | 2.72 | 4.10 | 3.58 | 1.84 | 2.33 | 0.84 | 2.37 | 4.78 | 4.84 | 7.57 | 4.87 | 5.51 | 4.56 |
| | Avg | 3.62 | 7.88 | 6.10 | 6.88 | 6.98 | 4.14 | 7.42 | 6.27 | 4.87 | 5.95 | 4.97 | 4.07 | 5.26 | 8.26 | 8.60 | 11.07 | 8.63 | 9.14 | 6.65 |
| RepVGG-A0 | Pred. s (0.7) | 5.92 | 8.44 | 6.24 | 6.53 | 7.11 | 2.88 | 4.82 | 4.41 | 3.70 | 3.92 | 5.45 | 6.06 | 4.63 | 9.37 | 7.48 | 11.63 | 10.01 | 9.62 | 6.47 |
| | Pred. s (0.8) | 4.81 | 7.87 | 6.58 | 6.16 | 6.89 | 3.22 | 5.26 | 4.49 | 3.59 | 4.01 | 4.71 | 4.94 | 4.38 | 8.69 | 7.78 | 10.40 | 9.70 | 9.14 | 6.17 |
| | Pred. s (0.9) | 3.90 | 7.13 | 6.62 | 5.84 | 6.55 | 3.34 | 4.57 | 4.43 | 3.36 | 3.82 | 4.63 | 3.97 | 4.03 | 8.08 | 7.23 | 9.13 | 8.12 | 8.14 | 5.64 |
| | Entropy | 5.81 | 7.76 | 6.45 | 6.65 | 6.98 | 2.95 | 4.41 | 4.35 | 3.79 | 4.01 | 4.99 | 5.55 | 4.42 | 9.21 | 7.29 | 11.61 | 10.38 | 9.62 | 6.33 |
| | ATC-MC | 3.80 | 7.05 | 6.74 | 5.69 | 6.51 | 3.38 | 4.28 | 4.57 | 3.49 | 3.86 | 4.78 | 3.55 | 3.98 | 7.53 | 6.85 | 8.49 | 7.97 | 7.71 | 5.49 |
| | DoC | 5.23 | 7.95 | 6.30 | 6.33 | 6.89 | 3.04 | 4.61 | 4.41 | 3.62 | 3.89 | 5.05 | 5.38 | 4.40 | 9.29 | 7.29 | 10.81 | 9.92 | 9.33 | 6.23 |
| | Nul.norm | 4.03 | 4.37 | 3.34 | 7.66 | 5.10 | 3.13 | 8.51 | 7.07 | 5.46 | 2.11 | 2.33 | 2.70 | 4.26 | 5.27 | 5.70 | 9.56 | 7.53 | 7.01 | 5.17 |
| | FD | 5.53 | 7.33 | 3.23 | 12.08 | 7.54 | 4.50 | 9.89 | 8.07 | 5.35 | 3.76 | 3.96 | 4.62 | 5.59 | 6.46 | 7.25 | 12.46 | 9.30 | 8.87 | 6.86 |
| | BoP | 3.25 | 4.61 | 2.06 | 8.45 | 5.03 | 1.53 | 9.19 | 5.60 | 3.64 | 5.27 | 4.53 | 2.50 | 4.43 | 3.99 | 4.20 | 6.35 | 4.86 | 4.85 | 4.66 |
| | MS-AoL | 3.86 | 8.55 | 6.72 | 2.93 | 6.14 | 2.18 | 3.37 | 3.60 | 3.01 | 3.42 | 3.11 | 1.74 | 2.83 | 5.89 | 4.58 | 6.10 | 4.22 | 5.20 | 4.14 |
| | Avg | 4.61 | 7.11 | 5.43 | 6.83 | 6.47 | 3.01 | 5.89 | 5.10 | 3.90 | 3.81 | 4.35 | 4.10 | 4.29 | 7.38 | 6.57 | 9.65 | 8.20 | 7.95 | 5.72 |
| ShuffleNetv2 | Pred. s (0.7) | 3.48 | 9.65 | 6.02 | 6.22 | 7.36 | 3.66 | 5.57 | 5.37 | 3.36 | 4.52 | 4.87 | 2.13 | 4.06 | 6.26 | 6.23 | 9.93 | 8.26 | 7.67 | 5.70 |
| | Pred. s (0.8) | 3.26 | 9.13 | 5.71 | 5.89 | 6.97 | 3.55 | 5.12 | 5.12 | 3.66 | 4.70 | 4.72 | 1.86 | 3.93 | 5.83 | 6.08 | 10.14 | 8.07 | 7.53 | 5.52 |
| | Pred. s (0.9) | 3.08 | 8.70 | 5.82 | 5.51 | 6.73 | 3.47 | 4.54 | 4.49 | 3.67 | 4.74 | 4.29 | 1.19 | 3.57 | 5.50 | 5.06 | 8.56 | 7.05 | 6.54 | 5.01 |
| | Entropy | 3.21 | 9.40 | 5.93 | 6.19 | 7.23 | 3.46 | 4.77 | 5.09 | 3.56 | 4.42 | 4.42 | 1.42 | 3.69 | 5.45 | 5.33 | 9.24 | 8.06 | 7.02 | 5.30 |
| | ATC-MC | 3.07 | 8.66 | 5.82 | 5.76 | 6.80 | 3.39 | 5.14 | 4.43 | 3.77 | 4.40 | 4.12 | 1.31 | 3.59 | 6.01 | 4.80 | 8.50 | 7.36 | 6.67 | 5.07 |
| | DoC | 3.20 | 9.25 | 5.86 | 5.91 | 7.07 | 3.57 | 4.80 | 5.04 | 3.49 | 4.52 | 4.51 | 1.40 | 3.71 | 5.52 | 5.56 | 9.20 | 7.84 | 7.03 | 5.29 |
| | Nul.norm | 3.58 | 3.88 | 4.22 | 8.19 | 5.39 | 3.61 | 10.25 | 8.10 | 6.61 | 3.70 | 2.16 | 3.89 | 5.23 | 5.79 | 5.85 | 9.69 | 8.46 | 7.45 | 5.86 |
| | FD | 4.52 | 5.30 | 7.25 | 15.62 | 9.28 | 5.42 | 10.28 | 8.93 | 6.82 | 4.59 | 3.47 | 5.33 | 6.22 | 5.13 | 7.58 | 13.56 | 10.49 | 9.19 | 7.64 |
| | BoP | 2.90 | 5.40 | 5.92 | 12.06 | 7.73 | 2.69 | 9.77 | 6.77 | 6.89 | 4.89 | 3.04 | 1.82 | 4.78 | 2.94 | 4.89 | 9.63 | 6.80 | 6.07 | 5.73 |
| | MS-AoL | 3.00 | 8.98 | 10.22 | 5.19 | 8.15 | 2.17 | 2.04 | 4.73 | 3.73 | 1.85 | 2.59 | 1.46 | 2.55 | 5.17 | 4.02 | 7.96 | 5.52 | 5.67 | 4.53 |
| | Avg | 3.33 | 7.83 | 6.28 | 7.65 | 7.27 | 3.50 | 6.23 | 5.81 | 4.56 | 4.23 | 3.82 | 2.18 | 4.13 | 5.36 | 5.54 | 9.64 | 7.79 | 7.08 | 5.57 |
| Avg multiple classifiers (40) | Pred. s (0.7) | 4.10 | 9.18 | 7.31 | 7.36 | 7.99 | 3.64 | 5.46 | 5.33 | 4.28 | 5.69 | 5.59 | 5.02 | 5.02 | 10.87 | 9.17 | 12.23 | 10.44 | 10.68 | 7.14 |
| | Pred. s (0.8) | 3.83 | 9.25 | 7.23 | 6.95 | 7.85 | 3.48 | 5.10 | 5.25 | 4.18 | 5.45 | 5.17 | 4.37 | 4.70 | 10.41 | 8.65 | 11.68 | 9.81 | 10.14 | 6.80 |
| | Pred. s (0.9) | 3.59 | 9.23 | 7.06 | 6.68 | 7.70 | 3.34 | 4.72 | 5.10 | 3.99 | 5.22 | 4.73 | 3.67 | 4.34 | 9.84 | 8.26 | 10.87 | 9.16 | 9.53 | 6.42 |
| | Entropy | 3.97 | 9.21 | 7.17 | 7.19 | 7.89 | 3.49 | 5.11 | 5.25 | 4.21 | 5.57 | 5.29 | 4.31 | 4.73 | 10.45 | 8.67 | 11.87 | 10.13 | 10.28 | 6.86 |
| | ATC-MC | 3.74 | 9.10 | 7.21 | 6.81 | 7.74 | 3.40 | 5.02 | 5.14 | 4.03 | 5.45 | 5.04 | 4.11 | 4.57 | 10.17 | 8.51 | 11.28 | 9.57 | 9.88 | 6.64 |
| | DoC | 3.81 | 9.27 | 7.22 | 6.96 | 7.86 | 3.45 | 5.02 | 5.18 | 4.13 | 5.45 | 5.13 | 4.20 | 4.63 | 10.30 | 8.61 | 11.65 | 9.83 | 10.10 | 6.75 |
| | Nul.norm | 3.58 | 4.32 | 3.15 | 7.60 | 5.00 | 2.99 | 10.86 | 7.33 | 5.69 | 3.29 | 2.46 | 2.94 | 4.82 | 5.98 | 5.91 | 9.34 | 7.34 | 7.15 | 5.48 |
| | FD | 6.04 | 6.04 | 3.83 | 12.55 | 7.43 | 3.98 | 12.98 | 8.17 | 6.07 | 3.86 | 3.61 | 5.91 | 6.24 | 7.47 | 7.32 | 13.19 | 9.93 | 9.48 | 7.35 |
| | BoP | 3.63 | 5.94 | 4.57 | 12.67 | 7.68 | 3.25 | 10.87 | 6.57 | 5.93 | 4.85 | 3.92 | 3.07 | 5.24 | 5.13 | 5.25 | 9.85 | 6.88 | 6.78 | 6.15 |
| | BoP | 3.63 | 5.94 | 4.57 | 12.67 | 7.68 | 3.25 | 10.87 | 6.57 | 5.93 | 4.85 | 3.92 | 3.07 | 5.24 | 5.13 | 5.25 | 9.85 | 6.88 | 6.78 | 6.15 |
| | MS-AoL | 3.32 | 8.09 | 8.35 | 4.99 | 7.17 | 2.23 | 2.96 | 4.19 | 3.45 | 3.30 | 2.66 | 1.25 | 2.72 | 6.01 | 4.61 | 7.13 | 5.36 | 5.78 | 4.45 |
| | Avg | 3.96 | 7.96 | 6.31 | 7.98 | 7.43 | 3.32 | 6.81 | 5.75 | 4.60 | 4.81 | 4.36 | 3.88 | 4.70 | 8.66 | 7.50 | 10.91 | 8.85 | 8.98 | 6.40 |

face more difficulties in obtaining accurate estimations on the KWC subsets compared to KW, where cartoon images in KWC are much more differently looking to the training set, CIFAR-10, of the classifier. On KWC subsets using the ResNet-44 classifier, MAE ranges between 5.37% to 11.50%, while MAE is about 2.37% - 6.30% on the KW subsets and is lowest in the CIFAR-10-Cs, indicating that the CIFAR-10-Cs test set most closely resembles CIFAR-10 and are easier for AccP. Similar observations can be obtained on other classifiers. These results are further visualized in Fig. 2, where KWC subsets exhibit larger domain gaps (*i.e.*, higher FD) with CIFAR-10 compared to other test sets.

## 3.3 More Results and Findings

**Compared with CIFAR-10-Cs, prediction-score methods prediction on CIFAR-10-W generally has larger variance for different classifiers.** In Fig. 3(A), we observe that variance in MAE increases significantly from ∼1% to ∼6%, especially for prediction-score methods. In comparison, variance for nuclear norm, FD, BoP and MS-AoL remains at a similar level. One possible explanation

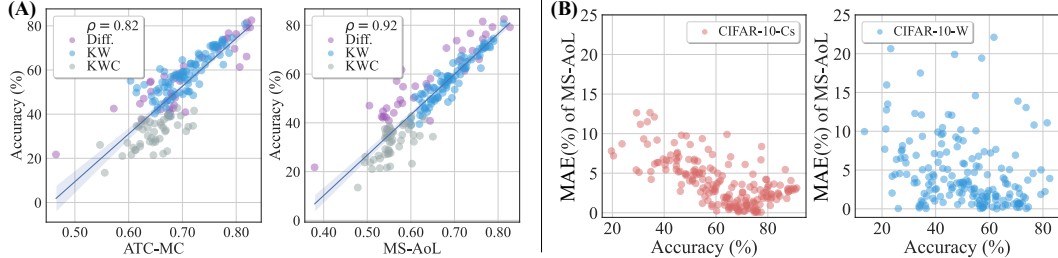

Figure 2: **Correlation studies.** (**A**) Visualizing correlation between accuracy and prediction scores on CIFAR-10-W. We use ResNet44 classifier and Spearman's rank correlation $\rho$. ATC-MC (left) and MA-AoL (right) are used. (**B**) Relationship between accuracy and accuracy prediction error (MAE, %) on the CIFAR-10-Cs (left) and CIFAR-10-W (right) testbeds. Both use the MS-AoL method.

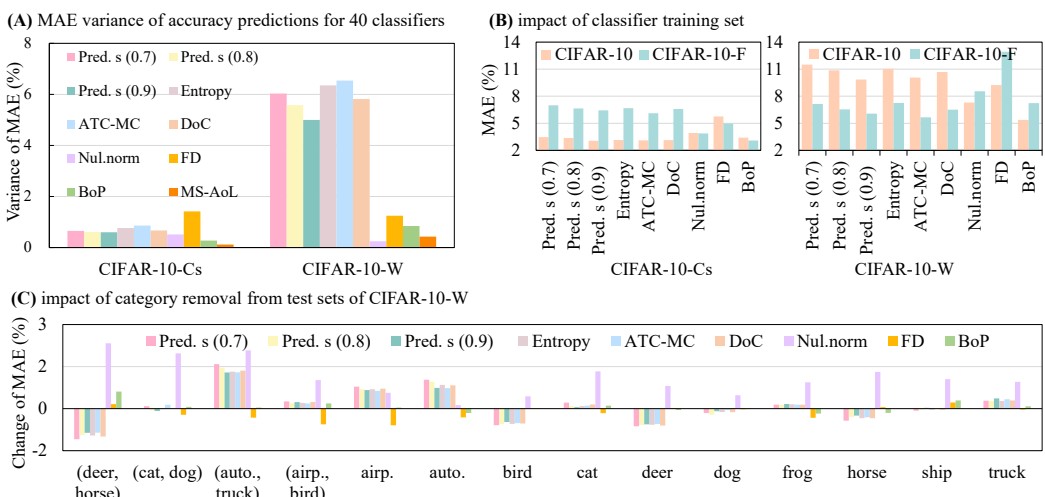

Figure 3: (**A**) Variance of MAE (%) caused by 40 classifiers. We compare the variance of different AccP methods on CIFAR-10-Cs and CIFAR-10-W. (**B**) Impact of the classifier training set: CIFAR-10 vs. CIFAR-10-F. (**C**) Impact of test category removal: the removed categories, deleted one or two at a time, are listed at the bottom. A positive change in MAE indicates worse performance and vice versa.

is that the averaging of simple scores for individual samples introduces noise when dealing with challenging test sets. In contrast, the nuclear norm considers the entire prediction matrix, making it more resilient against noise. This finding emphasizes the significance of enhancing the robustness of AccP methods for different classifiers, particularly on challenging test sets such as CIFAR-10-W.

**Impact of classifier training set.** In Fig. 3(B), when tested on CIFAR-10-Cs, using CIFAR-10 as training set gives very small MAE, suggesting that CIFAR-10 is too similar to CIFAR-10-Cs. On the other hand, when tested on CIFAR-10-W, the CIFAR-10-F (Sun et al., 2021) training set gives a lower error than CIFAR-10. This is probably because CIFAR-10-F is collected from Flickr, making it relatively more similar to CIFAR-10-W, but the overall error is much higher than testing on CIFAR-10-Cs. Interestingly, the trend of nuclear norm, FD and BoP is somehow very different from other methods. Understanding this phenomenon requires future endeavors. In all, this experiment suggests if the classifier training set is different from the test sets, AccP performance would deteriorate.

**Impact of missing test classes on accuracy prediction.** In Fig. 3(C), we remove one or two classes at a time from CIFAR-10-W and report MAE changes in accuracy prediction methods. We observe mixed results. If we remove confusing classes such as *deer* and *horse*, prediction-score methods have better performance. When removing single classes such as *cat*, *dog*, *frog* and *ship*, these prediction-score methods remain stable. In comparison, nuclear norm is sensitive to class removal, because the latter causes spurious responses to the prediction matrix. Apparently, the robustness of AccP methods against missing classes needs further study.

**Impact of test set size.** In Fig. 4(A), when we decrease the number of test images in CIFAR-10-W and CIFAR-10-Cs, the performance of accuracy prediction drops consistently for all methods. This

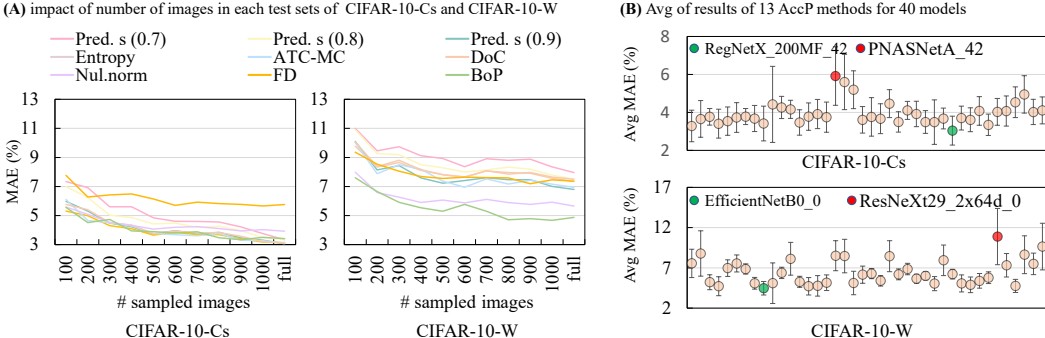

Figure 4: (**A**) Test set size on AccP methods and (**B**) Average and standard deviation of MAE values for each model of the 40 classifiers across 13 AccP methods. In (**A**), the test size is gradually reduced to 100 instances from the full dataset and the performance of methods is shown on both CIFAR-10-Cs and CIFAR-10-W. In (**B**), the easiest and hardest models to evaluate are indicated by green and red points, respectively. The ResNet44 classifier trained on CIFAR-10 is used.

is consistent with previous findings (Deng & Zheng, 2021). Besides, we observe the magnitude of performance drop is more significant on CIFAR-10-W, again demonstrating its challenging nature.

**Evaluation difficulty of different classifiers.** In Fig. 4(B), 40 classifiers are evaluated using 13 methods, and the average MAE values are displayed for each model. It can be observed that certain classifiers are more challenging to evaluate (such as PNASNetA42 and RrexNeXt29) compared to others. Additionally, the average MAE for the 40 classifiers is higher on CIFAR-10-W (2% to 12%) than on CIFAR-10-Cs (2% to 6%), which aligns with our previous observations from Table 2.

## 4 TASK II: DOMAIN GENERALIZATION

### 4.1 BENCHMARKING SETUP

**Datasets.** We use two settings: single-source DG and multi-source DG. For multi-source DG, we collected four datasets in addition to CIFAR-10-W. Two are searched from the Yandex search engine, using keywords(KW) and keywords plus cartoon (KWC), respectively. The rest two are generated using the diffusion model with the same prompts as the image search. For single-source DG, we use one of the four sets as the training set (diffusion model generated set is used in the paper. Results of the other three sets can be found in the Appendix). We train models using different DG methods on 1-4 sets/domains, respectively. A quarter of each source set is allocated for validation during training.

**Methods to be evaluated.** We conduct evaluations on both single-source DG and multi-source DG using different methods: Empirical Risk Minimization (ERM) (Gulrajani & Lopez-Paz, 2020), Style-Agnostic Networks (SagNet) (Nam et al., 2021), Self-supervised Contrastive Regularization (SelfReg) (Kim et al., 2021), Spectral Decoupling (SD) (Pezeshki et al., 2021), Fishr (Rame et al., 2022), Empirical Quantile Risk Minimization (EQRM) (Eastwood et al., 2022), Relative Chi-Square (RCS) (Chen et al., 2023), CORrelation ALignment (CORAL) (Sun & Saenko, 2016), Group-DRO (Sagawa et al., 2019), VREx (Krueger et al., 2021) and VNE (Kim et al., 2023). All these methods use the same model ResNet18 (He et al., 2016), to ensure fair comparisons and consistency.

**Evaluation Metrics.** We employ classifiers trained on the source to make predictions on CIFAR-10-W. We report the average top-1 classification accuracy (%) on 180 test sets for each DG method.

### 4.2 BENCHMARKING RESULTS AND DISCUSSIONS

**CIFAR-10-W offers a more comprehensive DG evaluation environment by providing testing domains with a wide variety of domain discrepancies.** In Table 3 and Fig. 5, it is apparent that classification accuracy on the target domain spans a wide range, from approximately 40% to 99%, where the majority of test set accuracy is between 60% and 90%. Generally, cartoon datasets tend to be more challenging, while images generated by the diffusion model are comparatively easier to recognize. This observation can be attributed to the fact that cartoons can significantly differ from

Table 3: **Benchmarking different domain generalization methods on CIFAR-10-W**. We report Top-1 classification accuracy (%) . The mean and standard deviation computed over three runs are reported in the last column. All other notations are the same as in Table 2.

| Setup | Method | $C_w^{10}$ - diffusion (37) | | | $C_w^{10}$ - KW (95) | | | | | | | $C_w^{10}$ - KWC (48) | | | | $C_w^{10}$ |
|---|---|---|---|---|---|---|---|---|---|---|---|---|---|---|---|---|
| | | DF.h | DF.c | DF | Goo | Bin | Bai | 360 | Sog | Fli | Pex | Goo | Bin | Bai | 360 | **All** |
| Single (1) DG | ERM | **79.79** | 77.00 | **95.39** | 77.89 | **85.47** | 69.66 | 72.14 | **82.78** | 86.37 | 90.85 | 48.72 | **48.01** | 39.78 | 41.39 | 72.27 ± 2.88 |
| | SagNet | 79.71 | 77.01 | 93.11 | 76.67 | 83.14 | 68.00 | 72.51 | 81.79 | 85.64 | 90.57 | 48.48 | 46.37 | 38.57 | 40.47 | 71.36 ± 3.20 |
| | SelfReg | 79.75 | **78.34** | 94.54 | 77.96 | 84.91 | 69.21 | 72.76 | 82.65 | 86.16 | 90.99 | **49.21** | 48.28 | 39.79 | **41.85** | 72.35 ± 3.69 |
| | SD | **80.94** | 77.49 | 93.50 | **78.33** | 84.73 | **69.78** | 74.43 | 83.86 | **87.53** | **91.56** | **49.21** | 47.55 | **40.17** | 41.71 | **72.70** ± 4.28 |
| | RSC | 78.15 | 75.38 | 92.05 | 71.98 | 78.73 | 65.30 | 68.41 | 78.81 | 81.92 | 87.20 | 46.22 | 43.48 | 38.18 | 38.51 | 68.63 ± 5.67 |
| Multi(2)-Source DG | ERM | 85.24 | 89.92 | 93.71 | 76.82 | 84.07 | 68.47 | 71.33 | 80.75 | 85.08 | 89.29 | 63.61 | 58.35 | 50.64 | 54.39 | 75.97 ± 1.56 |
| | SagNet | 86.30 | **91.35** | **95.05** | 79.90 | 86.31 | **71.78** | 74.60 | **83.84** | **87.64** | **91.23** | 66.69 | 60.85 | 52.33 | 57.71 | **78.37** ± 1.49 |
| | SelfReg | 86.42 | 89.64 | 93.45 | 78.20 | 84.84 | 70.20 | 73.48 | 82.59 | 87.08 | 90.91 | 65.69 | 60.53 | 52.28 | 57.56 | 77.50 ± 0.48 |
| | SD | **86.55** | 90.27 | 94.18 | **80.59** | **87.53** | **71.82** | 75.54 | 83.99 | **88.12** | **91.59** | 66.91 | 61.48 | 52.48 | 57.24 | **78.57** ± 0.97 |
| | Fishr | 85.79 | 90.06 | 94.53 | 79.07 | **86.58** | 69.67 | 72.71 | 81.86 | 86.66 | 90.28 | 63.92 | 59.61 | 50.67 | 54.27 | 76.98 ± 1.09 |
| | EQRM | 86.07 | **91.04** | **94.68** | 78.98 | 86.56 | 71.22 | 74.04 | 83.24 | 86.59 | 90.63 | **67.07** | **61.32** | **53.16** | **57.99** | 78.12 ± 1.24 |
| | RSC | 82.24 | 86.15 | 90.94 | 71.13 | 78.33 | 64.24 | 66.81 | 76.46 | 80.89 | 85.76 | 57.32 | 53.42 | 47.03 | 50.27 | 71.68 ± 1.76 |
| | CORAL | 85.21 | 90.25 | 94.08 | 78.76 | 86.07 | 70.15 | 73.09 | 82.81 | 86.30 | 90.32 | 65.80 | 60.23 | 51.10 | 55.90 | 77.26 ± 0.54 |
| | VREx | 85.91 | 90.89 | 94.83 | 78.07 | 84.94 | 70.81 | 72.77 | 82.06 | 85.59 | 89.36 | 66.72 | 60.10 | **52.91** | **57.84** | 77.40 ± 1.99 |
| | GroupDRO | 85.11 | 89.49 | 93.93 | 79.41 | 86.11 | 71.02 | **75.27** | 83.21 | 87.20 | 90.63 | 64.65 | 59.42 | 51.17 | 56.02 | 77.46 ± 1.71 |
| Multi(3)-Source DG | ERM | 86.69 | 93.90 | 98.71 | 88.18 | 94.51 | 79.78 | 85.10 | 91.73 | 93.08 | 95.99 | 70.60 | 64.74 | 55.87 | 58.49 | 83.46 ± 0.84 |
| | SagNet | 86.78 | 94.74 | 98.86 | 88.57 | 93.86 | 79.91 | 84.19 | 90.88 | 92.15 | 95.41 | 71.15 | 65.16 | 56.14 | 59.51 | 83.41 ± 0.82 |
| | SelfReg | 87.58 | **95.15** | 98.87 | 89.19 | 94.75 | 80.86 | 86.14 | 92.42 | 93.26 | 96.06 | **72.88** | 66.60 | 57.53 | 61.26 | 84.48 ± 0.48 |
| | SD | **88.08** | 95.07 | **98.96** | **90.43** | **95.58** | **81.70** | **86.69** | **92.75** | **93.96** | **96.70** | **73.56** | **66.87** | **58.41** | **61.76** | **85.05** ± 0.51 |
| | Fishr | 87.38 | 94.95 | **98.99** | 89.28 | 94.24 | 80.12 | 84.28 | 91.31 | 93.19 | 96.12 | 71.84 | 66.09 | 56.76 | 60.85 | 84.00 ± 0.73 |
| | EQRM | 86.42 | 94.30 | 98.87 | 87.98 | 93.38 | 78.71 | 82.73 | 89.79 | 91.86 | 95.51 | 70.49 | 64.98 | 54.97 | 59.27 | 82.87 ± 0.97 |
| | RSC | 83.98 | 93.76 | 98.54 | 86.07 | 92.87 | 77.46 | 82.05 | 88.86 | 90.23 | 94.11 | 67.92 | 62.94 | 54.03 | 57.71 | 81.52 ± 0.60 |
| | CORAL | 86.92 | **95.12** | 98.83 | **89.51** | 94.80 | 81.00 | **86.34** | 92.63 | 93.51 | 96.58 | 72.78 | 66.19 | 57.71 | **61.27** | **84.55** ± 0.92 |
| | VREx | **87.73** | 94.75 | 98.83 | 88.74 | 94.34 | **81.06** | 85.59 | 92.08 | 93.10 | 96.06 | 72.26 | **66.65** | **57.74** | 61.23 | 84.32 ± 0.68 |
| | GroupDRO | 86.38 | 94.42 | 98.89 | 87.59 | 93.79 | 78.59 | 83.42 | 91.08 | 92.50 | 96.00 | 70.29 | 65.00 | 54.90 | 58.50 | 83.05 ± 0.75 |
| Multi(4)-Source DG | ERM | 87.49 | 95.89 | 98.77 | 89.34 | 94.49 | 83.23 | 88.04 | 92.96 | 93.45 | 96.32 | 83.88 | 76.70 | 67.53 | 74.78 | 87.85 ± 0.52 |
| | SagNet | **88.14** | 96.50 | 98.89 | 89.88 | 95.08 | 83.38 | 88.35 | 93.46 | 93.60 | 96.56 | 84.98 | 76.87 | 68.25 | 75.37 | 88.30 ± 0.39 |
| | SelfReg | 87.47 | **96.63** | 98.89 | 90.46 | 95.21 | 83.99 | 88.38 | 93.59 | 93.57 | 96.54 | 85.04 | **77.48** | **69.39** | **76.18** | 88.48 ± 0.76 |
| | SD | **88.27** | 96.56 | 98.85 | **90.81** | **95.59** | **84.66** | **89.19** | **93.99** | 94.34 | **96.96** | **85.80** | **78.19** | 69.83 | 76.47 | **89.01** ± 0.49 |
| | Fishr | 87.73 | 96.20 | **98.94** | 89.79 | 94.88 | 83.53 | 88.27 | 93.09 | 93.53 | 96.46 | 83.99 | 76.01 | 67.27 | 73.97 | 87.91 ± 0.57 |
| | EQRM | 87.56 | 96.02 | **98.90** | 89.47 | 94.54 | 82.97 | 87.51 | 92.29 | 93.13 | 96.26 | 84.10 | 76.52 | 67.27 | 74.22 | 87.70 ± 0.76 |
| | RSC | 86.31 | 95.04 | 98.48 | 87.04 | 93.17 | 80.73 | 84.37 | 90.34 | 91.43 | 94.86 | 80.84 | 74.20 | 65.10 | 71.32 | 85.77 ± 1.29 |
| | CORAL | 87.98 | 96.32 | 98.81 | **90.49** | 95.36 | 84.28 | 89.06 | 93.78 | 94.17 | 96.95 | 85.14 | 77.14 | 69.20 | 75.42 | **88.64** ± 0.54 |
| | VREx | 87.60 | 95.97 | 98.76 | 89.15 | 94.42 | 82.67 | 87.47 | 92.65 | 93.20 | 96.58 | 84.06 | 76.39 | 67.79 | 74.62 | 87.75 ± 0.41 |
| | GroupDRO | 87.06 | 95.79 | 98.64 | 88.43 | 93.88 | 82.01 | 86.89 | 91.91 | 92.48 | 95.61 | 83.87 | 75.88 | 67.24 | 73.75 | 87.17 ± 0.61 |
| | VNE | 87.06 | 95.95 | 98.81 | 89.20 | 94.78 | 83.04 | 88.21 | 92.95 | 93.08 | 96.20 | 83.51 | 75.75 | 66.97 | 73.94 | 87.61 ± 0.31 |

the source domain, making it more difficult for models to generalize. On the other hand, generated images often exhibit high qualities, featuring large and distinct objects.

**DG improvement on different domains.** In Fig. 5(A), we observe mixed results regarding the improvement brought by the SD method compared to the baseline accuracy on the 180 target sets. When using four sources, most domains show improvement. However, when using only one source, improvement is primarily seen in real domains (KW), while diffusion-generated and cartoon domains do not exhibit significant improvement. Previous works have indicated limited improvement of DG methods on datasets like PACS and DomainNet, potentially due to the small number of test domains. While CIFAR-10-W may not cover all possible target domains, including more diverse domains in the evaluation can provide valuable insights into the performance and limitations of DG methods.

**Predicting accuracy of models after domain generalization.** Considering DG techniques only update the classification model based on the source domain(s), it is possible to predict the DG model accuracy on unseen target sets using methods described in Section 3. In Fig. 5(B), we use the FD and nuclear norm methods to predict the accuracy of ResNet18 models trained or domain generalized under the single-source DG setup. Interestingly, we observe that both accuracy prediction methods exhibit similar performance, regardless of whether DG is applied. This finding suggests that AccP techniques may be applicable and effective even for domain generalized models.

## 5    OTHER FIELDS THAT POTENTIALLY BENEFIT FROM CIFAR-10-W

**Learning from noisy data.** Most existing datasets (Wei et al., 2021; Song et al., 2019) in this area are manually created *e.g.*, labels are flipped between classes (Ghosh & Lan, 2021). There exist a few

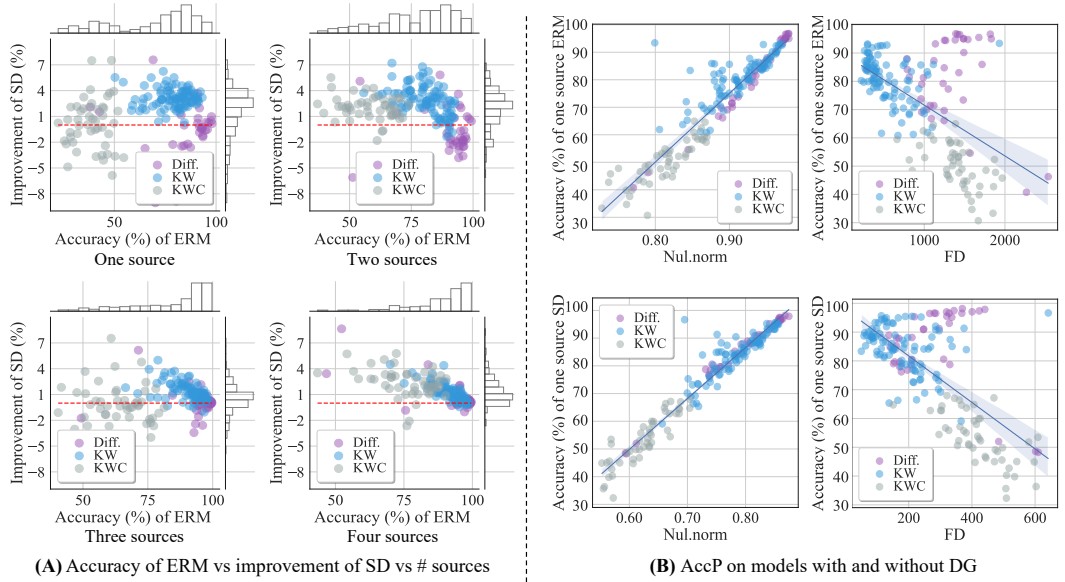

Figure 5: (**A**) Impact of increasing the number of source domains on domain generalization. The ResNet-18 classifier is trained using the domain generalization technique SD with our searched training sets as the source domains. The density plot on the y-axis illustrates the density of the test set at various levels of improvement. On the x-axis, the density plot shows the distribution of accuracies achieved by the baseline method ERM on CIFAR-10-W datasets. (**B**) Effectiveness of accuracy prediction methods (nuclear norm and FD) on CIFAR-10-W. We evaluate the performance using the ResNet-18 model trained with two different approaches: the normally trained model (top) and the model trained with the domain generalization technique SD (bottom).

real-world noisy datasets (Xu et al., 2018; Lee et al., 2018; Li et al., 2017b), but they are not cleaned, meaning that there is no ground truth to evaluate noise label spotting algorithms. In this regard, CIFAR-10-W offers a valuable real-world noisy dataset, because we have recorded the incorrectly labeled images during the cleaning and annotation process.

**Test time and unsupervised domain adaptation (DA).** Datasets in CIFAR-10-W generally have a few thousand images each, which might not be sufficient for full training. Nevertheless, it is possible to use them for unsupervised DA, where target domains with a few hundred unlabeled images target domain are often used (Csurka, 2017; Luo et al., 2023; Long et al., 2018; Wang et al., 2020). Moreover, it would be even easier to use CIFAR-10-W for test-time DA, because the latter usually assumes the use of a batch of images for online training. The broad coverage of distributions makes CIFAR-10-W an ideal testbed for evaluating and benchmarking DA algorithms.

**Out-of-distribution (OOD) detection.** In OOD detection, the in-distribution (ID) test data typically have the same distribution as the ID training data. In (Hendrycks & Dietterich, 2019; Mintun et al., 2021; Lu et al., 2020), in-distribution test data contain a few domains, which require the OOD detection algorithm to be generalizable to various ID distributions. In this regard, CIFAR-10-W will significantly expand the boundary of the ID domain.

# 6 CONCLUSION

This paper introduces CIFAR-10-**W**arehouse, a collection of 180 datasets with broad distribution coverage of the 10 categories in original CIFAR-10. Most of these datasets are real-world domains searched from various image search engines, and the rest are generated by stable diffusion. Diversity of the domains is reflected in rich color spectrum, styles, (un)naturalness and class imbalance. On CIFAR-10-W, we benchmark popular methods in accuracy prediction and domain generalization. We confirm that CIFAR-10-W creates challenging setups for the two tasks where interesting insights are observed. We also discuss other fields where this dataset can be used and believe it will contribute to model generalization analysis under more real-world and a large number of test domains.

ACKNOWLEDGEMENT

We thank Aditi Raghunathan for valuable discussions on early CIFAR-10-W prototypes, and thank the anonymous reviewers and AC for their insightful comments and suggestions that improved this paper. This research was funded in part by the ARC Discovery Project (DP210102801) to Liang Zheng, and ARC DP230101196, DP240101814, CE200100025 to Zi Huang.

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

APPENDIX

In the appendix, we present details regarding the data collection process, classifiers used for accuracy prediction evaluation, methods compared in our study, additional results and analysis.

## A  DETAILS AND MORE DISCUSSIONS OF DATASETS IN CIFAR-10-W

### A.1  PROMPTS AND DATA SOURCES

**Prompts.** In CIFAR-10-W, there are 37 sets generated by Stable-diffusion-2-1 (Rombach et al., 2022). Among these sets, 24 are generated using category names with 12 color options, where 12 sets have the keyword of 'cartoon'. Additionally, we generate 13 sets using special prompts reflecting entities that are not naturally combined. Table 4 shows the prompts.

Table 4: **Prompts** of CIFAR-10-W DF., DF. cartoon and DF. hard.

| | |
|---|---|
| Prompts of CIFAR-10-W DF. | 'High quality photo of {colour} {class-name}' |
| Prompts of CIFAR-10-W DF. cartoon | 'High quality **cartoon** picture of {colour} {class-name}' |
| Prompts of CIFAR-10-W DF. hard | 'photo of {class-name} in a cage'
'photo of {class-name} on a table'
'photo of {class-name} painted on a box'
'photo of {class-name} painted on a t-shirt'
'sketch of {class-name}'
'photo of {class-name} with messy background'
'photo of toy model of {class-name}'
'photo of 3D model of {class-name}'
'photo of origami of {class-name}'
'photo of sculpture of {class-name}'
'photo of lego model of {class-name}'
'photo of wooden model of {class-name}'
'oil painting of {class-name}' |

**Links of engines for data collection.**

Google: `https://www.google.com/imghp?hl=EN`

Bing: `https://www.bing.com/images/feed`

Baidu: `https://image.baidu.com/`, 360: `https://image.so.com/?src=tab_web`

Sogou: `https://pic.sogou.com/`, Pexels: `https://www.pexels.com/`

Flickr: `https://www.flickr.com/` and Yandex: `https://yandex.com/`

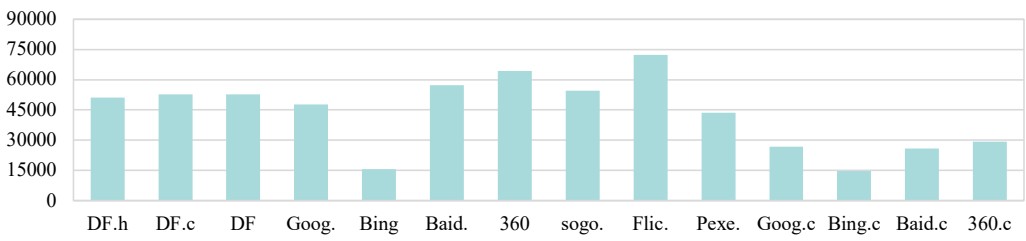

Figure 6: **Statistics of image number from different engines and diffusion models.**

**Statistics.** Fig. 6 shows the number of images from 14 origins, including 11 from search engines (*e.g.*, Google and Bing) and 3 generated by the diffusion model. Specifically, image counts from different engines range from a minimum of 14,683 in Bing cartoon to a maximum of 72,432 in Flickr, with most falling between 50,000 and 60,000.

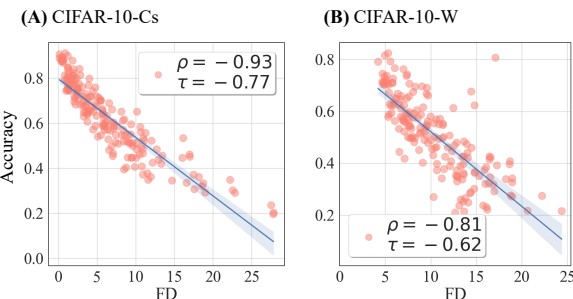

Figure 7: **Dataset distribution and performance correlation**. Correlation between model accuracy and Fréchet Distance (FD) across two settings: (A) CIFAR-10-Cs and (B) CIFAR-10-W. The FD is computed relative to the original CIFAR-10 dataset for each corresponding set within CIFAR-10-Cs and CIFAR-10-W.

**Training sets of multi-source DG.** Four sets are used: Yandex-C (category names) Yande-CC (category names with cartoon), diff-C, and diff-CC. They contain 4,857, 4,044, 4,000, and 4,000 images, respectively. For each set, we will report single-domain DG results in the following sections.

## A.2 MORE ANALYSIS OF CIFAR-10-W

### A.2.1 DIVERSITY OF DATASETS IN CIFAR-10-W

In the main paper, Fig. 2 and 5 have shown the distribution of ground truth accuracy for ResNet44 and ERM accuracy on CIFAR-10-W. These figures demonstrate varied performance across our datasets which can indicate the varied shifts among datasets. In Appendix, we provide more analysis as below:

It is, Fig. 7 shows the correlation between model accuracy and FD for both (**A**) CIFAR-10-Cs and (**B**) CIFAR-10-W. The FD is determined to the CIFAR-10 benchmark. Notably, CIFAR-10-W displays a broader range of FD values compared to CIFAR-10-Cs, indicating a larger variation in dataset distribution. For instance, CIFAR-10-W includes more sets with FD values exceeding 10. Moreover, the stronger correlation coefficients for CIFAR-10-Cs ($\rho = -0.93$ and $\tau = -0.77$) compared to CIFAR-10-W ($\rho = -0.81$ and $\tau = -0.62$) suggest that **the distribution of CIFAR-10-Cs, which comprises simulated datasets, is less complex than the real-world data variant in CIFAR-10-W**.

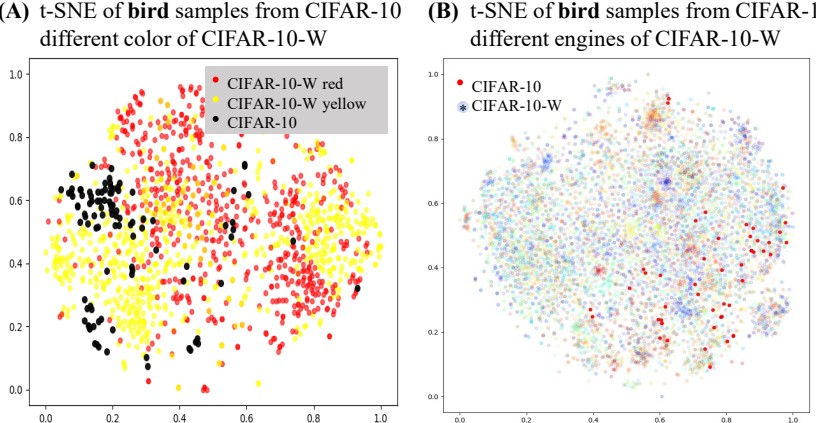

Figure 8: **t-SNE visualizations demonstrating data distribution variance**. (B) displays t-SNE plots of bird images from CIFAR-10 and the red and yellow color variations within CIFAR-10-W. (B) presents t-SNE plots of bird images from different search engines within CIFAR-10-W, alongside CIFAR-10. These visualizations highlight the broader distribution range of CIFAR-10-W compared to CIFAR-10, illustrating the diversity of the datasets.

Second, in Fig. 8, we explore the variation within CIFAR-10-W data by color and search engine, respectively. Fig. 8 (A) reveals that even within a single category—birds—the images searched with the same color option from different engines show significant variance. Notably, both the red and yellow options from CIFAR-10-W exhibit a wider distribution range than CIFAR-10, indicating the complexity of our dataset. Furthermore, images from the red and yellow categories show some

overlap showing the differences between color options within CIFAR-10-W. Fig. 8 (B) shows the diversity of bird images from different search engines in CIFAR-10-W. Here, the CIFAR-10-W data demonstrates considerable diversity, with its distribution encompassing that of the CIFAR-10. Moreover, distinct search engines within CIFAR-10-W exhibit unique data distributions, with most engines contributing their own characteristic spread.

In summary, these t-SNE visualizations confirm that CIFAR-10-W provides a rich and complex dataset that extends beyond the variability found in CIFAR-10, making it a valuable resource for evaluating AccP and DG methods.

In addition, we calculate the rank correlations of different method performances on three subgroups of data from CIFAR-10-W: (DF.h & DF), (Goog. & Flic.) and (Bing C & Baid.C) with correlations of 0.3050, 0.4384 and 0.648 There is no obvious correlation, which indicates the variance between these datasets. As for the consistency we mentioned in the paper is caused by the average statistics on **all** datasets. When methods are evaluated on **a large number of diverse datasets**, the rank of performance of methods will gradually be stable.

### A.2.2    CORRELATION BETWEEN IMAGE QUANTITY AND PERFORMANCE ACROSS ENGINES

We calculated Kendall's rank correlation coefficient between **the number of images** from each engine and **the average performance of various methods** on each engine. For AccP, the average correlation coefficients are -0.450, -0.406,-0.230 and -0.406 (number of images vs. Avg in Table 2) for ResNet44, RepVgg-A0, ShuffleNetV2 and 40 classifiers, respectively, which suggest only a weak relationship between dataset size and performance across different engines. For domain generalization (DG), the Kendall's rank correlations are 0.252, 0.252, 0.208 and 0.230 for single and multiple (2, 3 and 4) source DG, respectively, when comparing the number of images to the results of SD on different engines, reinforcing the conclusion that the performance drop is not due to the number of images.

### A.2.3    DOES CIFAR-10-W OFFER DISTINCT EVALUATIONS VERSUS CURRENT DATASETS?

For AccP methods, we calculate their rank correlations on CIFAR-10-Cs and CIFAR-10-W across different models, including ResNet44, RepVGG-A and ShuffleNetv2. The correlation $\tau$ are 0.066, 0.733and 0.644, respectively, suggesting a lack of strong rank consistency between performances on CIFAR-10-W and CIFAR-10-Cs. This is especially evident for ResNet44, where the correlation is notably low. Moreover, compared with other domain generalization benchmark datasets (usually involving less than six domains), our proposed benchmark dataset empowers researchers to conduct more statistically meaningful analyses with 180 domains. Specifically, from Table 3, we can see that with the increased number of training domain, the test performances of model tend to be more stable. For example, the variance of test performance drops from approximately 3.9% for single source domain to 0.67% for four source domains. This outcome contrasts with results reported in other studies, indicating that CIFAR-10-W may present a more challenging and realistic testing environment by involving more test sets from different domains.

These observations suggest that conclusions drawn from CIFAR-10-W could differ from those based on existing datasets. In particular, CIFAR-10-W tends to magnify performance differences among methods, providing clearer insights into their robustness for AccP. For DG tasks, the lack of noticeable improvement over the ERM baseline in our dataset suggests that existing DG methods may need further refinement to handle the diverse and complex scenarios presented by CIFAR-10-W.

In summary, CIFAR-10-W serve as a new diverse benchmark dataset, offering a wide array of test environments that are helpful for future research in AccP and DG methodologies.

### A.2.4    FAILURE CASES ON CIFAR-10-W

Our analysis of CIFAR-10-W has indeed shown new challenges encountered by both DG and AccP.

As evidenced in Tables 2 and 3 of our paper, there is notable variability in the performance rankings of AccP methods when comparing CIFAR-10-W with CIFAR-10-Cs. Specifically, the rank correlation for models such as ResNet44, RepVGG-A0, and ShuffleNetv2 is only 0.481, highlighting a significant discrepancy in evaluations between CIFAR-10-Cs and CIFAR-10-W. An example of this is the performance of the Pred. s (0.9) method. While it ranks as the best AccP method for

ResNet44 on CIFAR-10-Cs, it fails to maintain its superiority on CIFAR-10-W. This suggests that the generalizability or adaptability of Pred. s (0.9) may fail when transitioning from CIFAR-10-C to the more diverse CIFAR-10-W environment.

To study the potential failure modes, we first compare the test performance in the single source and the multiple source settings. We observe that when multiple distinct source domains are available at the training stage, it is more likely to achieve higher improvement than that of a single source domain. Furthermore, we compare the test performance of ERM and SagNet under a three-source domain regime. In the three-source domain training setting, we have two diffusion model-generated domains and a photo-realistic domain. By analysing the testing performance on three meta-categories (i.e., diffusion, KW, and KWC), we find that SagNet brings improvement on KWC (approx. +0.56% on average), but degrades the ERM result on KW (approx. -0.59% on average). This indicates that some domain generalization methods may be able to improve the generalization power of the model on domains that are largely different from training distribution, with the price of reducing some generalization power on similarly distributed domains.

### A.3 LIMITATIONS

CIFAR-10 is a relatively small dataset compared with some existing datasets, such as ImageNet. However, it is one of the most commonly used data in the community and is an important benchmark for AccP and DG algorithm evaluation. Our choice of CIFAR-10 is because of its simplicity and manageability, which makes it a suitable starting point for foundational research of AccP and DG tasks. We start by using CIFAR-10 as a basic reference to build CIFAR-10 to solve the limitation that existing test data have a small number of domains or use synthetic data for evaluation. Despite its limitations, the insights gained from CIFAR-10 can provide valuable preliminary understanding, which can be a foundation for analyzing more complex datasets. The findings of AccP and DG observed in CIFAR-10 can offer indicative trends that might be observed in larger datasets. For example, if a method cannot work well on the CIFAR-10 setup, it is hard to imagine it works well on a dataset with a large number of categories.

On the other hand, as shown in Fig. 1 (B), most datasets have 2,000 to 5,000 images. If we consider original CIFAR-10 dataset has 50,000 images, each dataset in CIFAR-10-W might not make a very good training set. However, dataset combinations might solve this limitation. On the other hand, compared with DomainNet (Peng et al., 2019) which has 300 classes, the number of classes in CIFAR-10-W is much less. In addition, we recognize that the domain coverage of CIFAR-10-W is broad but not exhaustive. We aim to publish future versions to address these points.

## B EXPERIMENTAL SETTINGS

In total, we conduct more than 1,500 experiments, run on 3 servers. Each server has 4 RTX-2080TI GPUs and a 16-core AMD Threadripper CPU @ 3.5Ghz.

**70 classifiers used in AccP.** Table 5 presents the names of the 70 classifiers. Among them, 40 are evaluated in our main paper and all are used in the Appendix. These models encompass diverse basic backbones and layers. All models are trained using PyTorch[2] on the CIFAR-10 dataset. The learning rate for all models is set to 0.01, and the number of epochs is 200 (code is from `https://github.com/kuangliu/pytorch-CIFAR`).

**The backbone used in DG.** ResNet18 is the backbone used in experiments of TaskII DG. For each method, experiments are run on 3 groups of parameters, respectively, and then select one group of parameters that works well on the in-distribution validation set. The learning rate for all experiments is set to 5e-05, and the optimizer is ADAM.

**Implementation of MS-AoL.** In our implementation of MS-AoL (Multiple Sets Agreement-on-the-Line), we adapt the original AoL method (Baek et al., 2022) to accuracy prediction (AccP) in *various environments*. AoL is originally a model-centric method that uses a pool of models, an in-distribution dataset (such as CIFAR-10), and an out-of-distribution dataset (such as CIFAR-10.1) to observe the agreement-on-the-line phenomenon. For AccP, the goal is to estimate the accuracy of a target model on *various* unlabeled test sets. In AoL, when given a new model, we can use AoL to predict

---

[2]`http://pytorch.org`

Table 5: **Classifiers evaluated in Task I.** Average results of the first 40 models have been reported in our paper. All the 70 models are used in the Appendix.

| | | | | |
|---|---|---|---|---|
| DPN26-0 | DPN26-42 | DPN92-0 | DPN92-42 | DenseNet121-0 |
| DenseNet121-42 | DenseNet169-0 | DenseNet169-42 | Efficient | NetB0-0 |
| EfficientNetB0-42 | GoogLeNet-0 | GoogLeNet-42 | MobileNetV2-0 | MobileNetV2-42 |
| MobileNet-0 | MobileNet-42 | PNASNetA-0 | PNASNetA-42 | PNASNetB-0 |
| PNASNetB-42 | PreActResNet101-0 | PreActResNet101-42 | PreActResNet18-0 | PreActResNet18-42 |
| PreActResNet34-0 | PreActResNet34-42 | PreActResNet50-0 | PreActResNet50-42 | RegNetX-200MF-0 |
| RegNetX-200MF-42 | RegNetX-400MF-0 | RegNetX-400MF-42 | RegNetY-400MF-0 | RegNetY-400MF-42 |
| ResNeXt29-2x64d-0 | ResNeXt29-2x64d-42 | ResNeXt29-32x4d-0 | ResNeXt29-32x4d-42 | ResNeXt29-4x64d-0 |
| ResNeXt29-4x64d-42 | ResNet101-0 | ResNet101-42 | ResNet18-0 | ResNet18-42 |
| ResNet34-0 | ResNet34-42 | ResNet50-0 | ResNet50-42 | SENet18-0 |
| SENet18-42 | ShuffleNetG2-0 | ShuffleNetG2-42 | ShuffleNetG3-0 | ShuffleNetG3-42 |
| ShuffleNetV2-0-5-0 | ShuffleNetV2-0-5-42 | ShuffleNetV2-1-0 | ShuffleNetV2-1-42 | ShuffleNetV2-1-5-0 |
| ShuffleNetV2-1-5-42 | ShuffleNetV2-2-0 | ShuffleNetV2-2-42 | VGG11-0 | VGG11-42 |
| VGG13-0 | VGG13-42 | VGG16-0 | VGG16-42 | VGG19-0 |

its accuracy on the original out-of-distribution dataset (CIFAR-10.1) by running the prediction only once. However, when given a new test set (such as CIFAR-10.2), all 70 models should be used to calculate their agreement on the new test set. This requires running the prediction $(70 \times 69)/2$ times. In MS-AoL, we modify the AoL method to handle multiple test sets. We run AoL 180 times using the 70 models and sample out 180 agreement values of each model on the 180 sets for AccP. MS-AoL performs well but it results in a significant computational burden.

**Implementation of Other AccP methods.** We strive to follow the same settings as described in their original papers. To ensure a fair and consistent comparison between different AccP methods, we use linear regression for all methods when using MAE as the evaluation metric.

## C   MORE EXPERIMENTAL RESULTS

### C.1   TASK I: MODEL ACCURACY PREDICTION ON UNLABELED SETS

**Results of AccP on Vggnet13 and 70 classifiers** are presented in Table 6. The trends observed in these results are consistent with those shown in the main paper. For example, the datasets in CIFAR-10-W KWC pose greater challenges for AccP. Average values of MAEs on sets of CIFAR-10-W KWC range from 4.54% to 13.74% across different methods. Peng et al. (2024) evaluate several classifiers, providing average results of different accuracy prediction (AccP) methods. However, the scope of classifiers might not be comprehensive for an extensive evaluation. To address this, expanding the number of classifiers is suggested. We will share the 70 classifiers on our project page.

**Correlation studies** are conducted to evaluate the performance of different AccP methods on three different classifiers. The results are presented in Fig. 9 and 10. On the ResNet44, RepVgg and ShufflenetV2 classifiers, the best-performing methods are MS-AoL with ($\rho$) values of 0.92, 0.95 and 0.92, respectively. On Vgg13, the best-performing method is BoP+JS with $\rho$ values of 0.94. These correlation coefficients indicate strong positive relationships between the predictions of the AccP methods and the actual accuracies of the classifiers.

**AccP on CIFAR-10-W $224 \times 224$.** Fig. 11 presents the results of different AccP methods on CIFAR-10-W with an image size of $224 \times 224$. The evaluation is performed on four classifiers trained on the CINIC dataset. The Spearman's rank correlation ($\rho$) is reported to indicate the strength of the correlation between the proxy output and classifier accuracy. When comparing these results to those obtained on CIFAR-10-W with an image size of $32 \times 32$, several trends can be observed:

**1)** *Correlation coefficients between the proxy output and classifier accuracy are generally higher on CIFAR-10-W $224 \times 224$ for different AccP methods compared to the results on CIFAR-10-W $32 \times 32$.* This observation can be attributed to several factors. Firstly, the CIFAR-10 setup consists of only 10 distinct categories, which are not very hard to distinguish from each other. This relative simplicity makes it easier for models to accurately predict the accuracy of classifiers on these datasets

Table 6: **Method comparison for predicting model accuracy on CIFAR-10-Cs and CIFAR-10-W.** We use MAE (%) to indicate the performance of accuracy estimation.

| Model | Method | $C_s^{10}$ All | $C_w^{10}$ - diffusion (37) DF.h | DF.c | DF.v | All | $C_w^{10}$ - vanilla (95) Goog. | Bing | Baid. | 360 | Sogo. | Flic. | Pexe. | All | $C_w^{10}$ - cartoon (48) Goog. | Bing | Baid. | 360 | All | $C_w^{10}$ All |
|---|---|---|---|---|---|---|---|---|---|---|---|---|---|---|---|---|---|---|---|---|
| VGGNet13 | Pred. s (0.7) | 4.28 | 13.08 | 5.00 | 5.43 | 7.98 | 5.54 | 4.07 | 7.23 | 5.30 | 5.46 | 4.61 | 6.13 | 5.51 | 12.59 | 7.27 | 11.98 | 9.47 | 10.33 | 7.30 |
| | Pred. s (0.8) | 4.11 | 12.45 | 5.08 | **4.39** | 7.44 | 5.42 | **3.56** | 6.74 | 5.65 | 4.64 | 3.92 | 4.96 | 4.95 | 11.11 | 6.85 | 10.31 | 8.88 | 9.29 | 6.62 |
| | Pred. s (0.9) | 3.64 | 12.10 | 5.57 | 4.58 | 7.55 | 4.66 | 3.59 | 6.66 | 4.93 | 5.12 | 3.66 | 4.47 | 4.67 | 11.02 | 6.15 | 11.52 | 8.26 | 9.24 | 6.48 |
| | Entropy | 4.11 | 13.39 | 5.85 | 5.88 | 8.51 | 4.95 | 3.98 | 6.86 | 5.25 | 5.61 | 4.38 | 5.16 | 5.14 | 12.90 | 7.18 | 11.96 | 10.19 | 10.56 | 7.28 |
| | ATC-MC | 3.47 | 11.84 | 5.79 | **3.76** | 7.26 | 4.34 | 3.71 | 5.96 | 4.67 | 5.16 | 3.71 | 3.65 | 4.37 | 10.76 | 6.06 | 11.03 | 8.74 | 9.15 | 6.24 |
| | DoC | 3.85 | 12.91 | 5.51 | 4.61 | 7.82 | 5.08 | 3.67 | 6.60 | 5.09 | 5.08 | 3.90 | 4.96 | 4.88 | 11.87 | 6.29 | 11.21 | 9.24 | 9.65 | 6.76 |
| | Nul.norm | 3.87 | **3.95** | **2.18** | 9.76 | **5.26** | 3.44 | 11.45 | 7.91 | 6.68 | 2.77 | 2.53 | 4.40 | 5.37 | 5.46 | 6.41 | 12.22 | 6.71 | 7.70 | 5.97 |
| | FD | 2.48 | 4.18 | 4.57 | 4.64 | **4.46** | 3.54 | 6.30 | 5.71 | **4.10** | 3.13 | 2.61 | **1.22** | **3.55** | **3.04** | 4.51 | **7.25** | 4.78 | **4.89** | **4.09** |
| | BoP+JS | 3.92 | 6.92 | 6.43 | 6.95 | 6.77 | **2.22** | 9.66 | **4.66** | 4.51 | 3.34 | 3.10 | 2.15 | 4.02 | 5.03 | **3.52** | 7.64 | **4.30** | 5.12 | 4.88 |
| | MS-AoL | **3.32** | 9.94 | 11.70 | 5.73 | 9.15 | 2.55 | **2.77** | **3.91** | 4.46 | **2.10** | 2.72 | **1.11** | **2.66** | 7.04 | **4.22** | 7.74 | 5.92 | 6.23 | 4.94 |
| | Avg | 3.71 | 10.08 | 5.77 | 5.57 | 7.22 | 4.18 | 5.27 | 6.22 | 5.06 | 4.24 | 3.50 | 3.82 | 4.51 | 9.08 | 5.85 | 10.28 | 7.65 | 8.22 | 6.06 |
| Avg multiple classifiers (70) | Pred. s (0.7) | 3.99 | 8.98 | 7.15 | 6.62 | 7.62 | 3.54 | 4.98 | 5.29 | 4.14 | 5.15 | 4.83 | 4.28 | 4.58 | 9.46 | 8.28 | 10.97 | 9.37 | 9.52 | 6.52 |
| | Pred. s (0.8) | 3.73 | 9.02 | 7.06 | 6.24 | 7.48 | 3.33 | 4.61 | 5.13 | 3.98 | 4.94 | 4.42 | 3.65 | 4.25 | 9.05 | 7.76 | 10.39 | 8.85 | 9.01 | 6.18 |
| | Pred. s (0.9) | 3.51 | 8.98 | 6.95 | **6.00** | 7.36 | 3.18 | **4.30** | **5.00** | **3.77** | 4.75 | 4.08 | 3.03 | **3.93** | 8.48 | 7.34 | 9.68 | 8.23 | 8.43 | **5.84** |
| | Entropy | 3.83 | 9.08 | 7.04 | 6.49 | 7.58 | 3.30 | 4.54 | 5.17 | 4.04 | 5.00 | 4.50 | 3.50 | 4.23 | 9.05 | 7.70 | 10.48 | 9.03 | 9.06 | 6.21 |
| | ATC-MC | 3.62 | 8.92 | 7.08 | 6.11 | 7.41 | 3.22 | 4.48 | 5.05 | 3.82 | 4.88 | 4.29 | 3.30 | 4.08 | 8.72 | 7.49 | 9.91 | 8.51 | 8.66 | 5.99 |
| | DoC | 3.70 | 9.07 | 7.08 | 6.25 | 7.51 | 3.29 | 4.48 | 5.09 | 3.95 | 4.93 | 4.39 | 3.47 | 4.17 | 8.92 | 7.68 | 10.34 | 8.79 | 8.93 | 6.13 |
| | Nul.norm | 3.66 | **4.15** | **3.38** | 7.74 | **5.06** | **2.97** | 10.93 | 7.36 | 5.75 | **3.31** | **2.42** | 2.96 | 4.84 | 5.63 | 5.71 | **9.49** | 7.27 | 7.03 | 5.47 |
| | FD | 6.02 | 6.03 | **3.98** | 13.14 | 7.67 | 3.91 | 13.24 | 8.33 | 6.16 | 3.77 | 3.52 | 5.96 | 6.28 | 7.05 | 7.16 | 13.74 | 10.36 | 9.58 | 7.45 |
| | BoP | **3.47** | **5.71** | 4.79 | 12.94 | 7.76 | 3.43 | 10.98 | 6.84 | 6.08 | 4.86 | 3.74 | **2.93** | 5.27 | **4.89** | **5.26** | 10.05 | **7.16** | **6.84** | 6.20 |
| | MS-AoL | **3.36** | 8.23 | 8.70 | **5.25** | 7.42 | **2.25** | **2.92** | **4.25** | **3.58** | 2.99 | 2.51 | **1.17** | **2.66** | 5.60 | **4.54** | 7.13 | 5.59 | **5.72** | **4.45** |
| | Avg | 3.89 | 7.82 | 6.32 | 7.68 | 7.29 | 3.24 | 6.55 | 5.75 | 4.53 | 4.46 | 3.87 | 3.43 | 4.43 | 7.69 | 6.89 | 10.22 | 8.32 | 8.28 | 6.04 |

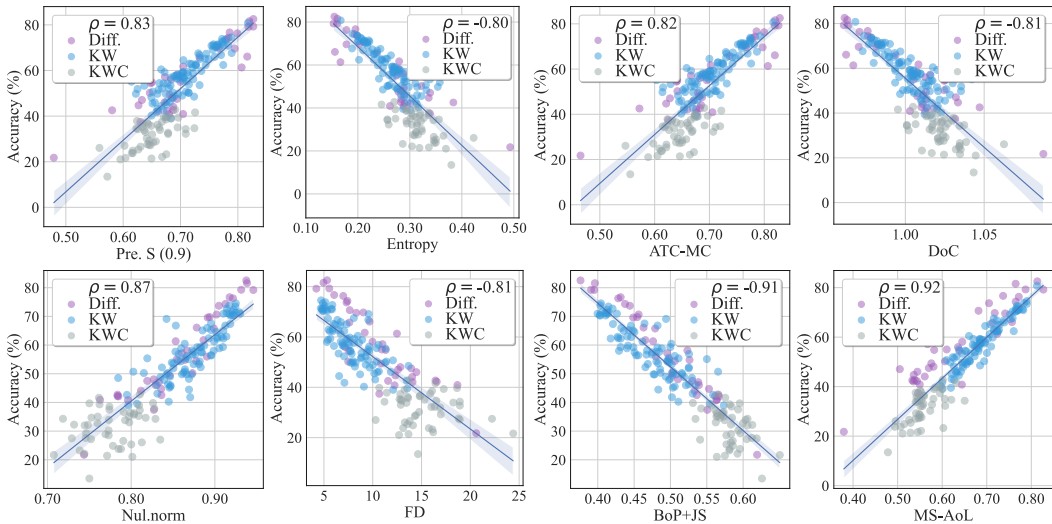

Figure 9: **Correlation studies of different AccP methods on ResNet44.** Visualizing correlation between accuracy and prediction scores on CIFAR-10-W. We use the Spearman's rank correlation $\rho$.

when using larger-sized images, such as $224 \times 224$. This is because large images can provide more visual details and clearer boundaries between classes, making the classification task relatively easier. This ease of classification can lead to higher correlation coefficients between the proxy output and classifier accuracy. However, it is important to note that this simplified scenario cannot simulate the complexities and challenges of real-world accuracy prediction tasks. By utilizing the smaller image size, the AccP task becomes more challenging as it requires models to predict the accuracy of classifiers based on limited visual information and potentially more ambiguous class boundaries. To increase the challenge, our paper uses smaller image sizes of $32 \times 32$ in the AccP task.

**2)** *Prediction-score based AccP methods tend to perform better than feature-based methods on CIFAR-10-W with an image size of $224 \times 224$.* One possible reason for this observation is related to the characteristics of the classifier and the complexity of its features. When using larger-sized images,

such as $224 \times 224$, the classifier may have higher confidence in its predictions due to the increased visual information and clearer boundaries between classes. This higher confidence can be reflected in the prediction scores, which are used by prediction-score based AccP methods. These methods directly utilize the confidence scores of the classifier to predict its accuracy, and the higher confidence on larger images can lead to more accurate predictions. On the other hand, feature-based methods rely on extracting and analyzing the underlying features learned by the classifier. With larger-sized images, the complexity of the features may increase, making it more challenging for feature-based methods to accurately estimate the classifier's accuracy.

**3)** *The challenge of evaluating different classifiers varies within the same group of sets.* For instance, in Fig. 11 (C) Vgg13, the $\rho$ values of many methods show a noticeable drop on the 360 and Bing sets compared to the results on other classifiers, such as Fig. 11 (A) Resnet50, Fig. 11 and (B) Densenet121. This observation could be related to the architecture, training procedure, or inherent characteristics of the Vgg13 model. Another factor that can contribute to the variation in performance is the nature of the 360 and Bing sets themselves. These sets may contain specific challenges or characteristics that make them more difficult for the Vgg13 classifier to accurately predict their accuracy. We will further study this in future work.

**Acc vs MAE.** The relationship between accuracy and accuracy prediction error (Mean Absolute Error, MAE) is depicted in Fig. 12. The left side of the figure represents the CIFAR-10-Cs testbed, while the right side represents the CIFAR-10-W testbed. It can be observed that the MAE values of the CIFAR-10-W sets exhibit a larger range compared to the CIFAR-10-Cs sets. This indicates that the CIFAR-10-W testbed is more challenging in terms of accuracy prediction. The broader range of MAE values suggests that the AccP methods face increased difficulty in AccP the accuracies of models on the CIFAR-10-W sets compared to the CIFAR-10-Cs sets.

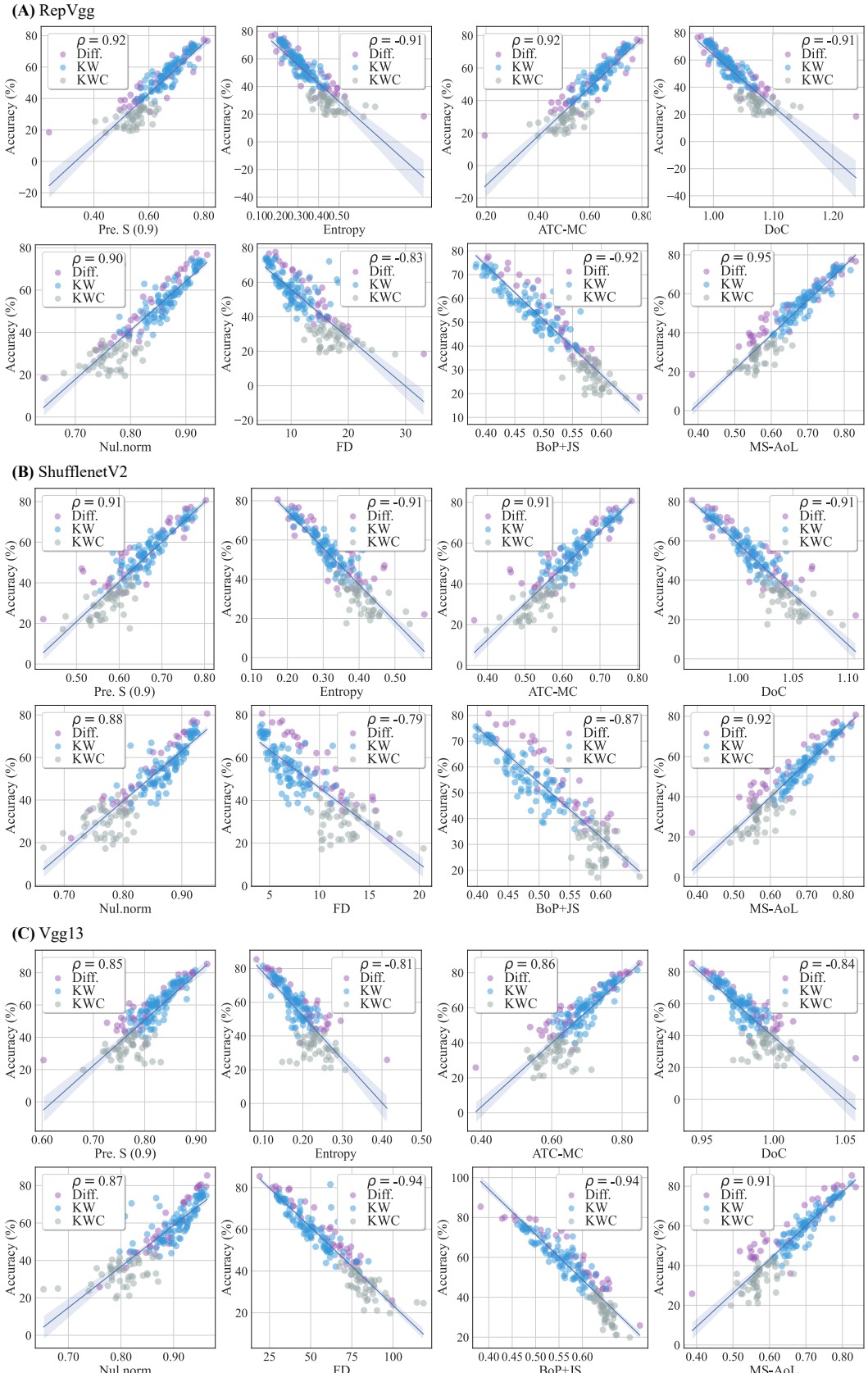

Figure 10: **Correlation studies of different methods on 3 different classifiers.** Visualizing correlation between accuracy and prediction scores on CIFAR-10-W. We use RepVgg, ShufflenetV2, Vgg13 classifiers and Spearman's rank correlation $\rho$.

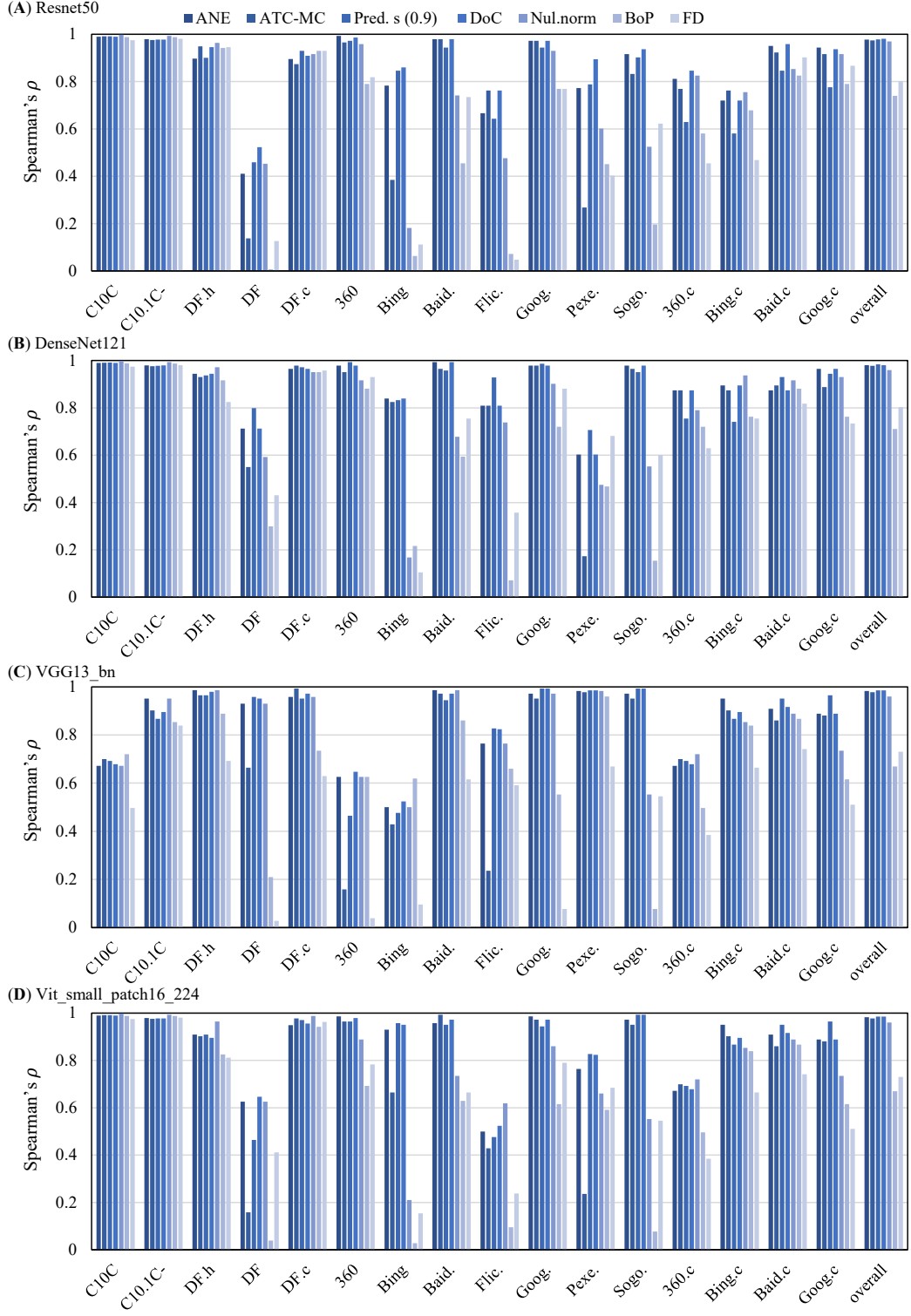

Figure 11: **Correlation studies of different AccP methods.** Visualizing correlation between accuracy and prediction scores on CIFAR-10-W. We use ResNet44 and Spearman's rank correlation $\rho$.

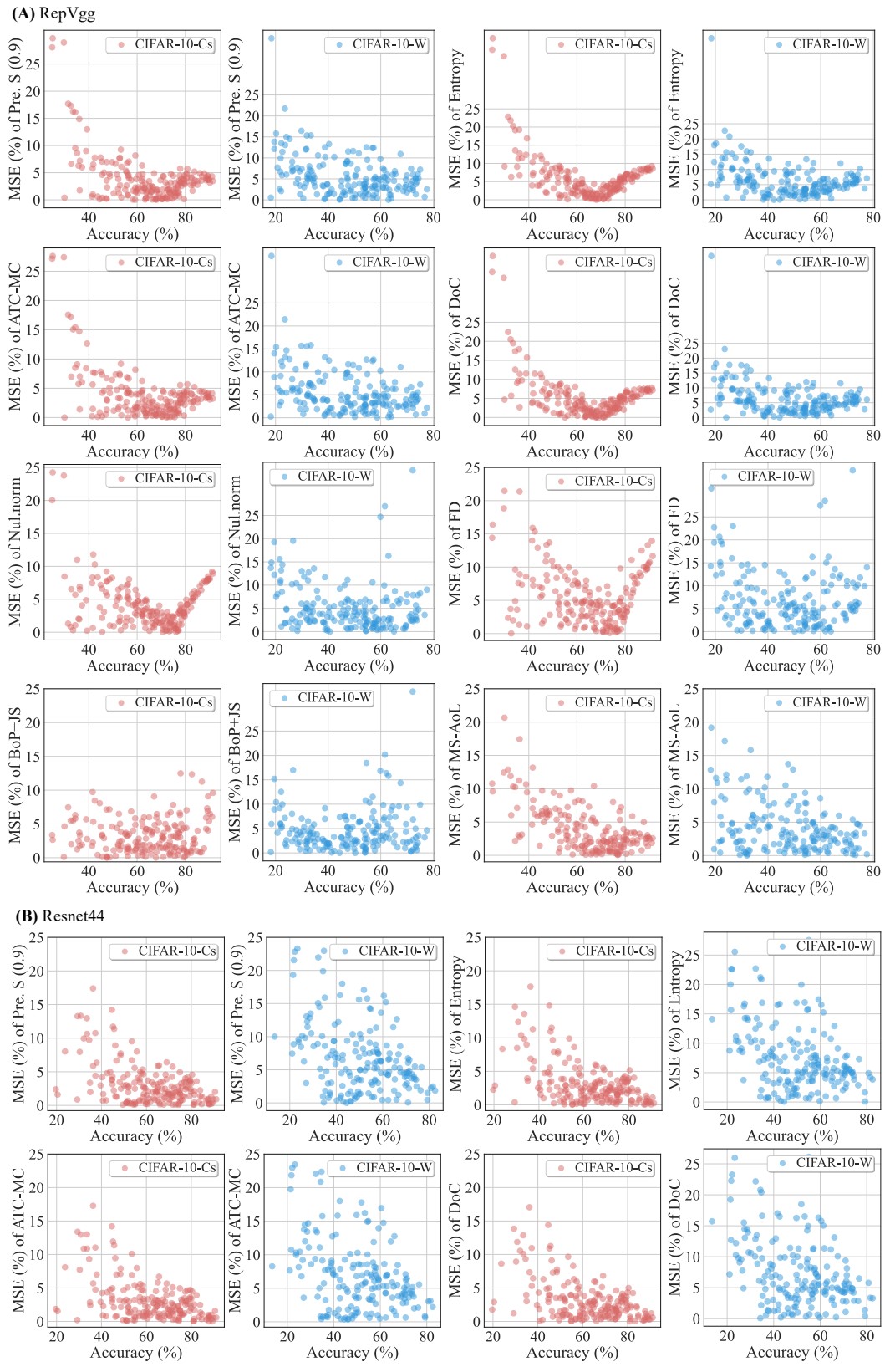

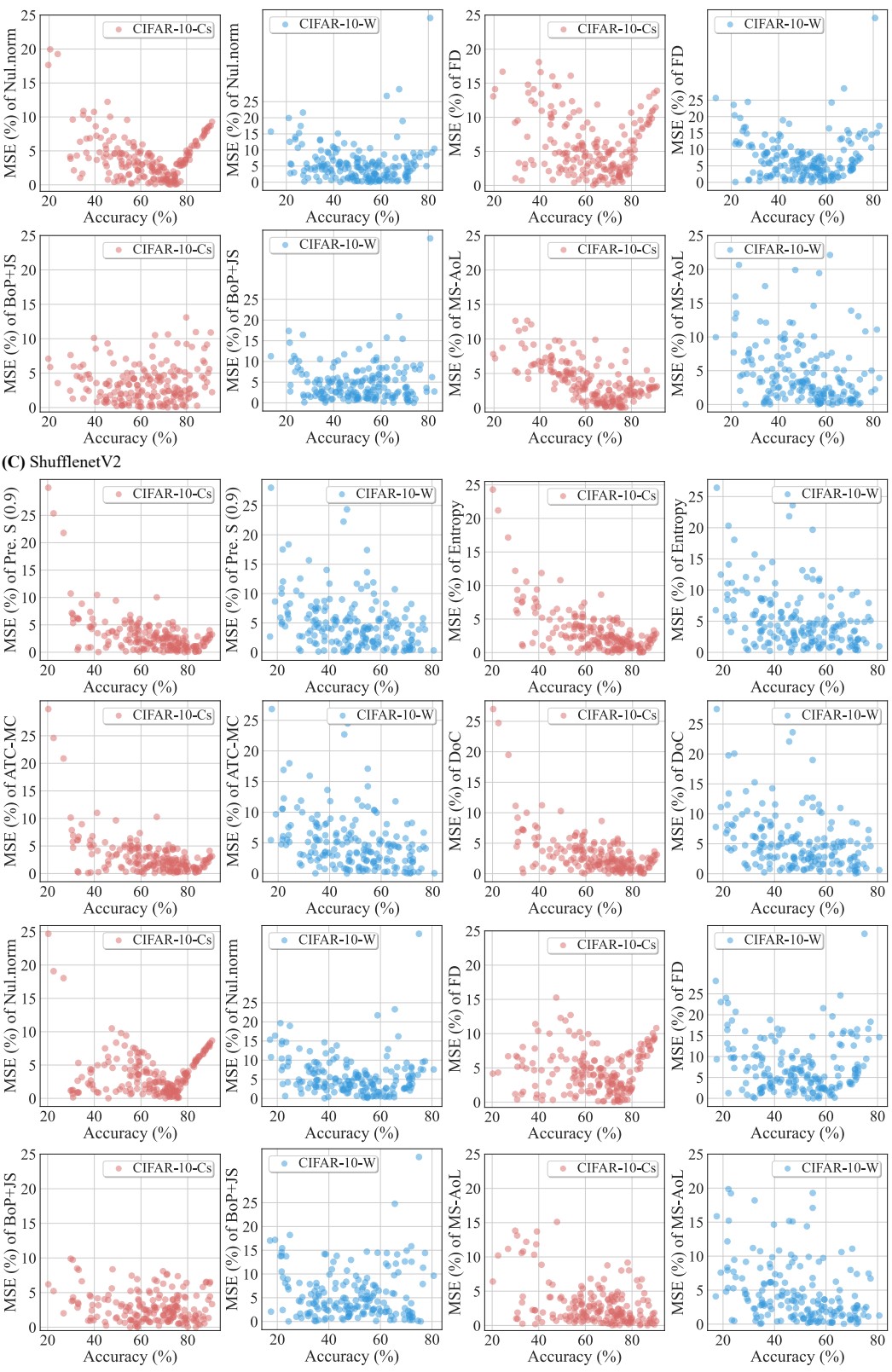

**(C)** ShufflenetV2

**(D)** Vgg13

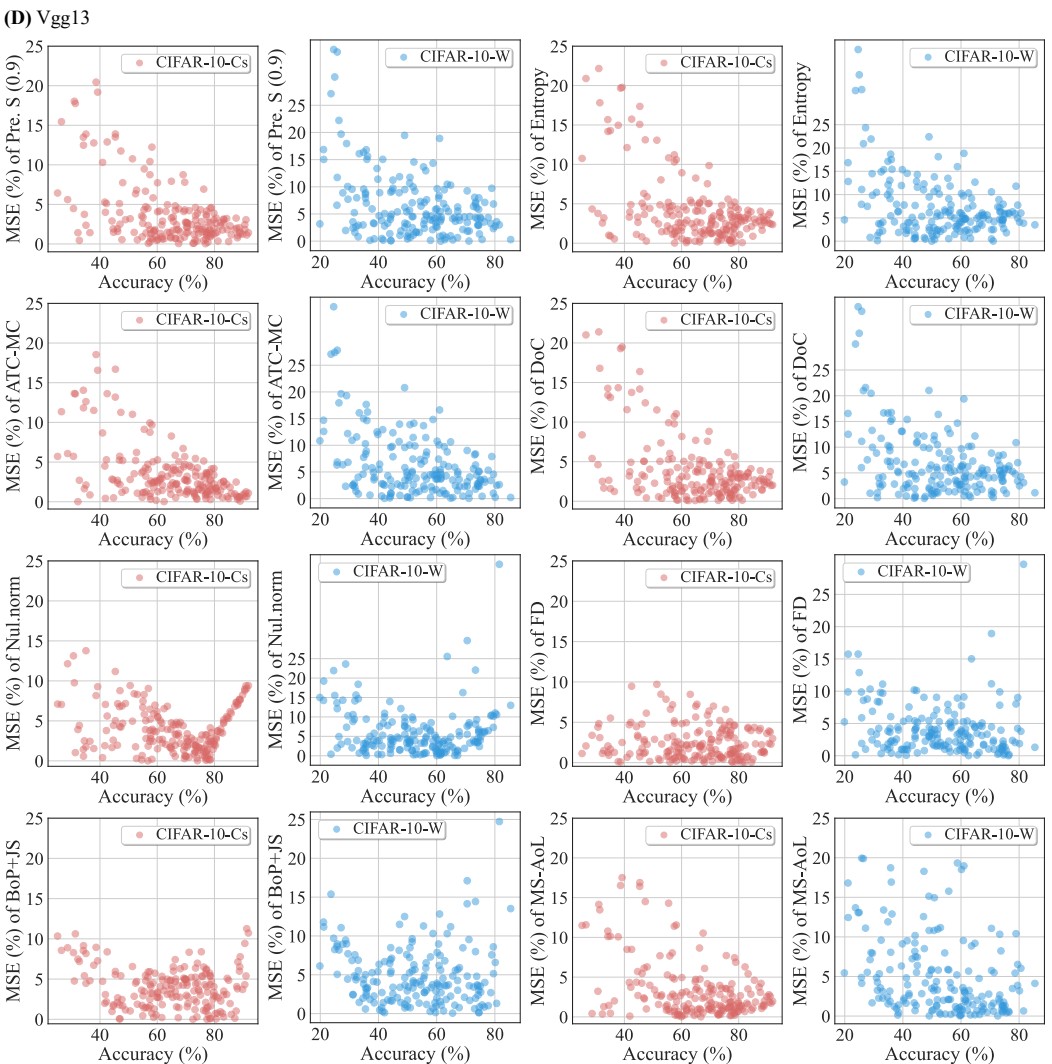

Figure 12: **Acc vs MAE** Relationship between accuracy and accuracy prediction error (MAE, %) on the CIFAR-10-Cs (left) and CIFAR-10-W (right) testbeds. We evaluate 8 methods on 4 classifiers (**A.** RepVgg, **B.** ResNet44, **C.** Shufflenetv2, and **D.** Vgg13)
.

## C.2 TASK II: DOMAIN GENERALIZATION

**Single-Source DG.** We report the results of Single-Source DG on the four training sets we newly collected respectively in Table 7. We can observe that: 1) Training DG models on datasets that include cartoon images, such as Diff-CC and Yandex-CC, generally leads to higher average accuracies on all sets of CIFAR-10-W compared to training on datasets without the inclusion of 'cartoon' in keywords or prompts, such as Diff-C and Yandex-C. Specifically, DG models trained on cartoon images perform better on CIFAR-10-W KWC and CIFAR-10-W diffusion, while their performance on CIFAR-10-KW is not significantly worse compared to models trained on Diff-C and Yandex-C datasets. One possible explanation is that models trained on cartoon images exhibit acceptable generalization capabilities on real data compared to models trained on "natural" images when it comes to handling cartoon images.

Table 7: **Benchmarking five domain generalization methods on CIFAR-10-W**. We report Top-1 classification accuracy (%). For each method under each setup, mean and standard deviation computed over three runs are reported in the last column.

| Setup | Method | $C_w^{10}$ - diffusion (37) | | | $C_w^{10}$ - KW (95) | | | | | | | $C_w^{10}$ - KWC (48) | | | | $C_w^{10}$ |
|---|---|---|---|---|---|---|---|---|---|---|---|---|---|---|---|---|
| | | Dif.h | Dif.c | Dif. | Goo | Bin | Bai | 360 | Sog | Fli | Pex | Goo | Bin | Bai | 360 | **All** |
| Diff.C | ERM | 79.79 | 77.00 | 95.39 | 77.89 | 85.47 | 69.66 | 72.14 | 82.78 | 86.37 | 90.85 | 48.72 | 48.01 | 39.78 | 41.39 | 72.27 ± 2.88 |
| | SagNet | 79.71 | 77.01 | 93.11 | 76.67 | 83.14 | 68.00 | 72.51 | 81.79 | 85.64 | 90.57 | 48.48 | 46.37 | 38.57 | 40.47 | 71.36 ± 3.20 |
| | SelfReg | 79.75 | 78.34 | 94.54 | 77.96 | 84.91 | 69.21 | 72.76 | 82.65 | 86.16 | 90.99 | 49.21 | 48.28 | 39.79 | 41.85 | 72.35 ± 3.69 |
| | SD | 80.94 | 77.49 | 93.50 | 78.33 | 84.73 | 69.78 | 74.43 | 83.86 | 87.53 | 91.56 | 49.21 | 47.55 | 40.17 | 41.71 | 72.70 ± 4.28 |
| | RSC | 78.15 | 75.38 | 92.05 | 71.98 | 78.73 | 65.30 | 68.41 | 78.81 | 81.92 | 87.20 | 46.22 | 43.48 | 38.18 | 38.51 | 68.63 ± 5.67 |
| Diff.CC | ERM | 88.91 | 93.21 | 88.93 | 77.46 | 84.48 | 69.62 | 72.96 | 81.75 | 85.88 | 89.65 | 63.68 | 59.36 | 50.65 | 55.79 | 76.43 ± 1.6 |
| | SagNet | 91.27 | 95.48 | 91.01 | 79.89 | 86.93 | 71.63 | 74.43 | 83.72 | 86.76 | 90.50 | 66.89 | 61.30 | 52.55 | 56.58 | 78.23 ± 1.55 |
| | SelfReg | 91.10 | 95.50 | 91.05 | 81.98 | 88.68 | 73.99 | 76.24 | 85.62 | 88.69 | 92.37 | 68.41 | 62.57 | 53.64 | 58.36 | 79.65 ± 1.16 |
| | SD | 89.39 | 93.65 | 89.80 | 79.78 | 86.50 | 71.67 | 75.24 | 83.68 | 87.85 | 91.18 | 65.95 | 61.01 | 52.55 | 57.11 | 78.12 ± 1.43 |
| | RSC | 88.10 | 94.06 | 88.05 | 73.96 | 81.41 | 65.92 | 67.94 | 78.08 | 81.44 | 85.98 | 58.93 | 54.39 | 47.61 | 51.03 | 73.02 ± 2.23 |
| Yandex-C | ERM | 72.64 | 97.72 | 83.16 | 86.79 | 92.95 | 76.97 | 83.68 | 90.11 | 89.90 | 93.95 | 52.68 | 49.83 | 41.76 | 39.95 | 75.98 ± 0.89 |
| | SagNet | 78.81 | 97.64 | 85.59 | 87.78 | 93.63 | 79.10 | 84.95 | 91.92 | 91.50 | 94.78 | 57.51 | 53.24 | 45.55 | 45.60 | 78.34 ± 0.99 |
| | SelfReg | 73.57 | 97.75 | 83.12 | 87.17 | 92.91 | 77.52 | 83.30 | 89.99 | 89.94 | 94.03 | 54.21 | 50.72 | 42.87 | 41.66 | 76.36 ± 1.81 |
| | SD | 75.66 | 97.52 | 84.01 | 87.97 | 93.41 | 78.50 | 84.15 | 91.16 | 91.59 | 95.27 | 55.21 | 51.24 | 43.52 | 41.49 | 77.23 ± 1.22 |
| | RSC | 68.43 | 96.33 | 79.95 | 83.84 | 91.47 | 74.98 | 81.67 | 88.55 | 88.32 | 92.77 | 49.38 | 47.81 | 41.47 | 39.39 | 73.98 ± 1.08 |
| Yandex-CC | ERM | 92.19 | 94.92 | 87.94 | 78.05 | 83.89 | 72.55 | 76.33 | 79.32 | 78.66 | 83.72 | 79.66 | 69.88 | 63.15 | 69.67 | 78.66 ± 1.66 |
| | SagNet | 93.81 | 96.65 | 90.06 | 79.34 | 85.58 | 75.01 | 77.35 | 82.13 | 81.38 | 86.60 | 82.34 | 72.34 | 65.67 | 72.06 | 80.98 ± 0.95 |
| | SelfReg | 94.77 | 97.22 | 90.73 | 82.14 | 87.62 | 77.36 | 79.55 | 85.59 | 84.95 | 89.79 | 83.51 | 74.14 | 67.47 | 73.97 | 83.07 ± 0.63 |
| | SD | 94.71 | 97.08 | 91.04 | 83.28 | 88.88 | 78.56 | 82.07 | 86.58 | 86.17 | 90.49 | 84.24 | 75.09 | 67.71 | 74.54 | 83.96 ± 0.76 |
| | RSC | 90.09 | 94.88 | 87.20 | 74.91 | 81.16 | 70.06 | 72.03 | 76.32 | 76.33 | 81.37 | 79.05 | 69.70 | 61.76 | 67.77 | 76.79 ± 0.85 |

**Evaluation on Balanced Subset for DG.** An almost balanced subset is built by sampling 100 images from each class in the 180 domains (if there are not 100 images, all the data of this class will be used). The results of different Domain Generalization (DG) methods in both Single-source and Multi-source settings on the balanced subset are presented in Table 8. When comparing these results with those on the original CIFAR-10-W (Table 3), we have several key observations: 1) SagNet performs well on balanced subsets under the Single DG scenario, while SD excels on imbalanced sets. 2) The single DG regime gets reduced performance variance on the balanced subset. 3) In multi-source DG, results on the balanced subset and the original set are similar. Hence, these observations suggest that single DA methods are more responsive to dataset balance changes compared to multi-source DG. We recognize the complexity of this relationship and plan to investigate it further in future work.

**Impact of different single source domains on domain generalization** is depicted in Fig. 13, which presents the results across four methods. The density plot on the y-axis showcases the density of the test set at various levels of improvement relative to the baseline method, ERM. On the x-axis, the density plot displays the distribution of accuracies achieved by ERM on CIFAR-10-W datasets. Several observations can be made: 1) When using Yandex-CC, Diff-CC, or Yandex-C as the training source, SD and SagNet exhibit improved testing results on the majority of CIFAR-10-W datasets. This suggests that these training sources contribute to better domain generalization performance. Diff-C data may be too simple to train a high-performing domain generalization model, resulting in less impressive results. 2) RSC, on the other hand, performs worse than other methods across all four training sources, indicating its limited effectiveness in domain generalization tasks.

**Impacts of increasing the number of source domains or images on domain generalization** are illustrated in Fig. 14. The results of four DG methods trained on two (A), three (B), and four (C)

Table 8: **Benchmarking domain generalization methods on CIFAR-10-W balanced subset**. We report Top-1 classification accuracy (%) . The mean and standard deviation computed over three runs are reported in the last column. All other notations are the same as in Table 2.

| Setup | Method | $C_w^{10}$- diffusion (37) | | | $C_w^{10}$- KW (95) | | | | | | | $C_w^{10}$- KWC (48) | | | | $C_w^{10}$ |
|---|---|---|---|---|---|---|---|---|---|---|---|---|---|---|---|---|
| | | DF.h | DF.c | DF | Goo | Bin | Bai | 360 | Sog | Fli | Pex | Goo | Bin | Bai | 360 | **All** |
| Single (1) DG | ERM | 81.12 ± 1.11 | 75.95 ± 1.36 | 95.47 ± 1.30 | 77.27 ± 1.68 | 85.13 ± 2.84 | 68.05 ± 1.35 | 73.89 ± 1.93 | 82.81 ± 1.39 | 85.54 ± 2.09 | 89.20 ± 2.18 | 47.99 ± 1.16 | 44.48 ± 1.59 | 39.66 ± 1.21 | 38.28 ± 0.93 | **71.50** ± 1.61 |
| | SagNet | 80.94 ± 1.35 | 78.25 ± 3.12 | 95.12 ± 0.67 | 78.24 ± 0.64 | 86.18 ± 0.71 | 68.49 ± 0.93 | 74.97 ± 0.76 | 83.56 ± 0.14 | 85.79 ± 0.25 | 90.18 ± 0.20 | 49.78 ± 3.22 | 46.64 ± 2.22 | 41.79 ± 1.65 | 40.93 ± 2.43 | **72.61** ± 1.24 |
| | SelfReg | 79.66 ± 0.52 | 72.09 ± 2.15 | 96.16 ± 1.11 | 77.59 ± 1.27 | 87.10 ± 1.53 | 68.18 ± 1.60 | 73.96 ± 2.10 | 83.47 ± 1.84 | 86.49 ± 1.34 | 90.90 ± 1.16 | 46.19 ± 1.25 | 43.93 ± 0.94 | 39.43 ± 1.05 | 37.87 ± 0.88 | 71.46 ± 1.33 |
| | SD | 79.45 ± 0.48 | 73.49 ± 1.33 | 94.96 ± 1.05 | 77.96 ± 2.06 | 85.46 ± 2.72 | 68.12 ± 1.52 | 74.05 ± 1.94 | 83.02 ± 1.38 | 85.82 ± 2.13 | 89.87 ± 1.87 | 46.19 ± 1.39 | 43.64 ± 1.43 | 39.69 ± 0.47 | 37.95 ± 1.15 | 71.18 ± 1.52 |
| | RSC | 78.17 ± 1.02 | 67.49 ± 1.95 | 94.52 ± 0.61 | 72.73 ± 1.61 | 82.21 ± 1.14 | 64.65 ± 1.16 | 69.39 ± 1.53 | 78.31 ± 0.94 | 80.92 ± 0.08 | 86.04 ± 0.79 | 43.14 ± 1.98 | 39.87 ± 1.04 | 37.39 ± 1.14 | 34.80 ± 1.87 | 67.58 ± 1.17 |
| Multi(4)-Source DG | ERM | 87.63 ± 0.73 | 95.41 ± 0.10 | 98.68 ± 0.18 | 88.20 ± 0.92 | 93.92 ± 0.44 | 81.47 ± 0.81 | 87.38 ± 0.49 | 92.15 ± 0.96 | 93.63 ± 0.53 | 96.40 ± 0.54 | 84.40 ± 0.11 | 77.28 ± 0.77 | 67.77 ± 0.26 | 73.86 ± 0.44 | 87.54 ± 0.52 |
| | SagNet | 88.45 ± 0.53 | 96.10 ± 0.70 | 98.84 ± 0.26 | 88.94 ± 0.14 | 94.63 ± 0.03 | 82.06 ± 0.38 | 88.03 ± 0.20 | 93.13 ± 0.39 | 93.84 ± 0.35 | 96.66 ± 0.20 | 85.49 ± 0.19 | 77.64 ± 0.27 | 68.79 ± 0.47 | 74.99 ± 0.53 | 88.19 ± 0.33 |
| | SelfReg | 87.78 ± 0.57 | 96.50 ± 0.52 | 98.92 ± 0.34 | 89.70 ± 0.13 | 95.00 ± 0.25 | 82.75 ± 0.51 | 88.70 ± 0.50 | 92.90 ± 0.74 | 93.98 ± 0.73 | 96.71 ± 0.53 | 85.89 ± 0.47 | 78.30 ± 0.61 | 70.04 ± 0.37 | 75.87 ± 0.72 | **88.56** ± 0.50 |
| | SD | 88.51 ± 0.31 | 96.13 ± 0.45 | 98.78 ± 0.16 | 89.81 ± 0.22 | 95.22 ± 0.40 | 83.31 ± 0.41 | 88.83 ± 0.56 | 93.57 ± 0.69 | 94.47 ± 0.43 | 97.00 ± 0.35 | 86.27 ± 0.10 | 78.86 ± 0.19 | 69.97 ± 0.49 | 75.94 ± 0.85 | **88.82** ± 0.40 |
| | Fishr | 87.89 ± 0.72 | 95.82 ± 0.15 | 98.94 ± 0.12 | 88.75 ± 0.92 | 94.43 ± 0.61 | 81.90 ± 0.69 | 87.68 ± 0.53 | 92.48 ± 0.52 | 93.72 ± 0.63 | 96.55 ± 0.51 | 84.37 ± 0.45 | 76.66 ± 0.36 | 67.54 ± 0.72 | 73.58 ± 0.35 | 87.70 ± 0.52 |
| | EQRM | 87.78 ± 1.17 | 95.49 ± 0.33 | 98.84 ± 0.09 | 88.54 ± 0.36 | 94.15 ± 0.21 | 81.77 ± 0.64 | 87.76 ± 0.47 | 92.15 ± 0.25 | 93.45 ± 0.62 | 96.39 ± 0.30 | 84.67 ± 0.77 | 77.29 ± 0.94 | 67.42 ± 0.64 | 73.49 ± 0.57 | 87.61 ± 0.52 |
| | RSC | 86.50 ± 1.10 | 94.58 ± 0.33 | 98.51 ± 0.28 | 86.09 ± 0.59 | 92.65 ± 0.44 | 79.11 ± 1.14 | 84.69 ± 1.38 | 90.33 ± 0.90 | 91.75 ± 0.78 | 94.99 ± 0.51 | 81.76 ± 1.62 | 75.02 ± 1.36 | 65.31 ± 1.98 | 70.97 ± 2.35 | 85.72 ± 1.03 |
| | CORAL | 88.25 ± 0.51 | 95.93 ± 0.28 | 98.74 ± 0.12 | 89.26 ± 0.15 | 94.97 ± 0.26 | 82.67 ± 0.38 | 88.53 ± 0.29 | 93.32 ± 0.21 | 94.28 ± 0.42 | 97.05 ± 0.37 | 85.75 ± 0.26 | 77.85 ± 0.28 | 69.77 ± 0.72 | 75.00 ± 0.49 | 88.47 ± 0.34 |

sources are compared, along with a single source (D) that has a similar number of images to the four sources. Notably, the focus of this comparison is on the relative benefits of multiple sources versus a larger number of images.

Upon comparing panels (A), (B), and (C) with panel (D), it can indeed be observed that having multiple sources may not necessarily provide more benefits than having a larger number of images for domain generalization. This observation suggests that the quantity and diversity of images within a single source can potentially have a greater impact on the performance of DG methods than the number of sources used for training.

However, it is important to note that these findings are based on rough experiments, and further investigation is required to validate and refine these observations. We will conduct more extensive and rigorous experiments in our future work to gain a deeper understanding of the impact of the number of sources and the number of images on domain generalization performance.

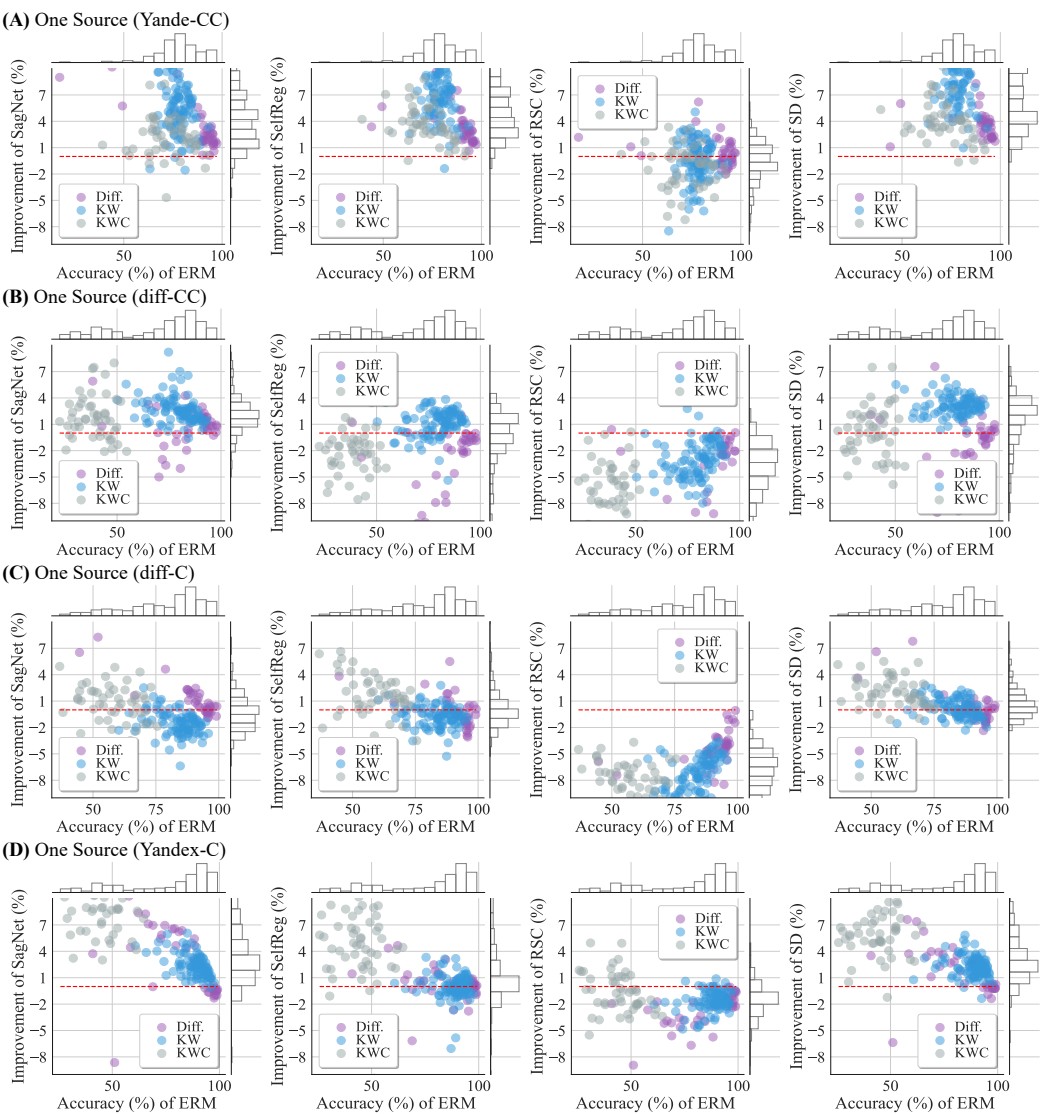

Figure 13: Impact of different single source domains on domain generalization.

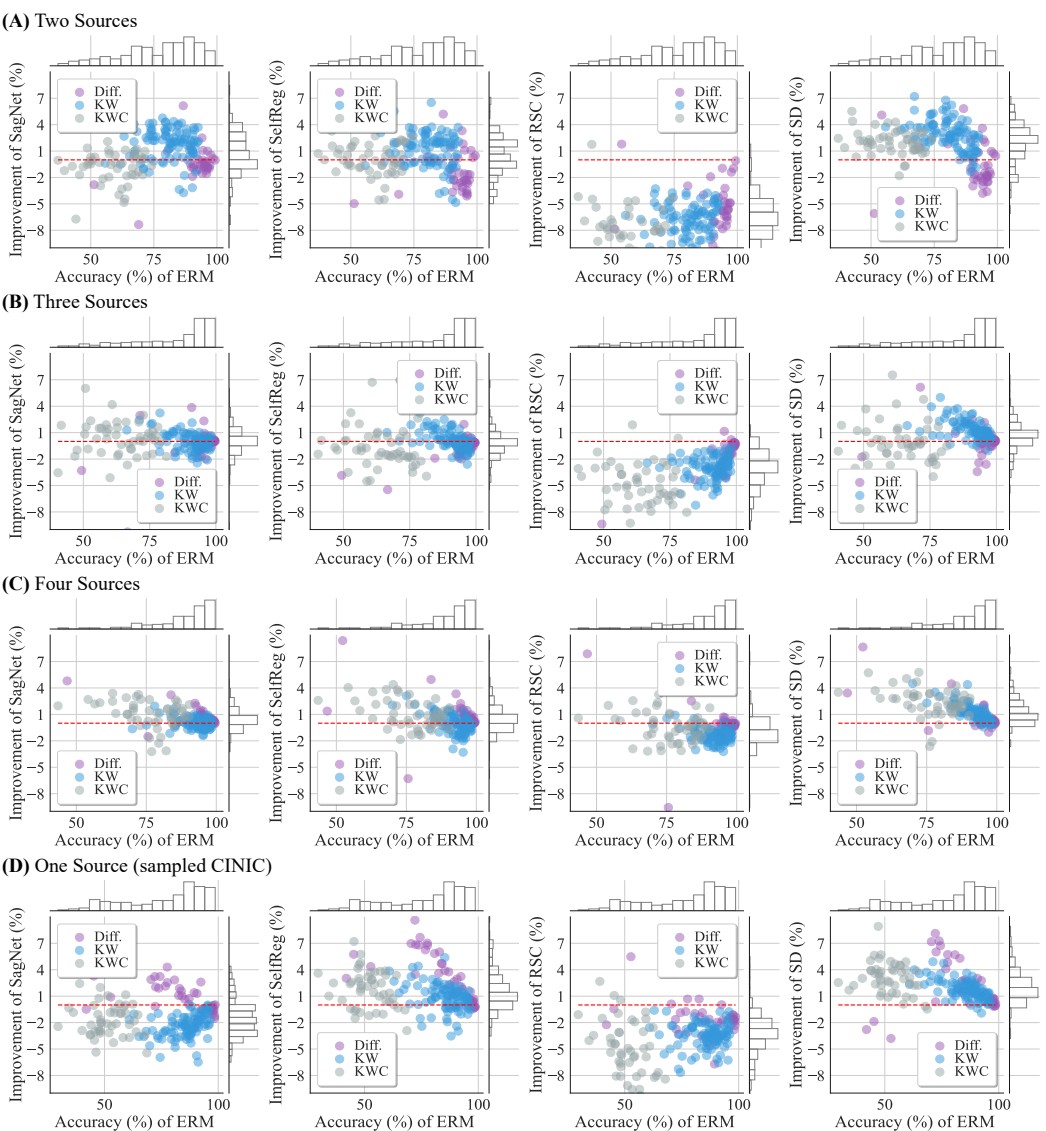

Figure 14: Impact of increasing the number of source domains and the number of images on domain generalization. DG methods trained on two (A), Three (B) and four(C) sources, as well as one source (D) that has a similar number of images to the four sources.

