# CIFAR-10-WAREHOUSE: BROAD AND MORE REALISTIC TESTBEDS IN MODEL GENERALIZATION ANALYSIS

**Xiaoxiao Sun**[1]*, **Xingjian Leng**[1]*, **Zijian Wang**[2]*, **Yang Yang**[1]*, **Zi Huang**[2], **Liang Zheng**[1]

[1] The Australian National University    [2] The University of Queensland

{first-name.last-name}@anu.edu.au[1]

zijian.wang@uq.edu.au, huang@itee.uq.edu.au

In the supplementary material, we show some example images of different sources in CIFAR-10-W.

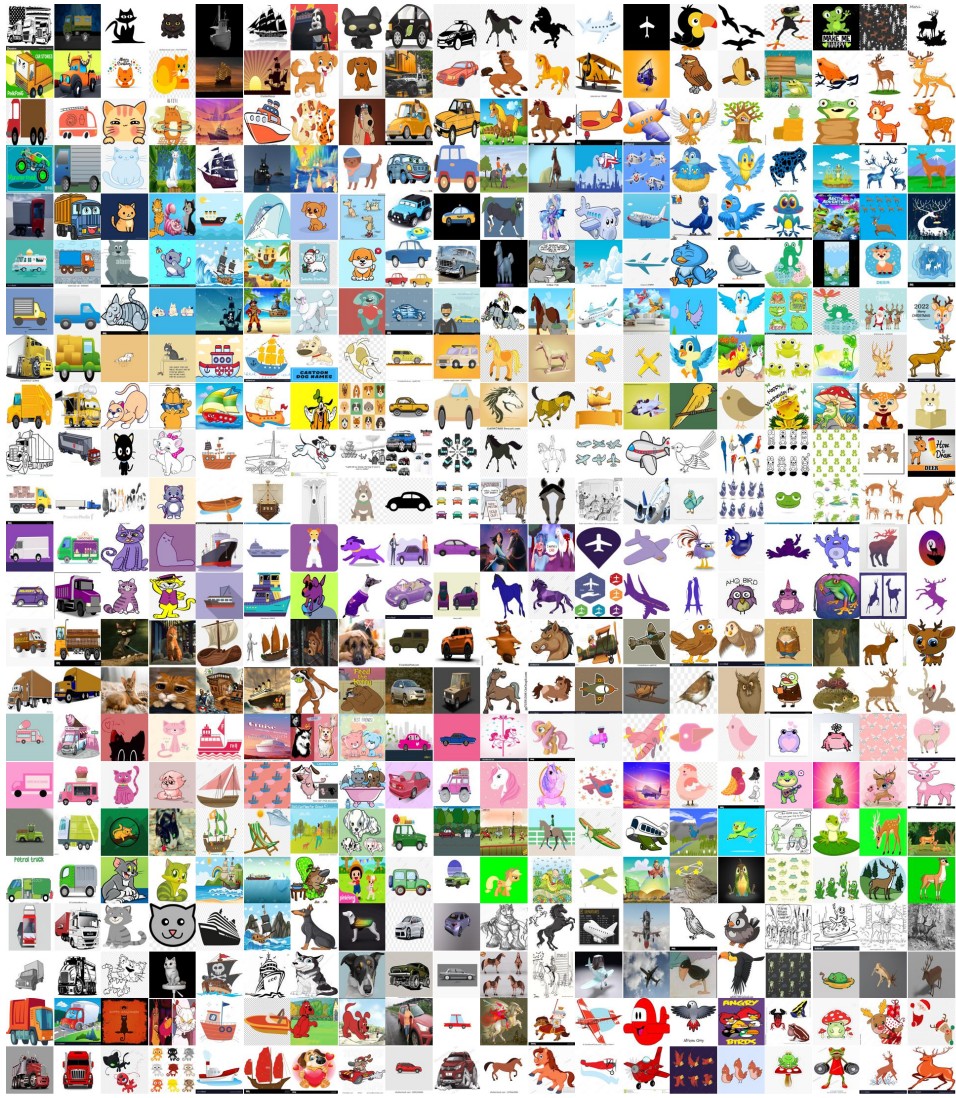

Figure 1: Cartoon images searched from Google with class names as keywords and 12 color options.

---

*Equal contribution.

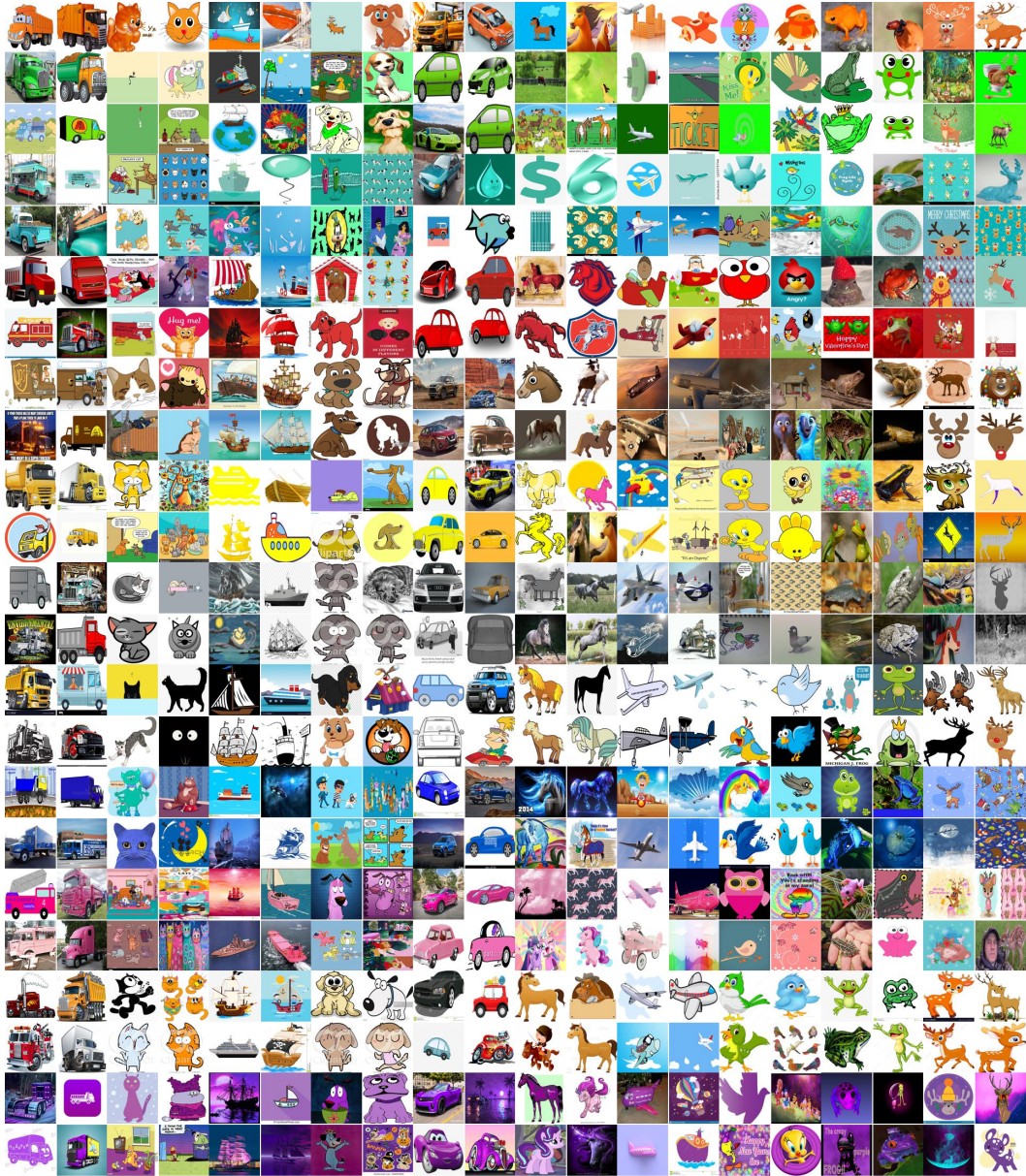

Figure 2: Cartoon images retrieved from Bing with class names as keywords and 12 color options.

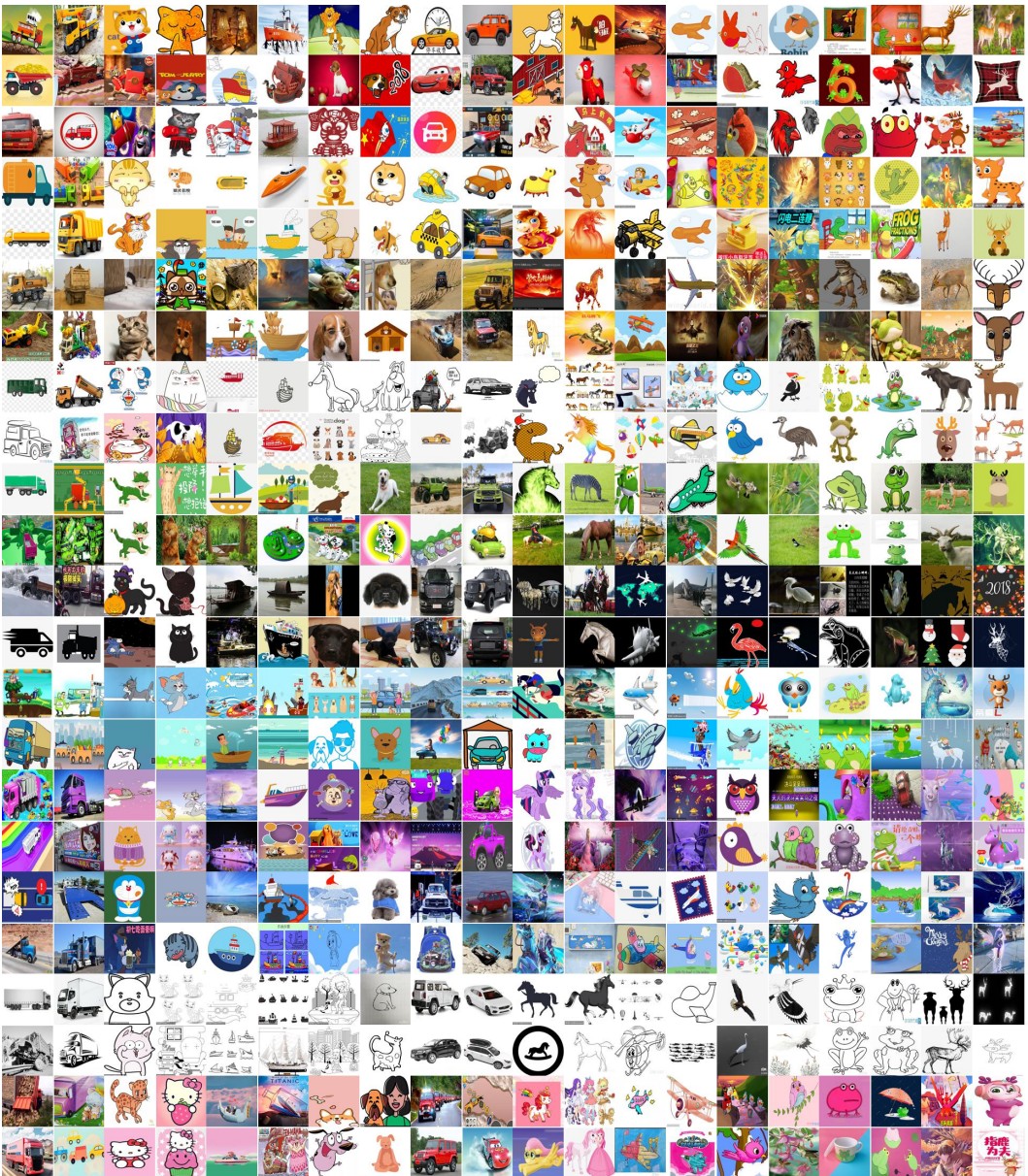

Figure 3: Cartoon images searched from Baidu with class names as keywords and 12 color options.

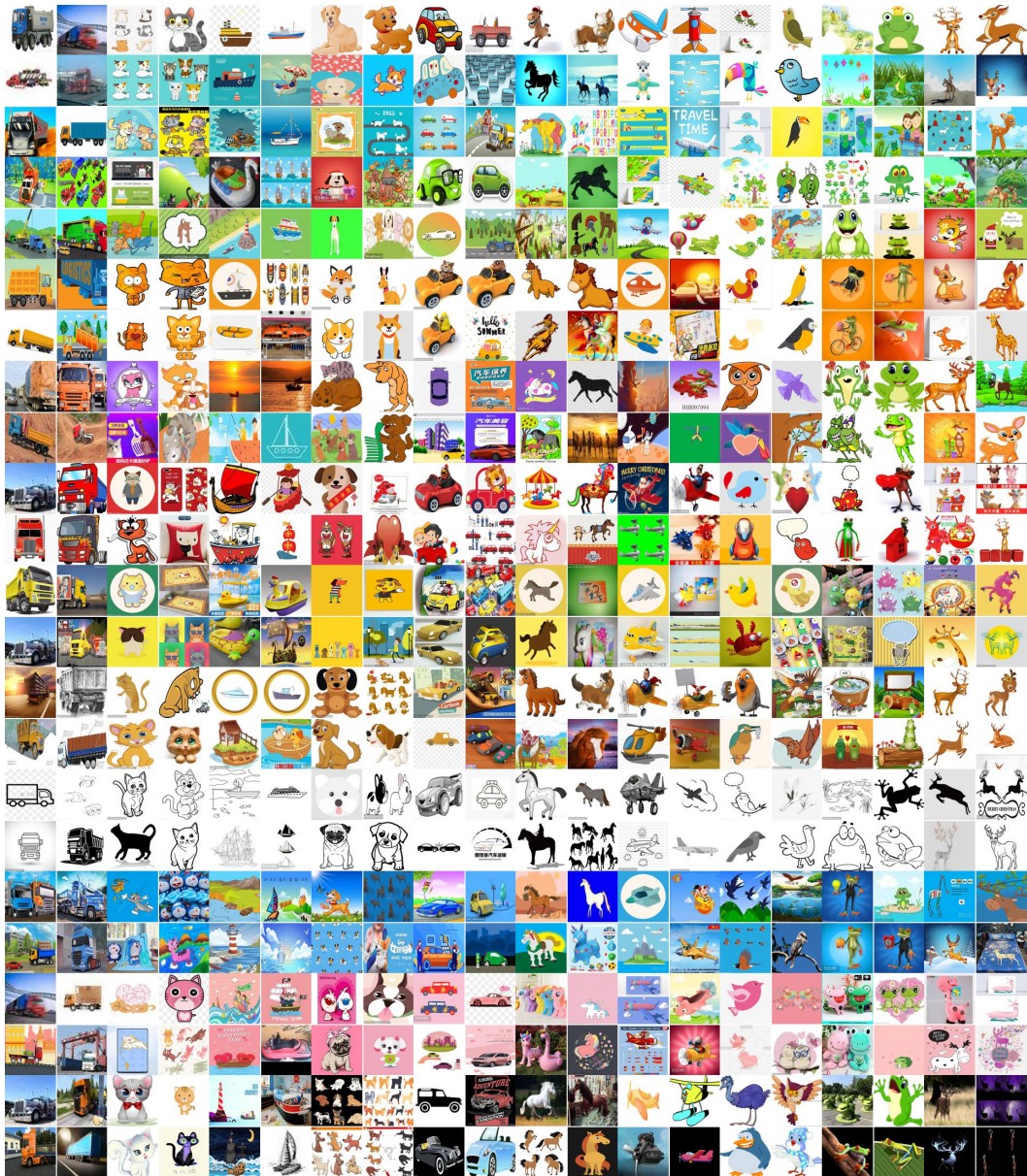

Figure 4: Cartoon images searched from 360 with class names as keywords and 12 color options.

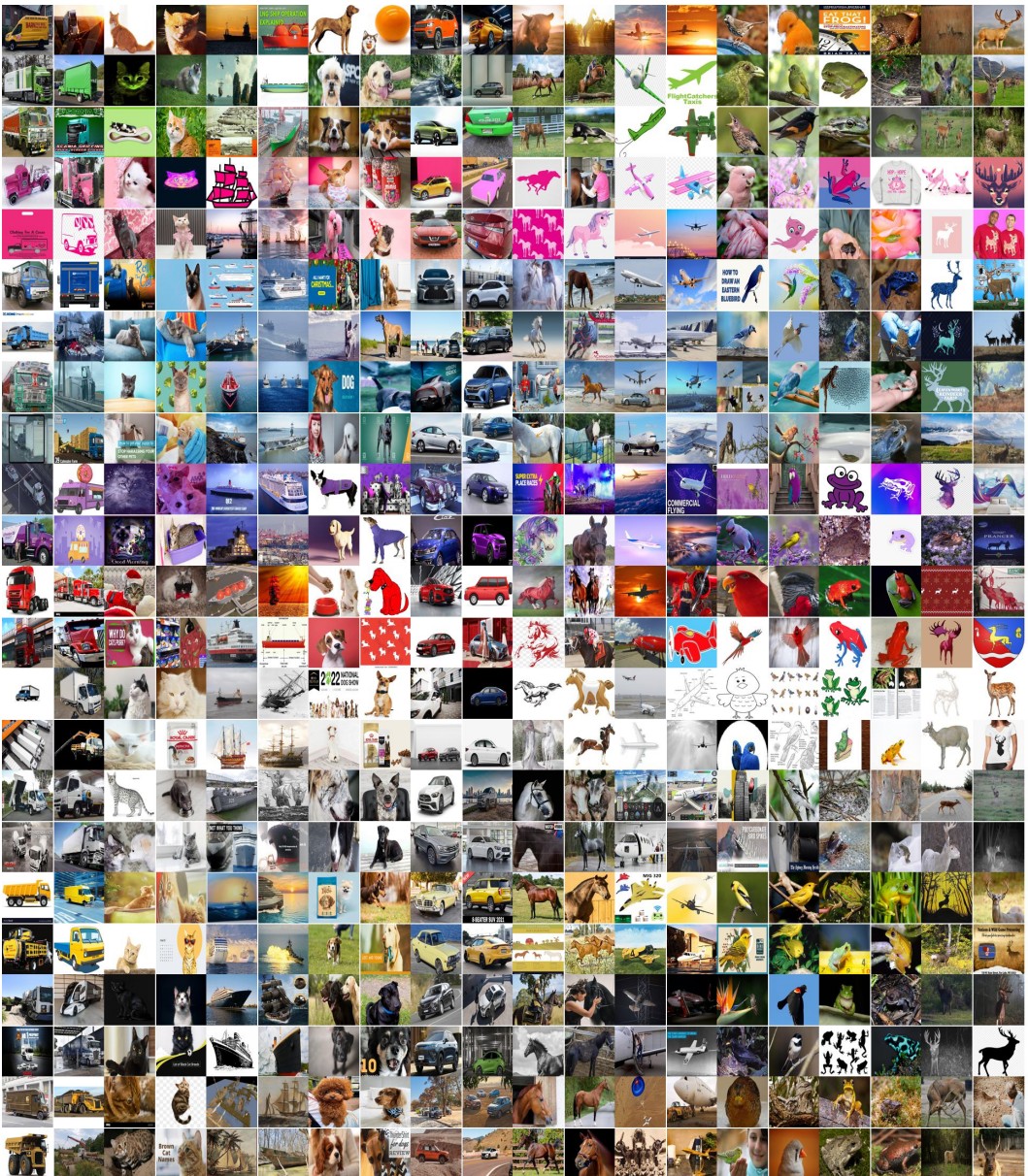

Figure 5: Images searched from Google with class names as keywords and 12 color options.

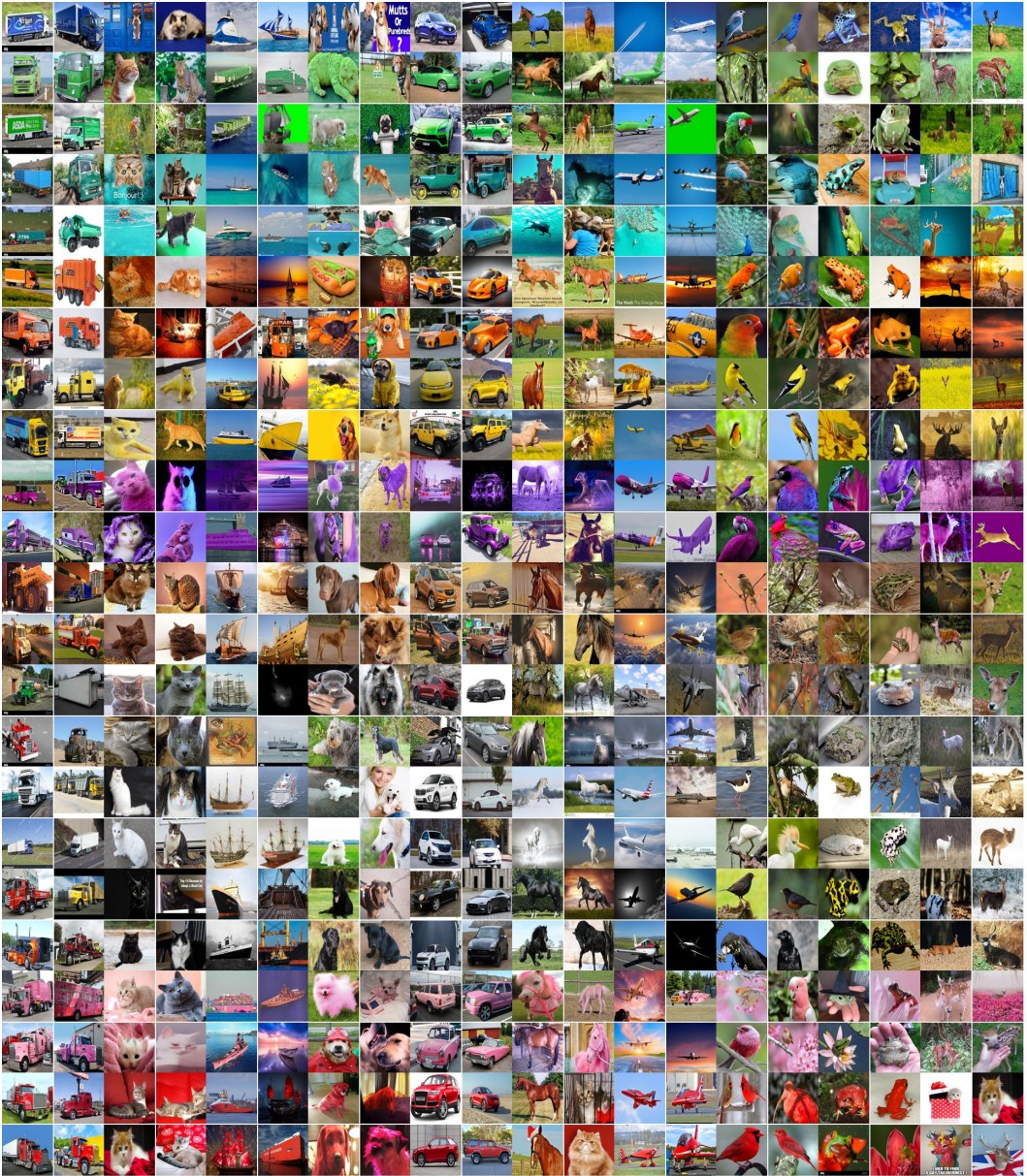

Figure 6: Images searched from Bing with class names as keywords and 12 color options.

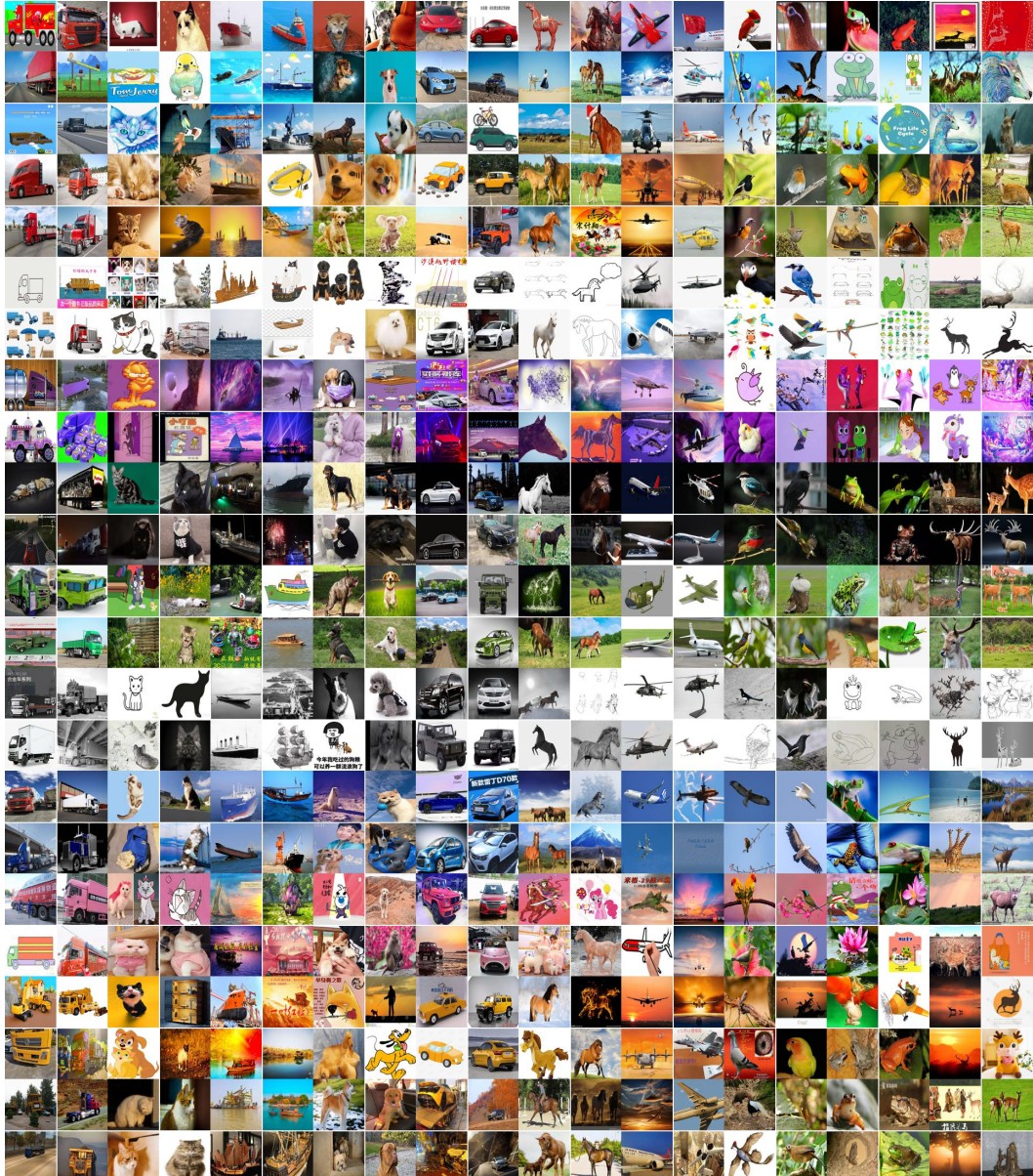

Figure 7: Images searched from Baidu with class names as keywords and 12 color options.

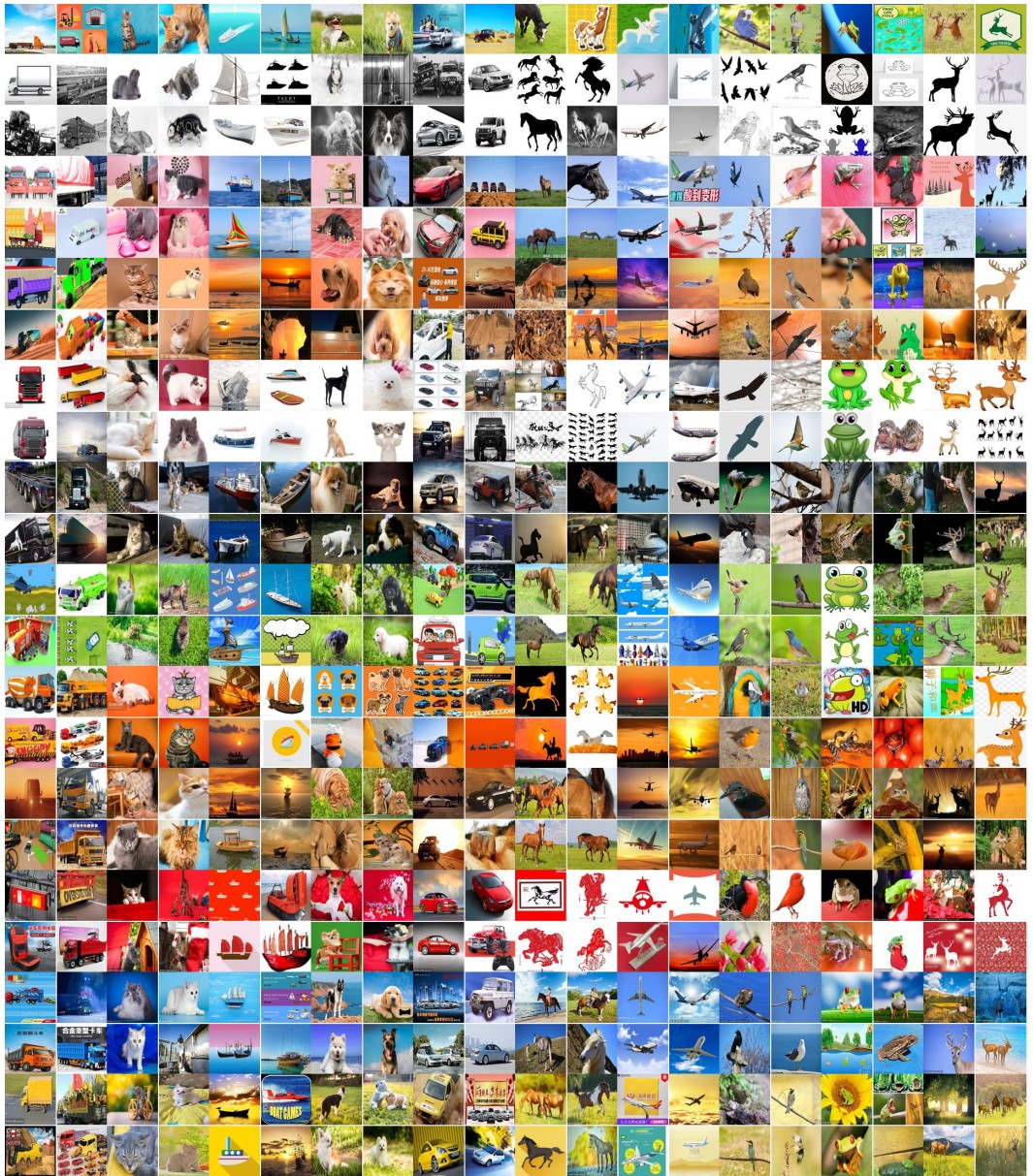

Figure 8: Images searched from 360 with class names as keywords and 12 color options.

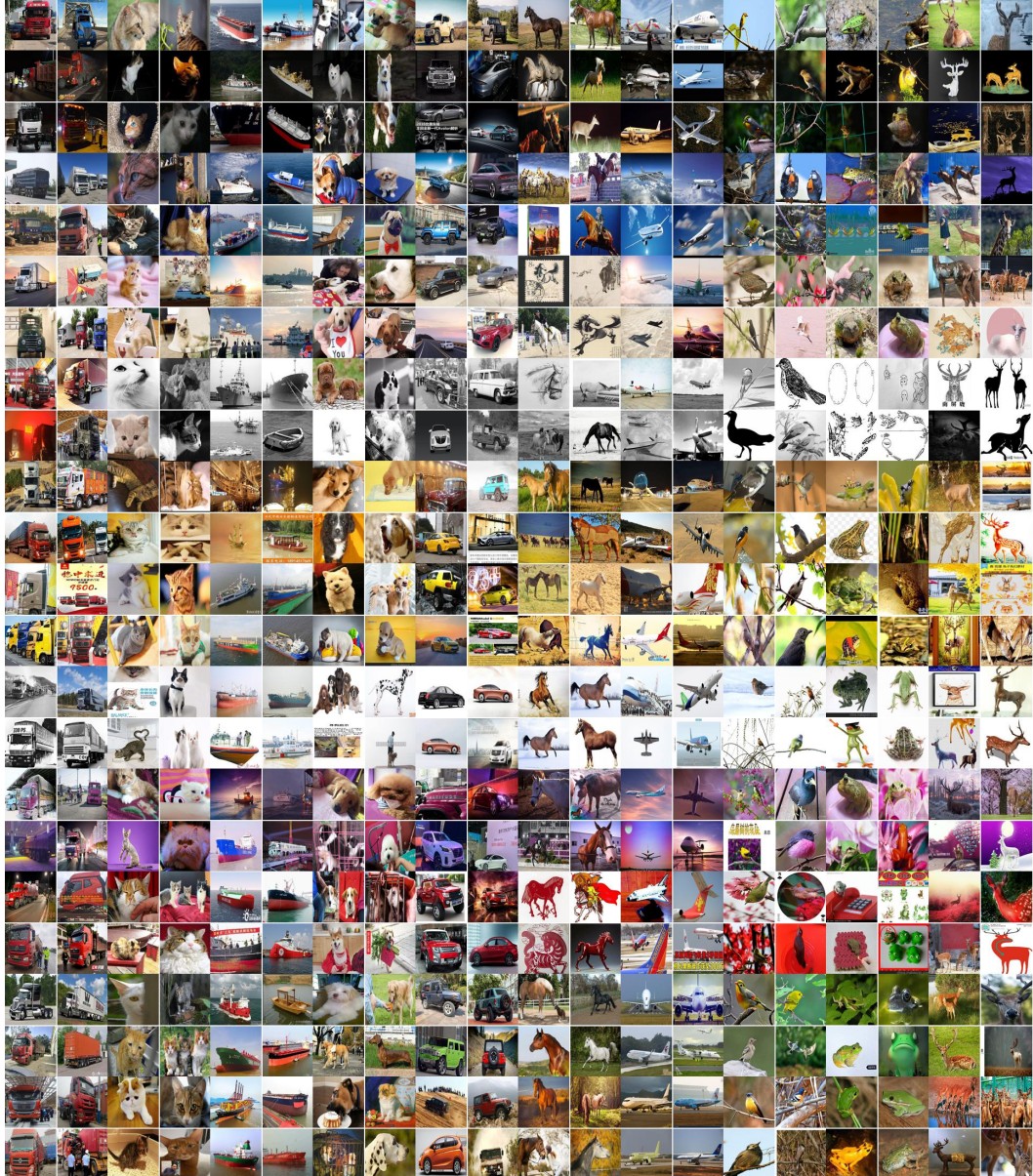

Figure 9: Images searched from Sogou engine with class names as keywords and 12 color options..

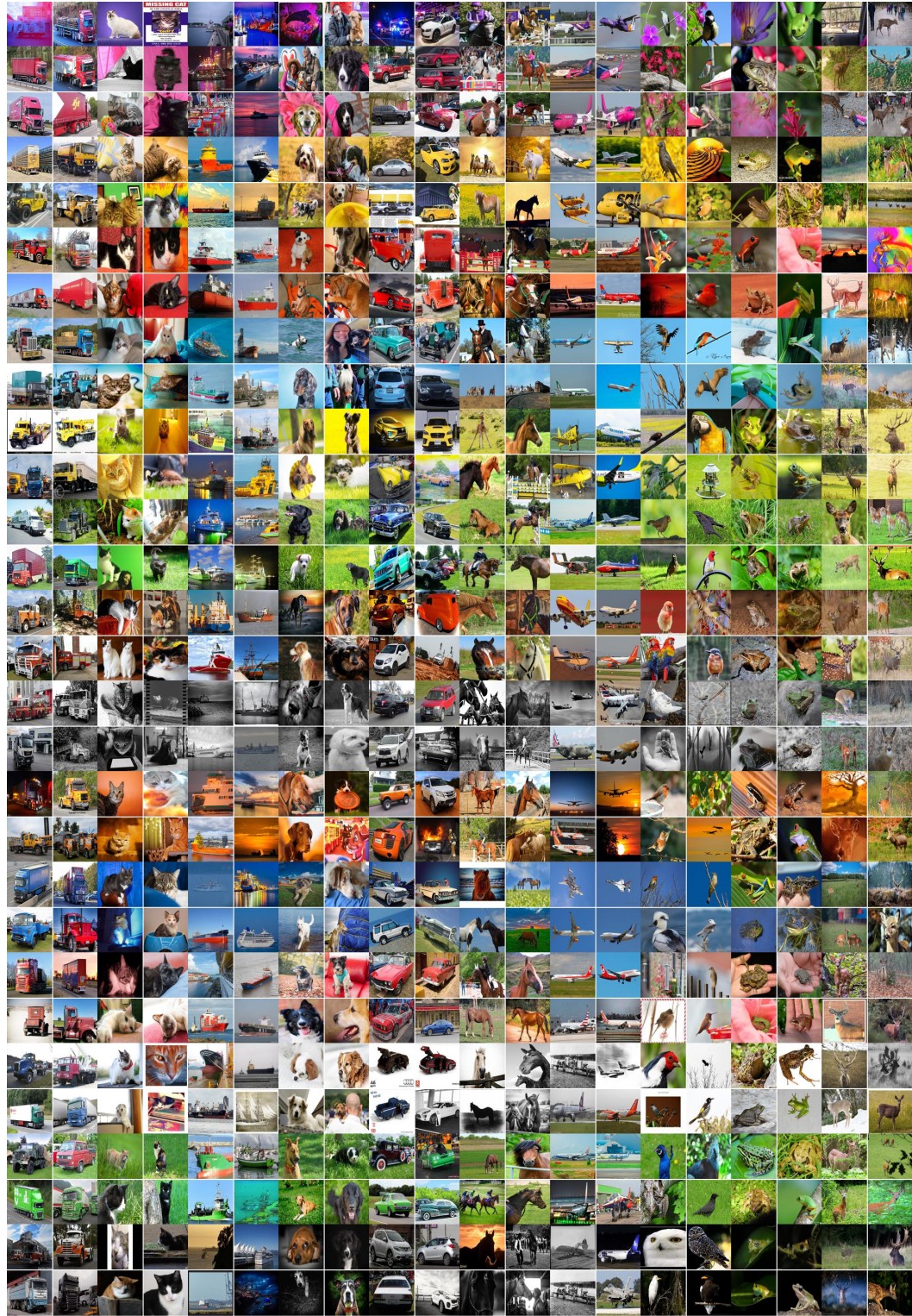

Figure 10: Images searched from Flickr engine with class names as keywords and 15 color options.

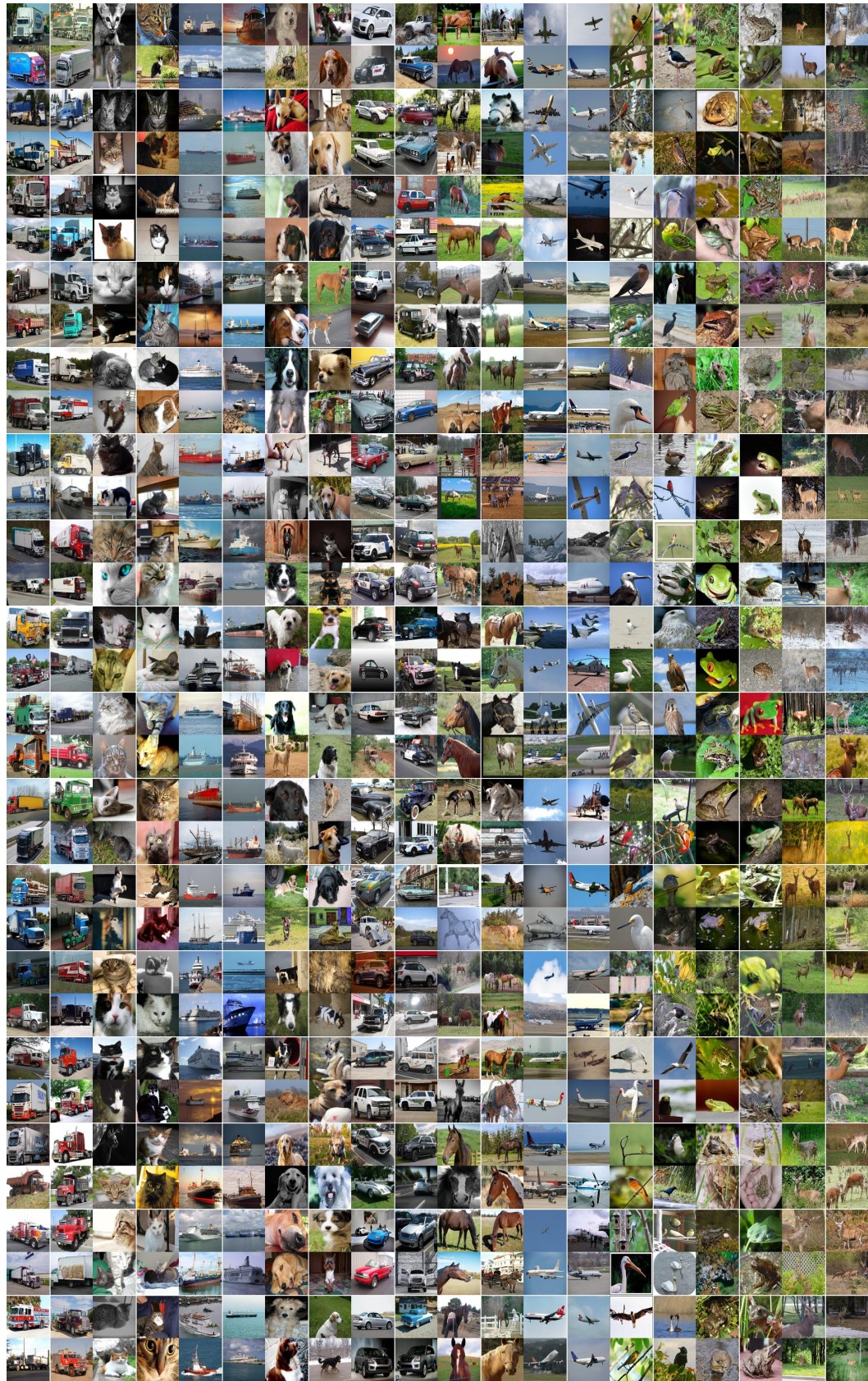

Figure 11: Images searched from Pexel with class names as keywords and 20 color options.

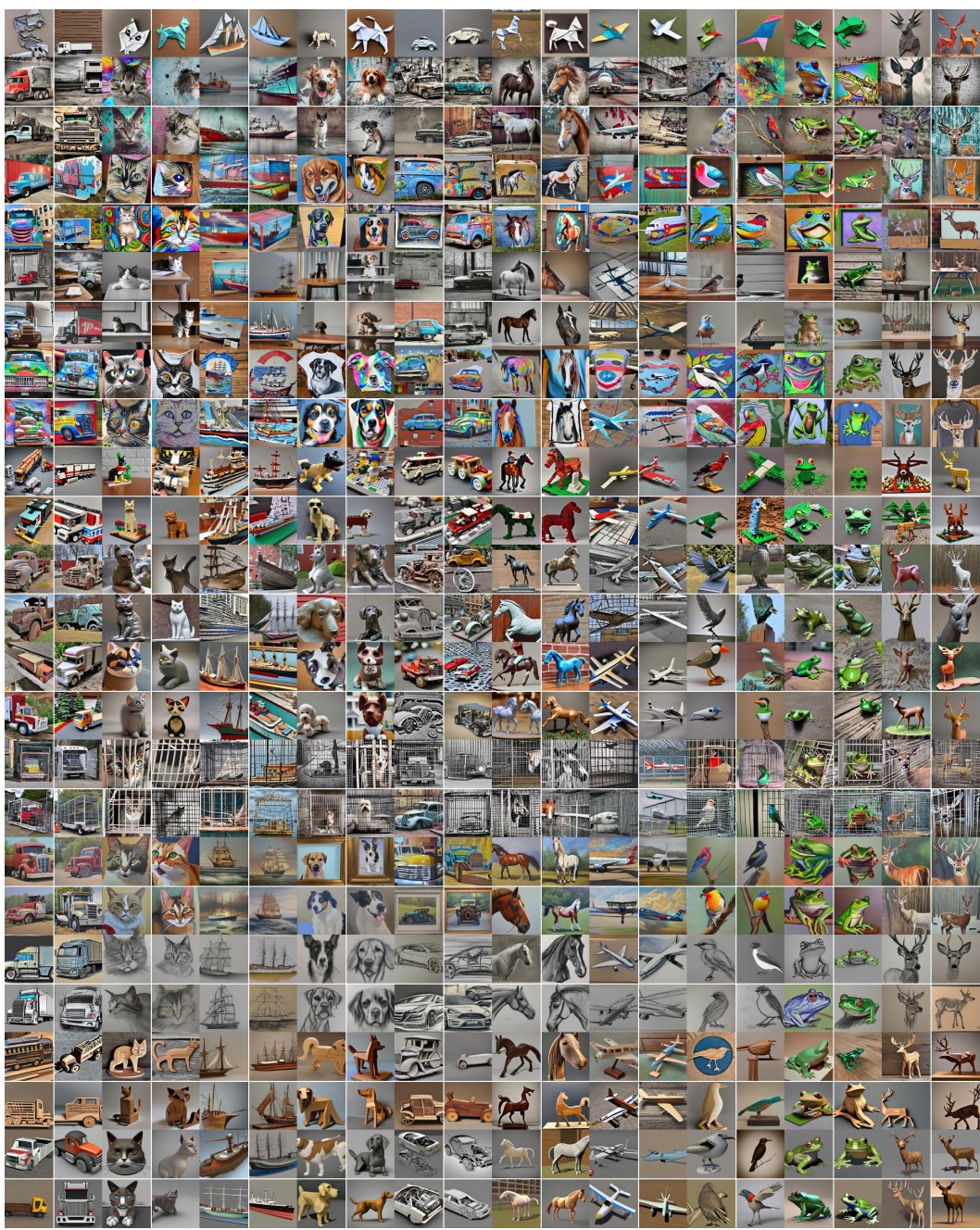

Figure 12: Images generated by the diffusion model using prompts of CIFAR-10-W diff. hard.

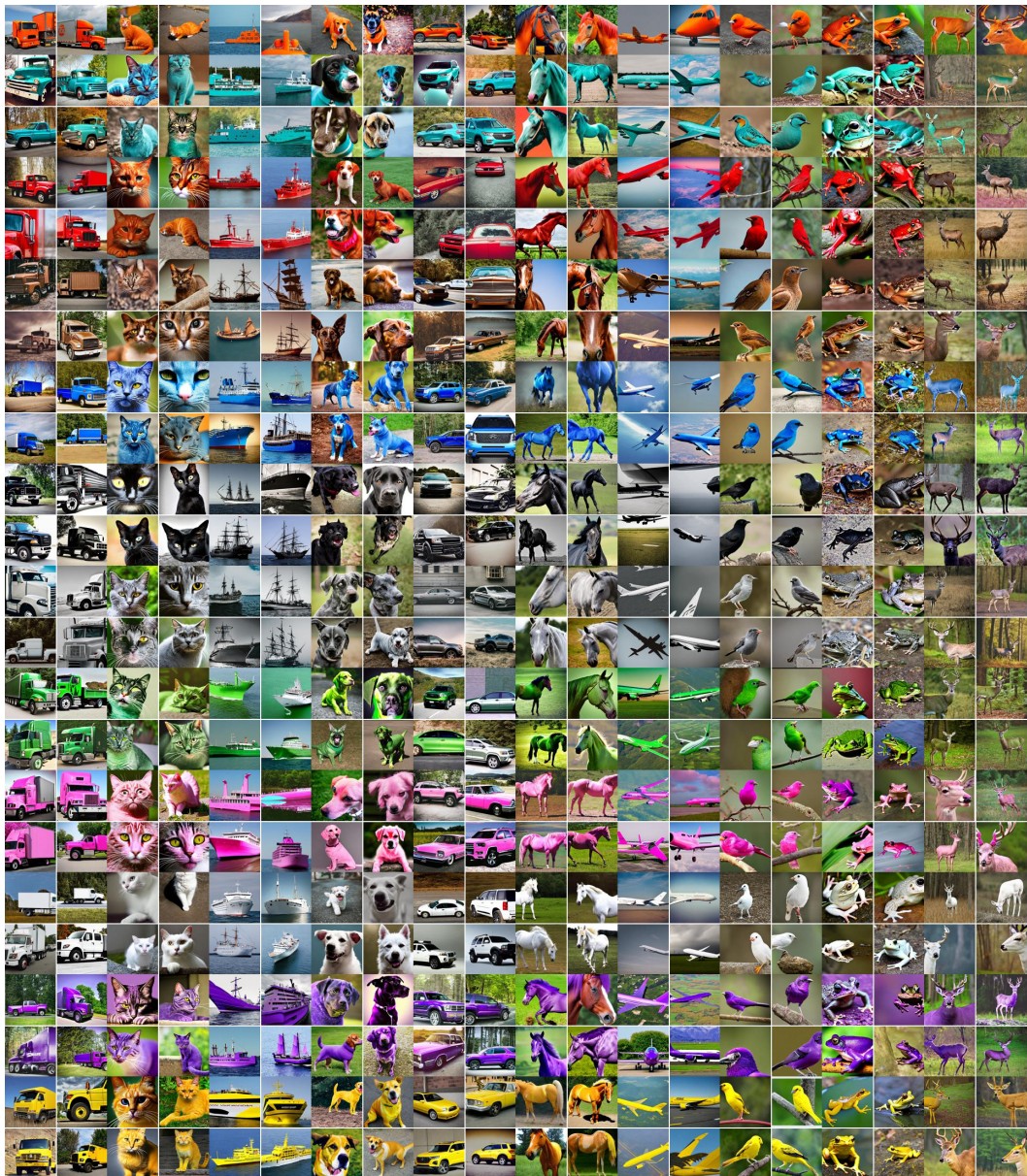

Figure 13: Iimages generated by the diffusion model using prompts of CIFAR-10-W diff..

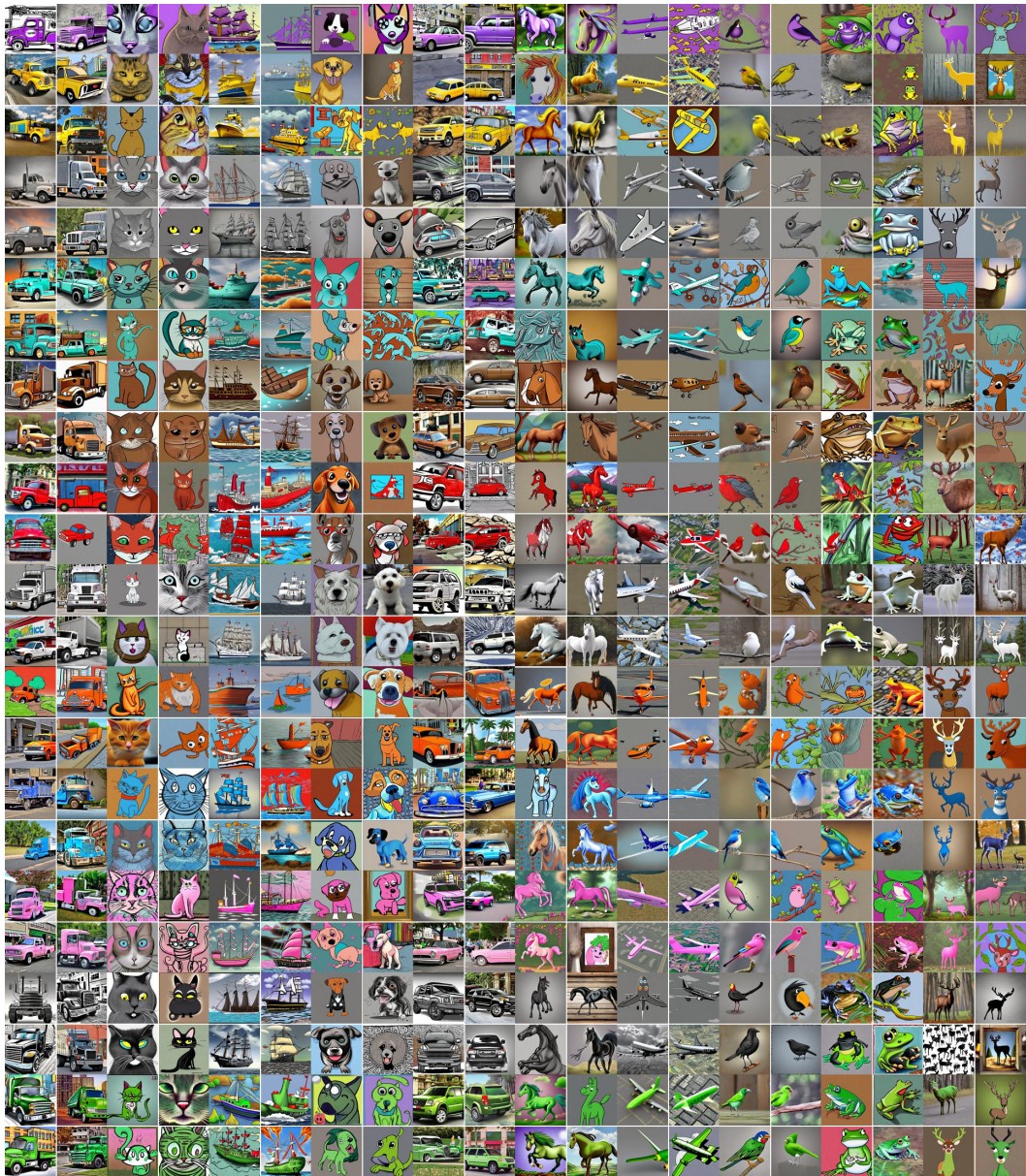

Figure 14: Images generated by the diffusion model using prompts of CIFAR-10-W diff. cartoon.