# OpenReview forum: "CIFAR-10-Warehouse: Broad and More Realistic Testbeds in Model Generalization Analysis"
_ICLR.cc/2024/Conference — ICLR 2024 poster_

### Official Review · Reviewer_6ZJ5 · 2023-10-14

**Soundness:** 3 good
**Presentation:** 3 good
**Contribution:** 3 good
**Rating:** 6
**Confidence:** 4

**Summary:**

The paper introduces CIFAR-10-Warehouse, a substantial dataset comprising 180 diverse datasets with images from the original CIFAR-10 categories, sourced from various sources including real-world searches and stable diffusion. CIFAR-10-Warehouse serves as a valuable resource for advancing research in model generalization analysis, accuracy prediction, and domain generalization. CIFAR-10-Warehouse creates a challenging testbed, shedding light on the complexities of model performance in diverse, real-world scenarios. Additionally, the paper highlights potential applications in fields such as learning from noisy data and out-of-distribution detection. This paper advances the understanding and evaluation of model generalization in machine learning research.

**Strengths:**

(1) The paper introduces a novel and extensive CIFAR-10-Warehouse dataset with a diverse collection of 180 datasets. The authors' approach of curating datasets from real-world searches and stable diffusion represents a novel and creative way to construct a comprehensive testbed for model generalization analysis.

(2) The paper maintains a high quality in dataset creation and experimentation. The dataset creation process appears thorough, involving both real-world image searches and diffusion model generation, while ensuring privacy and adhering to licenses. The experiments conducted on CIFAR-10-Warehouse are extensive, employing various methods and classifiers with detailed analysis of the results.

(3) The paper is generally well-written and organized, making it easy to follow. The methodology for dataset creation and evaluation tasks is clearly explained.

(4) CIFAR-10-Warehouse serves as a valuable resource for researchers, providing a unique dataset that covers a wide range of real-world scenarios and challenges the generalization abilities of machine learning models. The paper's exploration of potential applications in learning from noisy data, domain adaptation, and out-of-distribution detection highlights its significance in various domains.

**Weaknesses:**

(1) While the paper provides details about the data collection process, it lacks a discussion on potential biases and limitations introduced during data collection from search engines. Biases in search engine results can affect the diversity and representativeness of the dataset, which should be acknowledged and addressed.

(2) The paper focuses on domain generalization within the context of CIFAR-10-Warehouse. However, CIFAR-10 is indeed a relatively small dataset with low-resolution images and a limited number of categories compared to larger-scale and more diverse datasets like ImageNet or the Wilds[1]. This work does not extensively discuss how the findings from this dataset can be applied to real-world scenarios or other domains.

(3) This work could be strengthened by discussing potential real-world applications beyond the scope of image classification. Evaluating and extending the methods on datasets with different characteristics and applications would provide a more practical significance for real-world applications.

(4) While the paper introduces several domain generalization and accuracy prediction methods, it could benefit from including additional state-of-the-art baseline methods, e.g., GVRT[2], and VNE[3], for a more comprehensive comparison. This would help establish a clearer benchmark for the proposed methods.


[1] WILDS: A Benchmark of in-the-Wild Distribution Shifts

[2] Grounding Visual Representations with Texts for Domain Generalization. ECCV 2022

[3] VNE: An Effective Method for Improving Deep Representation by Manipulating Eigenvalue Distribution. CVPR 2023

**Questions:**

(1) Can you elaborate on the potential biases introduced during data collection from search engines or how did you ensure the collected data is diverse and representative?

(2) How might the findings from CIFAR-10-Warehouse generalize to other larger-scale datasets or tasks beyond image classification?

(3) Analyzing and discussing the failure modes of domain generalization would provide insights into scenarios where these methods might not work well. Can you discuss and provide examples of failure modes for the domain generalization and accuracy prediction methods on the CIFAR-10-Warehouse testbed?

(4)The paper mentions addressing limitations and publishing future versions of CIFAR-10-Warehouse but does not provide a concrete roadmap for future research. A discussion of potential future directions, and how the dataset can evolve would be insightful.

---

> ### Author Response · Authors · 2023-11-22
> **Response to Reviewer  6ZJ5 part1**
>
> We thank the reviewer for many insightful comments. We answer the questions in what follows. Please let us know if further clarification is needed.
>
> >**Q1: Can you elaborate on the potential biases introduced during data collection from search engines or how did you ensure the collected data is diverse and representative? Lacking a discussion on potential biases and limitations introduced during data collection from search engines. Biases in search engine results can affect the diversity and representativeness of the dataset, which should be acknowledged and addressed.**
>
> Thank you for your insightful comments. This paper has provided some related discussions, such as distribution bias between CIFAR-10-Cs and CIFAR-10-W and the influence of noise data, in the submitted paper. Following the suggestion of the reviewer, we would like to provide additional explanations and discussions.
>
> **Potential biases:** We agree with the reviewer that the dataset searched from search engines may be biased to the target sets and sets from different engines. However, our primary aim is to incorporate a wide range of distributions to evaluate the AccP and DG methods effectively. Hence, while these biases exist, they offer diverse perspectives for our evaluation, aligning with our objective to cover various distributions.
>
> **Diverse and representative:** The diversity of our collected datasets has been demonstrated through the results in our paper. For example, Tables 2 and 3 show the diversity of datasets by presenting varying accuracy prediction and DG results on the CIFAR-10-W. Fig. 2 and Fig. 5 show statistical distributions of different outcomes across the datasets. Each data point represents results on one set, indicating the variety within our data.  Regarding representativeness, we try to follow the natural distribution of each search engine without imposing additional restrictions. We believe that distributions mirroring real-world scenarios are invaluable for comprehensive evaluations.
>
> **Potential limitations:** Noisy data should be one of the most important limitations of collecting data from search engines. Section 5 has provided some discussion about it. To remove noisy data, we have asked human annotators to review data for several around, and then use the CLIP model to select out data with multiple labels together with human double-check. By doing this, we think most of the noise data has been removed. On the other hand, dataset imbalance may be the other limitation of using search engines because the databases of different engines are different. The balanced distribution of data in the dataset is similar to the data distribution on the engines. At the same time, we discussed the results, such as that shown in Fig.4 (A) and Table.8 (Appendix), of the data being sampled to be balanced in Sections 3.3 and E.
>
> We will incorporate the aforementioned discussions into our paper to enhance the overall understanding of the proposed CIFAR-10-W dataset.
>
> >**Q2: How might the findings from CIFAR-10-Warehouse generalize to other larger-scale datasets or tasks beyond image classification? CIFAR-10 is indeed a relatively small dataset with low-resolution images and a limited number of categories compared to larger-scale and more diverse datasets like ImageNet or the Wilds[1]. This work does not extensively discuss how the findings from this dataset can be applied to real-world scenarios or other domains.**
>
> We agree with the reviewer that CIFAR-10 is a relatively small dataset compared with some existing datasets, such as ImageNet. However, it is one of the most commonly used data in the community and is an important benchmark for AccP and DG algorithm evaluation. Our choice of CIFAR-10 is because of its simplicity and manageability, which makes it a suitable starting point for foundational research of AccP and DG tasks.
>
> We start by using CIFAR-10 as a basic reference to build CIFAR-10 to solve the limitation that existing test data have a small number of domains or use synthetic data for evaluation. Despite its limitations, the insights gained from CIFAR-10 can provide valuable preliminary understanding, which can be a foundation for analyzing more complex datasets. The findings of AccP and DG observed in CIFAR-10 can offer indicative trends that might be observed in larger datasets. For example, if a method cannot work well on the CIFAR-10 setup, it is hard to imagine it works well on a dataset with a large number of categories.
>
>
> In future work, we will try to build testbeds with datasets of more classes. We will include the above discussion in our revised paper on the potential application of our findings from CIFAR-10 to other real-world scenarios and more diverse domains.

---

> ### Author Response · Authors · 2023-11-22
> **Response to Reviewer  6ZJ5 part2**
>
> >**Q3: This work could be strengthened by discussing potential real-world applications beyond the scope of image classification. Evaluating and extending the methods on datasets with different characteristics and applications would provide a more practical significance for real-world applications.**
>
> Thank you for your kind suggestion. Indeed, while our current research primarily centres on image classification, there is a large potential for applying our findings to a wider range of real-world applications, such as AccP and DG on object detection and image retrieval. Furthermore, we will consider conducting studies in our future work on these applications. In future revisions of the paper, we will add discussions about other potential applications.
>
> >**Q4: This paper could benefit from including additional state-of-the-art baseline methods, e.g., GVRT[2], and VNE[3], for a more comprehensive comparison.**
>
> Thank you for this valuable suggestion. In response to your feedback, we have conducted additional experiments using the VNE method. The results are as follows, indicating VNE's performance across various subsets of the CIFAR-10-W dataset:
>
> | dataset | DF (37)     | KW (95)     | KWC (48)   | All (180)   |
> |---------|--------|--------|--------|--------|
> | VNE     | 0.9396 | 0.9113| 0.7517 | 0.8768 |
>
> Due to the time constraints of the rebuttal period, we were unable to complete experiments using GVRT. However, we have cited both GVRT and VNE (Min et al.,2022; Kim et al., 2023) as important baselines in our paper In the revised version, we will incorporate a comprehensive analysis that includes GVRT, further enriching our comparative study.
>
> >**Q5: Can you discuss and provide examples of failure modes for the domain generalization and accuracy prediction methods on the CIFAR-10-Warehouse testbed?**
>
> Thank you for this valuable suggestion. Our analysis of the CIFAR-10-W testbed has indeed shown new challenges encountered by both DG and AccP methods.
>
> As evidenced in Tables 2 and 6 of our paper, there is notable variability in the performance rankings of AccP methods when comparing CIFAR-10-W with CIFAR-10-Cs. Specifically, the rank correlation for models such as ResNet44, RepVGG-A0, and ShuffleNetv2 is only 0.481, highlighting a significant discrepancy in evaluations between CIFAR-10-Cs and CIFAR-10-W. An example of this is the performance of the Pred. s (0.9) method. While it ranks as the best AccP method for ResNet44 on CIFAR-10-Cs, it fails to maintain its superiority on CIFAR-10-W. This suggests that the generalizability or adaptability of Pred. s (0.9) may fail when transitioning from CIFAR-10-C to the more diverse CIFAR-10-W environment.
>
> To study the potential failure modes, we first compare the test performance on single source setting and multiple source setting. We observe that when multiple distinct source domains are available at the training stage, it is more likely to achieve higher improvement than that of a single source domain. Furthermore, we compare the test performance of ERM and SagNet under a three source domains regime. In the three source domain training setting, we have two diffusion model-generated domains and a photo-realistic domain.  By analysing the testing performance on three meta-categories (i.e., diffusion, KW, and KWC), we find that SagNet brings improvement on KWC (approx. +0.56\% on average), but degrades the ERM result on KW (approx. -0.59\% on average). This indicates that some domain generalization methods may be able to improve the generalization power of the model on domains that are largely different from training distribution, with the price of reducing some generalization power on similarly distributed domains.
>
> We will incorporate these discussions into the revised version of the paper and add additional discussion on the failure cases of the DG and AccP methods on CIFAR-10-W, providing valuable insights into their limitations and areas for improvement.
>
> >**Q6: A discussion of potential future directions, and how the dataset can evolve would be insightful.**
>
> We appreciate your insightful suggestion. In our paper, particularly within Section 5, we have provided some discussions about the potential impact of CIFAR-10-W in various fields, including the challenges of learning from noisy data. Building on this foundation, and considering the reviewers' feedback, we recognize the critical importance of developing specialized datasets, particularly in fine-grained areas. For instance, the development of fine-grained datasets, as well as those covering a broad spectrum of categories, would be highly beneficial. These datasets are essential not only for addressing specific research queries but also for fostering innovation by introducing more complex and diverse scenarios for model evaluation and testing. We will add further discussion on these potential future directions and the importance of specialized datasets in advancing the field in Section 5 of our paper.

---

> > ### Comment · Reviewer_6ZJ5 · 2023-11-23
> >
> > Thank you for the detailed response. I recommend including the additional experiments and discussions during the rebuttal in the paper. Based on the current content, I am inclined to accept this paper.

---

> > > ### Author Response · Authors · 2023-11-23
> > >
> > > Thank you very much for your reply. Your comments have been invaluable in enhancing the quality of our work. We will include the additional experiments and discussions in the revised paper, ensuring a more informative presentation.

---

### Official Review · Reviewer_tRct · 2023-10-30

**Soundness:** 3 good
**Presentation:** 3 good
**Contribution:** 2 fair
**Rating:** 6
**Confidence:** 3

**Summary:**

This paper proposes a new benchmark dataset for domain generalizations. Compared to the previous datasets, CIFAR10-Warehouse contains a lot more number of domains with both real-world images and images synthesized by stable diffusion. Extensive benchmarking and comparisons are conducted on this new dataset in terms of two generalization tasks.

**Strengths:**

1. This paper provides a new dataset with a much larger number of domains compared to existing domain generalization dataset. The idea of multi-domain dataset gives the researcher a new perspective on how to evaluate the domain generalization methods.
2. The experiments are quite extensive and gives some interesting insights on domain generalization.

**Weaknesses:**

1. Can the authors give more analysis and justification on why they divide different domains based on color and cartoon/no cartoon? since there are a lot of other ways to categorize different domains, such as other styles besides cartoon.
2. Can the authors give more empirical analysis on the advantage and difference of the proposed dataset compared to existing datasets. For example, are the performance comparison on CIFAR-10-W and existing datasets aligned? Are there any contradicted conclusions or new observations based on the experiment results on CIFAR-10-W?

**Questions:**

Please refer to the weaknesses.

**Details Of Ethics Concerns:**

May need to check the license of the images in the dataset.

---

> ### Author Response · Authors · 2023-11-22
> **Response to Reviewer  tRct part1**
>
> We thank the reviewer for many insightful comments. We answer the questions in what follows. Please let us know if further clarification is needed.
>
> >**Q1: Why does this paper divide different domains based on color and cartoon/no cartoon?**
>
> Great question. When dividing different domains, various factors can be considered, such as style, color, search engine features, and collection location. Our decision to focus on color and cartoony styles was driven by below reasons:
>
>  1) Color: we chose to use color as a distinguishing factor because all the search engines we utilized come equipped with color filtering options. This functionality allows us to naturally leverage these pre-existing data divisions for our analysis.
>
>  2) Cartoony Style: Cartoony images represent one of the most common styles beyond natural images. This makes them a valuable category for our study, providing a clear contrast with more realistic image styles. Meanwhile, there are also a large number of cartoon images on the Internet, making it easier for us to collect data.
>
> Furthermore, to diversify our dataset, we also used diffusion models to generate additional datasets in different styles, such as oil painting and sculpture. This helps enhance the range and depth of our domain-based analysis.
>
> We will integrate these discussions into our paper to provide a comprehensive understanding of our current findings. Additionally, will expand CIFAR-10-W in our future work to encompass a broader array of diverse distributions, enhancing its utility.
>
> >**Q2: Providing more empirical analysis on the advantage and difference of the proposed dataset compared to existing datasets.**
>
> Thanks for your kind suggestion. In our paper, we have provided experiments and empirical analysis for comparing CIFAR-10-W with CIFAR-10-Cs when they are used for evaluation. For example, Table 2 shows the results of using CIFAR-10 Cs and CIFAR-10-W respectively for AccP method evaluations, which are analysed in detail in Section 3.2. Fig. 2 (B) shows distributions of ground-truth accuracy and MAE values of method MS-AoL on 180 sets of CIFAR-10-W and 188 sets of CIFAR-10-Cs, respectively. we can observe that CIFAR-10-W has a wider range of dataset accuracy than CIFAR-20-Cs.
>
> During the rebuttal, we have provided new analyses in Section A of the Appendix. Fig. 7 shows the correlation between model accuracy and FD for both CIFAR-10-Cs and  CIFAR-10-W. Notably, CIFAR-10-W displays a broader range of FD values compared to CIFAR-10-Cs, indicating a larger variation in dataset distribution. For instance, CIFAR-10-W includes more sets with FD values exceeding 10. Moreover, the stronger correlation coefficients for CIFAR-10-Cs ($\rho = -0.93$ and $\tau = -0.77$) compared to CIFAR-10-W ($\rho = -0.81$ and $\tau = -0.62$) suggest that the distribution of CIFAR-10-Cs, which comprises simulated datasets, is less complex than the real-world data variant found in CIFAR-10-W. We also add t-SNE visualizations of variation within CIFAR-10-W data by color and by the search engine in Fig. 8, which shows that CIFAR-10-W provides a rich and complex dataset that extends beyond the variability of CIFAR-10.
>
> In addition to the above new content, we will try to provide further analysis to enrich our paper with additional insights after the rebuttal process.

---

> ### Author Response · Authors · 2023-11-22
> **Response to Reviewer tRct part2**
>
> >**Q3: Are the performance comparison on CIFAR-10-W and existing datasets aligned? and Are there any contradicted conclusions or new observations based on the experiment results on CIFAR-10-W?**
>
> Thank you for this important question. Indeed, our analysis reveals some inconsistencies in the performance of methods when evaluated on CIFAR-10-W as compared to existing datasets like CIFAR-10-Cs. For instance, Tables 2 and 6 present differing rankings of AccP methods across these two datasets. Specifically, for AccP, we observed the following rank correlations across different models:
>
> |             | ResNet44 | RepVGG-A0 | ShuffleNetv2 |
> | ----------- | -------- | --------- | ------------  |
> | correlation $\tau$ |  0.066    | 0.733   | 0.644       |
>
> These correlations suggest a lack of strong rank consistency between performances on CIFAR-10-W and CIFAR-10-Cs. This is especially evident for ResNet44, where the correlation is notably low.
>
> Moreover, compared with other domain generalization benchmark datasets (usually involving less than six domains), our proposed benchmark dataset empowers researchers to conduct more statistically meaningful analyses with 180 domains. Specifically, from Table 3, we can see that with the increased number of training domain, the test performances of model tend to be more stable. For example, the variance of test performance drops from approximately 3.9\% for single source domain to 0.67\% for four source domains. This outcome contrasts with results reported in other studies, indicating that CIFAR-10-W may present a more challenging and realistic testing environment by involving more test sets from different domains.
>
> These observations suggest that conclusions drawn from CIFAR-10-W could differ from those based on existing datasets. In particular, CIFAR-10-W tends to magnify performance differences among methods, providing clearer insights into their robustness for AccP. For DG tasks, the lack of noticeable improvement over the ERM baseline in our dataset suggests that existing DG methods may need further refinement to handle the diverse and complex scenarios presented by CIFAR-10-W.
>
> In summary, CIFAR-10-W serve as a new diverse benchmark dataset, offering a wide array of test environments that are helpful for future research in AccP and DG methodologies.  We will include the discussions highlighted above in the revised version of our paper, further indicating the relevance and utility of CIFAR-10-W for future studies.

---

### Official Review · Reviewer_VNrf · 2023-10-31

**Soundness:** 2 fair
**Presentation:** 4 excellent
**Contribution:** 3 good
**Rating:** 6
**Confidence:** 2

**Summary:**

The paper proposes a dataset called CIFAR10-W. I detail the construction of the dataset as the creation of the testbed seems to follow standard practices in DG, but done over a wide model zoo of classifiers.

**Construction**
A subset consists of all 10 classes of CIFAR10 with one colour and from the same source (all images are 224x224 but resized to 32x32 in experiments)

It consists of the following 180 subsets by (a-d):

*a) Querying 4 search engines*: Google, Bing, Baidu, 360 with a total of 12 colours (Google and Bing differing on 1-2 colours) using the two queries:
1) category_name
2) category_name cartoon
This creates {2 queries} x {12 colours} x {4 engines} = 96 subsets

*b) Querying other search engines*: Sogou, with the same 12 colours as Baidu/360, Pexe with 20 colours and Flickr with 15 colours. No cartoons queried from here.
This creates 12+20+15 = 47 subsets.

This creates 143 subsets which contain real images in total. 95 of these search by keywords (CIFAR-10-W KW) and 48 belong to the additional cartoon domain (CIFAR-10-W KWC).

*c) Querying Stable Diffusion 2.1*: This is done the two prompts for the same 12 colours as Baidu/360/Sogou for synthetic versions of the real data:
1) high quality photo of {color}{class name}
2) high quality cartoon photo of {color}{class name}

This creates 12 x 2 = 24 subsets

*d) 13 subsets created by using special prompts with SD-2.1*: Prompts given in Table 4 -- with background, context where target objects do not naturally co-exist.

Additionally:
- Cleaned annotations and details on previously labeled incorrect labels.

**Comparison**: CIFAR10-Cs benchmarks refer to a collection of CIFAR10-C, 10.1 and 10.2.

**Testbed**: Testbed comprises of two tasks:
- Model Accuracy Prediction using Unlabeled Test Sets – I am unfamiliar with this task
- Domain Generalization, similar to DomainBed – I am vaguely familiar with this task

It is important to note that they performed evaluation over a wide range of models.

**Strengths:**

**S1) Good benchmark [Critical]**: I like the proposed dataset, which is one of the core contributions as:
(1) It is a realistic domain shift in contrast to synthetic corruptions
(2) It is large-scale and has 224x224 images (albeit is a 10 class classification problem)
(3) It has a large number of domains (36-180 domains)
(4) It has cleaning annotations available!

**S2) Comprehensive evaluation [Critical]**: The paper seems to have tested methods across a lot of different models and across different subsets of the C10-W on two different tasks. Appendix sections were quite a delight to go through.

**S3) DG results quite cleverly done [Important]**: I liked the setup using alternative sources like Yandex, etc. These aspects seemed quite thoughtful to me

**Weaknesses:**

**W1) [Critical] Why is MAE of prediction scores the main metric reported in the paper for comparing methods in Table 2?**

Why I ask this (my understanding, could be caused by misinterpretation):
- MAE of prediction scores, rather than prediction scores seems like a bad measure as indicated by Figure 2 and elaborated in Fig 9 – no correlation to accuracy, while Figure 7 shows prediction scores have high spearman rank order correlation (a strong measure of correlation!).
- Hence, I can discern very little about the predictiveness of the accuracy from the MAE score as they’re uncorrelated!

**W2) [Important] Conclusions made from Task 1 need improvement-- Similar but to a lesser degree for Task 2.**

Details for Task 1:

*(C1) This benchmark is a harder benchmark compared to CIFAR-10-Cs.*

- Corruptions can be made harder by increasing the magnitude easily, increased hardness as the primary feature of the testbed seems weird. I would be interested in rather highlighting whether networks don't capture certain aspects introduced here.

*(C2) Predictions are more consistent across classifiers here.*

- One can easily argue that predictions are consistent here because the shift is not varied– it simply changes colours rather than the diverse, varied corruptions studied in CIFAR-10-C  benchmarks.
- Note that this does have a distribution shift across cartoons and SD classifier, but SD classifier results are varying across classifiers too!

**W3) [Important] Little analysis of the dataset itself, far more focus on task/models**

- While this weakness is vaguely stated, I have fleshed out some components in Q1 to concretely ask what analysis would be helpful from my view.
- However, I do not work in either of the fields so it is hard for me to accurately ask, but there seems to little analysis done which is concerning.

**Questions:**

**Q1) How many visually distinct domains exist within these 180 subsets? Specifically, how different are images sampled from different search engines?**

I found 36 distinct domains:
- The 12 colour palettes seem like distinct domains
- Cartoon, real-world images and SD images seem distinctly different

By visual inspection by me, images across different search engines look similar (ordering the images by the same colour would have made my job, and an interested readers’ easier in Figures in Appendix F). Note that popular datasets are also collected from different search engines but treated as one domain-- do different engines introduce a noticeable shift?

*Counterclaim to the point: Significant drop in performance in Baidu and 360.*
- Is it due to a lesser number of images or because of the domain gap? [Maybe separating the search engines in Figure 1 would be have been very informative for a reader, alongside more analysis of drift]
- I suspect those engines have fewer images which might be causing the accuracy drop.

The benchmark seems to pitch that there are 180 clearly distinct domains, would be concerning if rather there are only 36 visually distinct domains.

**Q2) What all components would be released publicly?**

I presume the dataset, along with the cleaned annotations and licenses would be released.
- Would the scraping code be released?
- Would the code for classifiers tested be released?
- Will the trained models/features be released?

Could the authors address the weaknesses, these questions and check if I missed pointing out some strengths? That would help me make a more balanced evaluation. I like the benchmark itself but the experiments and conclusions drawn from it need improvement in my view. Note that I am not familiar at all with Task 1 and only vaguely familiar with Task 2, indicated in my confidence.

---

> ### Author Response · Authors · 2023-11-22
> **Response to Reviewer VNrf part1**
>
> We thank the reviewer for many insightful comments. We answer the questions in what follows. Please let us know if further clarification is needed.
>
> > **Q1: Why is MAE of prediction scores the main metric reported in the paper for comparing methods in Table 2? The reason for the concern: Fig. 9 shows there is no correlation between MAE and accuracy, while Fig. 7 shows that prediction scores have a high Spearman rank order correlation with ground truth accuracy.**
>
> Thank you for your question. In the Accuracy Prediction (AccP) task, existing research often employs Mean Absolute Error (MAE) or Root Mean Square Error (RMSE) as the primary metrics. Consistent with this norm, we use MAE in our paper. The objective in AccP is to predict the classifier accuracy $ \hat{Acc}_i, i = 1, \ldots, n $, across $n $ given unlabeled sets. The MAE, defined as
>
> $$
> \text{MAE} = \frac{1}{n}\sum_{i=1}^{n} |\text{Acc}_i - \hat{\text{Acc}}_i|,
> $$
>
> effectively indicates the gap between predicted accuracies and the ground-truth accuracies.
>
> There is a misinterpretation regarding the results presented in Fig. 2 (also in Figs. 7 and 9). Fig. 2 (A) illustrates the correlation between prediction scores $ \hat{\text{Acc}_i} $ and actual accuracy $ \text{Acc}_i $ Fig. 2 (B) illustrates the relationship between the actual accuracy $ \text{Acc}_i $ and the prediction error $ |\text{Acc}_i - \hat{\text{Acc}}_i| $ of the AccP method on different datasets. It is expected that there may not be a clear correlation, as the value of prediction error is not necessarily directly influenced by the ground-truth accuracy of a dataset.
>
> We hope the above explanation can help the reviewer have a clear understanding of MAE.
>
>
> > **Q2: Conclusions made from task 1 need improvement**
>
>
> **1) This benchmark is a harder benchmark compared to CIFAR-10-Cs. The increasing hardness of the testbed seems weird. It is more interesting to highlight whether networks don't capture certain aspects introduced here.**
>
> We agree with the reviewer that the goal of building a new testbed is not only to increase the difficulty of the dataset but to propose test data that are more indicative of algorithm performance. The primary aim of developing CIFAR-10-W is to offer a testbed that better reflects the complexities in real-world scenarios and provides a more comprehensive evaluation of algorithms.
>
> Specifically, the proposed CIFAR-10-W can provide a more comprehensive evaluation of AccP methods by including more challenging sets. The challenge here not only means that the data set is more difficult but it is reflected in many aspects, such as more diverse data and more different predicted models. Meanwhile, the evaluation on CIFAR-10-W datasets indeed shows more obvious performance differences of existing methods, which is helpful for highlighting the performance of different AccP methods.
>
> For example, the ranges of y-axis of Fig. 2 (B) show that CIFAR-10-W have more diverse test sets for AccP algorithms evaluation, which helps indicate the robustness of AccP methods under variance environments. Meanwhile, the evaluation of diverse models provides a more accurate comparison of AccP methods when facing different target classifiers. For example, the standard deviations (SD) of different methods for ResNet44 on the CIFAR-10-Cs and CIFAR-10-W datasets, respectively, are **0.81** and **1.80**. Obviously, the results difference can be magnified on CIFAR-10-W.
>
> In conclusion, CIFAR-10-W is not only a hard testbed, it is a more diverse and indicative testbed. We will include the above discussion in our revised paper.

---

> ### Author Response · Authors · 2023-11-22
> **Response to Reviewer VNrf par2**
>
> **2) One can easily argue that predictions are consistent here because the shift is not varied – it simply changes colours rather than the diverse, varied corruptions studied in CIFAR-10-C benchmarks.**
>
>  Thank you for your comments. We respectfully disagree with the comment that the observed consistency is due to a lack of varied shifts in the data.
>
>  In our paper, Fig. 2 and 5 illustrate the distribution of ground truth accuracy for ResNet44 and ERM accuracy on CIFAR-10-W. These figures clearly demonstrate varied performance across our datasets which can indicate the varied shifts among datasets.
>
> Additionally, during the rebuttal phase, the newly added Fig. 7 in the Appendix highlights the distribution differences between datasets from CIFAR-10-Cs and CIFAR-10-W by showing their FD with the CIFAR-10 dataset. It is evident from this analysis that CIFAR-10-W has more datasets with Fréchet Distance (FD) values greater than 10, suggesting a greater deviation from the CIFAR-10 dataset. In contrast, most datasets in CIFAR-10-Cs exhibit smaller FD values, indicating closer similarity to CIFAR-10. Meanwhile, the newly added Fig. 8 in the Appendix shows t-SNE visualizations of CIFAR-10-W and CIFAR-10, which indicates the diversity of images from the same color or different search engines.
>
> Furthermore, we have calculated the rank correlations of different method performances on two subgroups of data from CIFAR-10-W:
>
> |             | DF.h & DF | Goog. & Flic. | Bing C & Baid.C |
> | ----------- | --------- | --------------- | ---------------------------- |
> | correlation | 0.3050     | 0.4384          | 0.648                        |
>
> There is no obvious correlation, which indicates the variance between these datasets. As for the consistency we mentioned in the paper is caused by the average statistics on \textbf{all} datasets. When methods are evaluated on \textbf{a large number of diverse datasets}, the rank of performance of methods will gradually be stable.
>
> Following the valuable suggestions of the reviewer, we will revise our conclusions and incorporate the above-discussed points into the revised version of our paper.
>
> >**Q3: Little analysis of the dataset itself, far more focus on task/models**
>
> Thank you for your constructive suggestion. In response to your feedback, we have enriched our paper with more analysis of the CIFAR-10-W dataset. The new contents are in Appendix Section A.
>
> Specifically, we have included comparisons of distribution ranges between CIFAR-10 and CIFAR-10-W, highlighting the diversity present within CIFAR-10-W. Additionally, we present an analysis of the variance within datasets categorized by a single color option, illustrating the intricate differences that exist even within seemingly similar groupings.
>
> We believe this added analysis will provide readers with a more comprehensive understanding of the CIFAR-10-W dataset, showing its complexity. This information should further clarify the relevance and applicability of CIFAR-10-W for AccP and DG evaluation.
>
>
> >**Q4: How many visually distinct domains exist within these 180 subsets? Specifically, how different are images sampled from different search engines?**
>
> Thank you for this insightful question. To illustrate the visual distinction between domains, we have included a t-SNE visualization in Fig. 8 (B) of the Appendix. This figure displays the distribution of bird images from CIFAR-10-W in relation to those from CIFAR-10. The t-SNE plot reveals that CIFAR-10-W encompasses a substantial diversity of images. Each search engine within CIFAR-10-W contributes uniquely distributed data, indicating that the images sampled from different engines are indeed distinct. The spread of data points in the visualization reflects the individual characteristics of each engine's dataset, thereby suggesting that a considerable number of visually distinct domains exist within these 180 subsets. This diversity is critical for evaluating the robustness and generalizability of computational models across a wide range of real-world scenarios.

---

> ### Author Response · Authors · 2023-11-22
> **Response to Reviewer VNrf part3**
>
> >**Q5: Significant drop in performance in Baidu and 360. Is it due to a lesser number of images or because of the domain gap?  I suspect those engines have fewer images which might be causing the accuracy drop.**
>
> We appreciate the reviewer's point, but our analysis suggests a different conclusion. The additional results presented in Table 8 of the Appendix, which are based on the CIFAR-10-W balanced subset, show that performance drops for Baidu and 360 remain significant even after adjusting for dataset size. This indicates that the number of images is not the primary factor affecting performance.
>
> To further substantiate this, we calculated Kendall's rank correlation coefficient between the number of images from each engine and the performance of various methods. The average correlation was found to be **0.3730**, which suggests only a weak relationship between dataset size and performance across different engines. For domain generalization (DG), the Kendall's rank correlation is similarly low at **0.2355** when comparing the number of images to the standard deviation (SD) of results, reinforcing the conclusion that the performance drop is not due to the quantity of images.
>
> We will include the above-detailed discussion of these analyses in the revised version of our paper to clarify the factors contributing to the differences in performance.
>
> >**Q6: Separating the search engines in Figure 1 would be have been very informative for a reader, alongside more analysis of drift.**
>
> Great suggestion. We have added a new figure of statistics on search engines in Fig. 6 of the APPENDIX. Meanwhile, some analysis of data from different engines is provided. We believe these additions will enhance the reader's understanding of the dataset's composition and the role of different search engines in contributing to its diversity.
>
> >**Q7: The benchmark would be concerning if there are only 36 visually distinct domains.**
>
> We appreciate the reviewer's perspective but would like to offer a different viewpoint. We believe that there are more than 36 visually distinct domains in our dataset. This is because, for each color category, the data from different search engines show significant diversity. We visualize the distribution of different colors and engines during rebuttal. The t-SNE visualizations shown in Fig. 8 of the Appendix clearly demonstrate that the distribution of these images is quite varied, exhibiting a range that is broader than what we observe in CIFAR-10. This diversity justifies our decision not to merge data of the same color from different engines.
>
> Furthermore, the color schemes across different engines are not uniform. To maintain objectivity and avoid the introduction of human bias, we chose not to manually merge data of the same color from various engines in CIFAR-10-W.
>
> Regarding the number of domains, we do not necessarily consider one option—180 or 36 domains—superior to the other. Each approach provides insights into different aspects of the methods being evaluated. We use 180 domains to keep the real-world data distribution native and to prevent the introduction of any subjective bias. Using 36 domains may be able to provide more complex domains for evaluation.
>
> We will add the above discussion to our paper and will explore different domain division ways in our future work.
>
>
> >**Q8: What all components would be released publicly?**
>
> Thank you for your question. We will release all key components of this paper to the public, including 1) the datasets of CIFAR-10-W, 2) all training sets used in Domain Generalization evaluations, 3) the codes for all the methods, and 4) all the models evaluated for AccP in the paper.

---

### Official Review · Reviewer_Xmsr · 2023-11-03

**Soundness:** 4 excellent
**Presentation:** 4 excellent
**Contribution:** 4 excellent
**Rating:** 8
**Confidence:** 4

**Summary:**

This paper constructed a thorough dataset for OOD evaluation based on CIFAR-10. They also provide a benchmarking analysis for a thorough list of accuracy prediction and domain generalization methods, which reveals interesting findings including identifying difficult settings that current methods fails. They also pointed out other directions that this dataset might be useful including denoising, unsupervised domain adaptation and OOD detection.

**Strengths:**

The paper is cleanly written and easy to read. The contribution can be pretty beneficial to the field as a thorough and high-quality benchmark dataset is the foundation for methods improvements, not to mention that OOD is a crucial problem in the field. The dataset built in this paper is quite thorough and high quality in my opinion as it includes not only many more domains compared with previous efforts but also includes state-of-art generative methods as well as real-world data for the dataset build. The author also provides interesting experiments that point out the limit of current state-of-art accuracy prediction as well as domain adaptation methods, which can certainly inspire corresponding methods improvements to be developed in the future.

**Weaknesses:**

Only want to point out this one typo: you seem to have an unfinished sentence at the last line of page 6.

**Questions:**

I am curious whether a finer-grained dataset like CIFAR-100-warehouse can be built as well. I know that many current state-of-art can suffer at finer-grained classification tasks. It could be interesting future work.

---

> ### Author Response · Authors · 2023-11-22
> **Response to Reviewer Xmsr**
>
> We thank the reviewer for many insightful comments. We answer the questions in what follows. Please let us know if further clarification is needed.
>
> > **Q1: One typo: there is an unfinished sentence at the last line of page 6.**
>
> Thanks. We have corrected the errors in the revised version and double-checked the paper to avoid typos.
>
> >**Q2: Is it possible to build finer-grained dataset like CIFAR-100-warehouse?**
>
> Thank you for your insightful question. We think it is feasible to build a finer-grained dataset like CIFAR-100-warehouse, but it will face increased challenges and need more effort. For example, if we use the way of building  CIFAR-100-warehouse to build a finer-grained dataset, we will face  challenges:
>
> 1) lower quality web tags, as fine-grained data often require more specialized knowledge.
>
> 2) more data cleaning efforts compared to generic classification tasks due to the fine-grained classification.
>
> Despite these obstacles, we believe the construction of such a dataset is achievable. We will consider this in our future work.

---

### Meta-Review · Area_Chair_Rvhg · 2023-12-12

**Metareview:**

This paper provides a new benchmark called CIFAR-10-Warehouse, designed to be useful for research on domain adaptation and domain generalization. The dataset is constructed by querying four different search engines in 2 different query templates and additionally specifying one of 12 different color options. The authors additionally complement the search-engine-derived data by prompting diffusion models. In all this yields 180 "domains". While the data still suffers some of the problems that broadly frustrate the benchmark-driven approach to domain adaptation, it seems like a useful tool for researchers and can offer a complementary view to the more contrived semi-synthetic datasets like CIFAR-C that have been the focal point of past empirical research in this area. The reviewers all appreciated the work, and the authors took the time to compose thorough replies to all reviews, after which the reviewers all appeared satisfied.

**Justification For Why Not Higher Score:**

I believe that this is a useful contribution but not that it requires a talk to communicate its contribution.

**Justification For Why Not Lower Score:**

N/A.

---

### Decision · Program_Chairs · 2024-01-16

Accept (poster)